# Changes of air pollutant emissions in China during two clean air action periods derived from the newly developed Inversed Emission Inventory for Chinese Air Quality (CAQIEI)

Lei Kong[1,3], Xiao Tang[*,1,3], Zifa Wang[*,1,3,4], Jiang Zhu[1,2], Jianjun Li[5], Huangjian Wu[1,3], Qizhong Wu[6], Huansheng Chen[1,3], Lili Zhu[5], Wei Wang[5], Bing Liu[5], Qian Wang[7], Duohong Chen[8], Yuepeng Pan[1,3], Jie Li[1,3], Lin Wu[1,3], and Gregory R. Carmichael[9]

[1]State Key Laboratory of Atmospheric Boundary Layer Physics and Atmospheric Chemistry (LAPC), Institute of Atmospheric Physics, Chinese Academy of Sciences, Beijing 100029, China
[2]CAS-TWAS Center of Excellence for Climate and Environment Sciences (ICCES), Institute of Atmospheric Physics, Chinese Academy of Sciences, Beijing 100029, China
[3]College of Earth and Planetary Sciences, University of Chinese Academy of Sciences, Beijing 100049, China
[4]Center for Excellence in Regional Atmospheric Environment, Institute of Urban Environment, Chinese Academy of Sciences, Xiamen 361021, China
[5]China National Environmental Monitoring Centre, Beijing, 100012, China
[6] College of Global Change and Earth System Science, Faculty of Geographical Science, Beijing Normal University, Beijing 100875, China
[7]Shanghai Environmental Monitoring Centre, Shanghai, 200030, China
[8]Guangdong Ecological Environment Monitoring Centre, National Key Laboratory of Regional Air Quality Monitoring for Environmental Protection, Guangzhou 510308, China
[9]Center for Global and Regional Environmental Research, University of Iowa, Iowa City, IA 52242, USA

*Correspondence to*: Xiao Tang (tangxiao@mail.iap.ac.cn) and Zifa Wang (zifawang@mail.iap.ac.cn)

## Abstract

A new long-term emission inventory called the Inversed Emission Inventory for Chinese Air Quality (CAQIEI) was developed in this study by assimilating surface observations from the China National Environmental Monitoring Centre (CNEMC) using the ensemble Kalman filter and the Nested Air Quality Prediction Modeling System. This inventory contains the constrained monthly emissions of $NO_x$, $SO_2$, CO, primary $PM_{2.5}$, primary $PM_{10}$, and NMVOCs in China from 2013 to 2020, with a horizontal resolution of 15 km × 15 km. This paper documents detailed descriptions of the assimilation system and the evaluation results for the emission inventory. The results suggest that CAQIEI can effectively reduce the biases in the *a priori* emission inventory, with the normalized mean biases ranging from −9.1% to 9.5% in the *a posteriori* simulation, which are significantly reduced from the biases in the *a priori* simulations (−45.6% to 93.8%). The calculated RMSE (0.3 mg/m$^3$ for CO and 9.4–21.1 μg/m$^3$ for other species, on the monthly scale) and correlation coefficients (0.76–0.94) were also improved from the *a priori* simulations, demonstrating good performance of the data assimilation system. Based on CAQIEI, we estimated China's total emissions (including both natural and anthropogenic emissions) of the 6 species in 2015 to be as follows: 25.2 Tg of $NO_x$, 17.8 Tg of $SO_2$, 465.4 Tg of CO, 15.0 Tg of $PM_{2.5}$, 40.1 Tg of $PM_{10}$, and 46.0 Tg of NMVOCs. From 2015 to 2020, the total emissions reduced by 54.1% for $SO_2$, 44.4% for $PM_{2.5}$, 33.6% for $PM_{10}$, 35.7% for CO, and 15.1% for $NO_x$, but increased by 21.0% for NMVOCs. It is also estimated that the emission reductions were larger during 2018–2020 (from -26.6% to -4.5%) than during 2015–2017 (from -23.8% to 27.6%) for most species. Particularly, the total Chinese $NO_x$ and NMVOC emissions were shown to increase during 2015–2017, especially over the Fenwei Plain area (FW) where the emissions of particulate matter (PM) also increased. The situation changed during 2018–2020 when the upward trends were contained and reversed to downward trends for both the total emissions of $NO_x$ and NMVOC, and the PM emissions over FW. This suggests that the emission control policies may be improved in 2018–2020 action plan. We also compared the CAQIEI with other air pollutant emission inventories in China, which verified our inversion results in terms of total emissions of $NO_x$, $SO_2$ and NMVOCs, and more importantly identified the potential uncertainties in current emission inventories. Firstly, the CAQIEI suggested higher CO emissions in China, with CO emissions estimated by CAQIEI (426.8 Tg) being more than twice the

amount in previous inventories (120.7–237.7 Tg). Significantly higher emissions were also suggested over the western and
northeastern China for other air pollutants. Secondly, the CAQIEI suggested higher NMVOC emissions than previous emission
inventories by about 30.4–81.4% over the North China Plain (NCP) but suggested lower NMVOC emissions by about 27.6–
0.0% over the Southeast China (SE). Thirdly, the CAQIEI suggested smaller emission reduction rates during 2015–2018 than
previous emission inventories for most species except of CO. Particularly, China's NMVOC emissions were shown to have
increased by 26.6% from 2015 to 2018, especially over the NCP (by 38.0%), northeast China (by 38.3%), and central China
(60.0%). These results provide us with new insight into the complex variations of the air pollutant emissions in China during
two recent clean air actions, which has the potential to improve our understanding of air pollutant emissions in China and their
impacts on air quality. The whole datasets are available at https://doi.org/10.57760/sciencedb.13151 (Kong et al., 2023).
**1 Introduction**
Air pollution is a serious environmental issue owing to its substantial impacts on human health, ecosystems, and climate
change (Von Schneidemesser et al., 2015; Cohen et al., 2017; Bobbink et al., 1998). According to the World Health
Organization, air pollution–induced strokes, lung cancer, and heart disease are causing millions of premature deaths worldwide
every year (WHO, 2016). The fine particulate matter ($PM_{2.5}$) in the atmosphere not only degrades visibility but also affects the
radiative forcing of the climate, both directly and indirectly (Martin et al., 2004). After removal from the atmosphere through
dry and wet deposition, air pollutants such as sulfur, nitrate, and ammonium contribute significantly to soil acidification,
eutrophication, and even biodiversity reduction (Krupa, 2003; Hernández et al., 2016).
China has experienced severe $PM_{2.5}$ pollution in recent decades, due to its large emissions of air pollutants associated
with rapid urbanization and high consumption of fossil fuels (Kan et al., 2012; Song et al., 2017). The annual concentrations
of $PM_{2.5}$ in 2013 reached 106, 67 and 47 $\mu g/m^3$ over the Beijing–Tianjin–Heibei, Yangtze River Delta, and Peral River Delta
region, respectively, which were all higher than China's national standard (35 $\mu g/m^3$), and 5–10 times higher than that of the
World Health Organization (10 $\mu g/m^3$). To tackle this problem, strict emission control policies (so-called "clean air action
plans") have been proposed by China's government, including the "Action Plan on the Prevention and Control of Air Pollution"
from 2013 to 2017 (hereinafter called the "2013–2017 action plan"), and the "Three-year Action Plan for Winning the Bule
Sky War" from 2018–2020 (hereinafter called the "2018–2020 action plan"). With the successful implementation of these two
action plans, the air quality was substantially improved in China, as evidenced in both observational and reanalysis datasets
(Li et al., 2020b; Zheng et al., 2017; Krotkov et al., 2016; Zhong et al., 2021; Li et al., 2017a; Kong et al., 2021). However,
with the deepening of air pollution control, unexpected changes have occurred in China, bringing about new challenges for the
mitigation of air pollution in the future. On the one hand, despite a significant decline in $PM_{2.5}$ concentrations in China, severe
haze still occasionally occurs during the wintertime (Zhou et al., 2022b; Li et al., 2017c). In addition, field measurements in
cities over different regions of China consistently show different responses of aerosol chemical compositions to the emission
control policies (Tang et al., 2021; Zhou et al., 2019; Wang et al., 2022; Zhang et al., 2020; Li et al., 2019a; Xu et al., 2019b;
Lei et al., 2021; Zhou et al., 2022a). Compared with other aerosol species that showed substantial decreases during the clean
air action plans, nitrate has shown a weaker response to the control measures, remaining at high levels and in some cases
having even increased slightly. As a result, nitrate is playing an increasingly important role in heavy haze episodes in winter,
and dominates the chemical composition of $PM_{2.5}$ (Fu et al., 2020; Xu et al., 2019a), leading to a rapid transition from sulphate-
to nitrate-driven aerosol pollution (Li et al., 2019a; Wang et al., 2019b). On the other hand, photochemical pollution has
deteriorated in China, with ozone ($O_3$) concentrations having increased substantially in eastern China during 2013–2017 (Li et
al., 2019b; Lu et al., 2018; Lu et al., 2020; Wang et al., 2020b).
These unexpected changes have raised considerable concern among the scientific community and policymakers regarding
the overall effects of the clean air action plans, and how to coordinate the control of $PM_{2.5}$ and $O_3$ pollution. Addressing this
problem requires a comprehensive understanding of the effects of the clean air action plans on the emissions of different air
pollutants. In this respect, previous studies have compiled several long-term air pollutant emission inventories in China using
the bottom-up approach – for example, the Multi-resolution Emission Inventory for China (MEIC) developed by Tsinghua
University for 2010–2020 (Zheng et al., 2018); the Air Benefit and Cost and Attainment Assessment System-Emission
Inventory version 2.0 (ABaCAS-EI v2.0) developed by Tsinghua University for 2005–2021 (Li et al., 2023); the Regional
Emission Inventory in Asia (REAS) for 1950–2015 developed by Kurokawa and Ohara (2020); the Emissions Database for
Global Atmospheric Research (EDGAR) for 1970–2018 developed by Jalkanen et al. (2012); the Hemispheric Transport of
Air Pollution (HTAP) Inventory for 2000–2018 developed by Crippa et al. (2023); and the Community Emissions Data System
(CEDS) Inventory for 1970–2019 developed by Mcduffie et al. (2020). These emission inventories have provided the
community with important insights into the long-term changes in the air pollutant emissions in China, thus playing an
indispensable role in our understanding of the effects of the country's clean air action plans on emissions and air quality.
However, due to the lack of accurate activity data and emission factors, bottom-up emission inventories are subject to large
uncertainties, particularly during the clean air action periods when the activity data and emission factors changed considerably
and were difficult to track. Consequently, the estimated emission rates from different bottom-up emission inventories could
differ by more than a factor of 2 (Elguindi et al., 2020). For example, the estimated emissions for the year 2010 from different
bottom-up inventories were 104.9–194.5 Tg for carbon monoxide (CO), 15.6–25.4 Tg for nitrogen oxides ($NO_x$), 22.9–27.0
Tg for non-methane volatile organic compounds (NMVOCs), 15.7–35.5 Tg for sulfur dioxide ($SO_2$), 1.28–2.34 Tg for black
carbon (BC), and 2.78–4.66 Tg for organic carbon (OC), reflecting the large uncertainty in current bottom-up estimates of air
pollutant emissions in China, which hinders the proper assessment of the effects of the clean air action plans.
Inverse modeling of multiple air pollutant emissions (i.e., a top-down approach) provides an attractive way to constrain
bottom-up emissions by reducing the discrepancy between the model and observation through the use of data assimilation.
Numerous studies have confirmed the effectiveness of such a top-down method in verifying bottom-up emission estimates and
reducing their uncertainties (e.g., Elbern et al., 2007; Henze et al., 2009; Miyazaki and Eskes, 2013; Tang et al., 2013; Koohkan
et al., 2013; Koukouli et al., 2018; Jiang et al., 2017; Muller et al., 2018; Paulot et al., 2014; Qu et al., 2017. Based on long-
term satellite observations, the top-down method has also been used to track the long-term variations of emissions. For example,
Zheng et al. (2019) estimated the global emissions of CO for the period 2000–2017 based on a multi-species atmospheric
Bayesian inversion approach; Qu et al. (2019) constrained global $SO_2$ emissions for the period 2005–2017 by assimilating
satellite retrievals of $SO_2$ columns using a hybrid 4DVar/mass balance emission inversion method; by assimilating satellite
observations of multiple species, Miyazaki et al. (2020b) simultaneously estimated global emissions of CO, $NO_x$, and $SO_2$ for
the period 2005–2018; and, most recently, a regional top-down estimation of $PM_{2.5}$ emissions in China during 2016–2020 was
carried out by Peng et al. (2023) by assimilating surface observations. These studies provide us with valuable clues for
evaluating bottom-up emissions and improving our knowledge on the changes in emissions of different species in China during
the clean air action plans. However, most of these studies focused on emission trends at the global scale, which involved the
use of coarse model resolutions (>1°) that may be insufficient to capture the spatial variability of emission variations at the
regional scale. Meanwhile, current long-term, top-down estimates mainly focus on single species and do not fully cover the
two clean air action periods in China. Indeed, to date, there are still no long-term, top-down estimates of major air pollutant
emissions in China that fully cover the two clean air action periods.
In a previous study performed by our group, we developed a high-resolution air quality reanalysis dataset over China
(CAQRA) for the period 2013–2020 to track the air quality trends in China during the clean air action periods (Kong et al.,
2021). In the present study, as a follow up to this work, we constrained the long-term emission trends of major air pollutants
in China for 2013–2020 (which will be extended in the future on a yearly basis) by assimilating surface observations of air
pollutants from the China National Environmental Monitoring Centre (CNEMC) using an ensemble Kalman filter and the
Nested Air Quality Prediction and Forecasting System (NAQPMS). In the following sections, we present detailed descriptions
of the chemical data assimilation, the evaluation results of the inversed emission inventory, and the estimated emission trends
of different air pollutants in China during the clean air action periods.

## 2 The chemical data assimilation system

We used the chemical data assimilation system (ChemDAS) developed by the Institute of Atmospheric Physics, Chinese
Academy of Sciences, to constrain the long-term emission changes of different air pollutants in China, which was used in the
development of CAQRA in our previous work (Kong et al., 2021). Since the chemical transport model (CTM) and the
observations used in the top-down estimation were the same as those used in CAQRA, we only briefly describe these two
components in the following two subsections, instead concentrating on providing a fuller description (in the third subsection)
of the inversion scheme in ChemDAS.

### 2.1 Chemical transport model

The NAQPMS model was used as the forecast model to represent the atmospheric chemistry in this study, and the Weather
Research and Forecasting (WRF) model was used as the meteorological model to provide the meteorological input data.
NAQPMS contains comprehensive modules for the emission, diffusion, transportation, deposition, and chemistry processes in
the atmosphere, and has been used in previous inversion studies (Tang et al., 2013; Kong et al., 2019; Wu et al., 2020a; Kong
et al., 2023). Detailed configurations of the different modules used in NAQPMS are available in these publications.
Figure 1 shows the domain of the inverse model, which is the same as that used in CAQRA, with a fine-scale horizontal
resolution of 15 km. The HTAPv2.2 emission inventory was used as the a priori estimate of anthropogenic emissions in China,
which includes emissions from the energy, industry, transport, residential, agriculture, air and ship sectors with a base year of
2010 (Janssens-Maenhout et al., 2015). It is a harmonized global emission inventory that comprises of different regional
gridded inventories. Within the region of China, the air pollutant emissions were mainly provided by the MEIC emission
inventory (Janssens-Maenhout et al., 2015). The *a priori* estimates of emissions from other sources includes the biogenic
emissions obtained from the Monitoring Atmospheric Composition and Climate (MACC) project (Sindelarova et al., 2014);
biomass burning emissions obtained from the Global Fire Emissions Database (GFED), version 4 (Van Der Werf et al., 2010;
Randerson et al., 2017); soil and lightning $NO_x$ emissions obtained from Yan et al. (2003) and Price et al. (1997); and marine
volatile organic compound emissions obtained from the POET database (Granier et al., 2005). The dust emissions were
calculated online in NAQPMS as a function of the relative humidity, frictional velocity, mineral particle size distribution, and
the surface roughness (Li et al., 2012), while the sea salt emissions were calculated using the scheme of Athanasopoulou et al.
(2008). Note that since we aimed to estimate the air pollutant emissions and their changes from the surface observation, we
did not consider the temporal variation in the *a priori* emission inventory. This would ensure that the top-down estimated
emission trends were only derived from the surface observations, without being influenced by the trends in the prior emission
inventory. In this way, our top-down estimation can serve as an independent estimation of the air pollutant emission changes
in China. Meanwhile, we used the constant diurnal variation of the emissions in this study due to the lack of information on
the diurnal variation of the emissions from different sectors, which is a potential limitation in our current work. However, since
the emission inversion was performed on the daily basis (Sect. 2.3.3), the diurnal variations of the emission may not
significantly influence the simulation results of the daily mean concentrations of air pollutants (less than 1 ppbv for $SO_2$, $NO_2$
and $O_3$) according to the sensitivity experiments conducted by Wang et al. (2010). The initial condition was treated as clean
air in NAQPMS, with a 2-week spin-up time. Top and boundary conditions were provided by the Model for Ozone and Related
Chemical Tracers (MOZART) (Brasseur et al., 1998; Hauglustaine et al., 1998) data products provided by National Center for
Atmospheric Research (NCAR). Note that since the MOZART data products were not available for years after 2018, the multi-
year average results from 2013 to 2017 were used for the simulations after 2018. Because most of the model boundaries were
set in the clean areas and are located at distance from China, we assumed that the differences in boundary conditions would
not significantly affect the modeling results over the China. To improve the performance of meteorological simulation, a 36-
h free run of the WRF model was conducted for each day by using the NCAR/NCEP 1°×1° reanalysis data. The simulation
results of the first 12 h were treated as the spin-up run, and the remaining 24 h were used to provide the meteorological inputs
for the NAQPMS model. The evaluation results for the WRF simulation are available in Text S1 in the Supplement, which
suggests acceptable performance of the WRF simulation for the inversion estimates (Table S1).
**2.2 Assimilated observations**
The assimilated observational dataset in this study was the same as that used in CAQRA, which includes surface
concentrations of $PM_{2.5}$, $PM_{10}$ (coarse particulate matter), $SO_2$, $NO_2$ (nitrogen dioxide), CO, and $O_3$, from 2013 to 2020,
obtained from CNEMC (Fig. 1). Before the assimilation, outliers of the observations were filtered out by using an automatic
quality control method developed by Wu et al. (2018). Four types of outliers characterized by temporal and spatial
inconsistencies, instrument-induced low variances, periodic calibration exceptions, and lower $PM_{10}$ concentrations than those
of $PM_{2.5}$, were filtered out to prevent adverse impacts on the inversion process. As estimated in Kong et al. (2021), about 1.5%
of observational data were filtered out after quality control, but further assessment showed that it had few effects on the average
concentrations of different species, which were estimated to be less than 1 $\mu g/m^3$ for the gaseous air pollutants and less than
5 $\mu g/m^3$ for the particulate matter. Estimation of observation error is also important for the inversion of emissions since the
observational error and background errors determine the degree of adjustment to the emissions. The observational error
comprises the measurement error and the representativeness error induced by the different spatial scales that the model and
observations represent. The estimations of these two components of observational error were the same as those used in CAQRA,
detailed descriptions of which are available in Kong et al. (2021).
It should be noted that the number of observation sites were not constant throughout the whole inversion period, being
approximately 510 in 2013 and then increasing to 1436 in 2015. According to Fig. S1, the observation sites were mainly
concentrated in the megacity clusters (e.g., North China Plain, Yangtze River Delta and Pearl River Delta) and the capital
cities of each province in 2013. The number of observation sites continued to increase across the China in 2014 and 2015. In
particular, many areas that were previously unobserved have added monitoring stations in 2014 and 2015, which significantly
increased the observation coverage of China and could lead to spurious trends in the top-down estimated emissions. Figure 2
shows the changes in the observational coverage over different regions of China from 2013 to 2020 indicated by the ratio of
areas that were influenced by observations to the total area of each region. It can be clearly seen that the observational coverage
increased from 2013 to 2015 with the expansion of the air quality monitoring network in China, and became stable after 2015.
However, the influence of the variation in the number of observation sites varied among different regions. Over the North
China Plain (NCP) region, the observational coverage was approximately 90% in 2013, and reached 100% in 2014, suggesting
that the variation in the observation sites may have little influence on the estimated emission changes there. A similar
conclusion can be drawn for the Southeast China (SE) region, where the observational coverage was about 75% in 2013 and
reached 100% in 2015. Elsewhere, in the other four regions, the influence of the variation in observation sites is expected to
be larger because of the low observational coverage in both 2013 and 2014. For example, the observational coverage over the
Northwest China (NW) region was less than 10% in 2013, but increased to about 60% in 2015. To better illustrate the impact
of changes in observational coverage on the inversions, a sensitivity analysis of the emission increments with the fixed
observation sites or varying observation sites is performed in this study (Text S2 and Fig. S2). It shows that the additional
emission increments caused by the increases of observation sites would weaken the decreasing trends estimated in the fixed-
site scenario for the emissions of $PM_{2.5}$, $NO_x$ and NMVOC and even lead to increasing trends for the emissions of $PM_{10}$ and
CO. In contrast, the increases of observation sites would enhance the decreasing trends of $SO_2$ estimated in the fixed-site
scenario. Such different behaviors are mainly related to the different sign of the emission increment of different species as we

illustrated in Text S2. These results highlighted the significant influences of the site differences on the estimated emissions and their trends, which should be noted by the potential users. Therefore, in order to reduce this influence on the estimated emission trends, in our following analysis we mainly analyze the emission trends after 2015, when the observational coverage had stabilized in all regions.

**2.3 Data assimilation algorithm**

We used the modified EnKF coupled with state augmentation method to constrain the long-term emissions of different air pollutants. EnKF is an advanced data assimilation method proposed by Evensen (1994) that features representing the background error covariance matrix with a stochastic ensemble of model realizations. Through the use of ensemble simulations, it has the ability to consider the indirect relationship between the emissions and chemical concentrations caused by the complex physical and chemical processes in the atmosphere. It also allows for the estimation of flow-dependent emission–concentration relationships that vary in time and space depending on the atmospheric conditions. The modified EnKF is an offline application of the EnKF method that works by decoupling the analysis step from the ensemble simulation, which has benefits in the reuse of costly ensemble simulations and makes high-resolution long-term inversion affordable (Wu et al., 2020a). In this method, the ensemble simulation was performed firstly with the perturbed emissions, and then the observations were assimilated to constrain the emissions (Wu et al., 2020a). The state augmentation method is a commonly used parameter estimation method (Tandeo et al., 2020) in which the air pollutant emissions are taken as the state variable and are updated according to the error covariance between the emissions and the concentrations of related species.

**2.3.1 State variable and ensemble generations**

The state variable used in this study was chosen following our previous multi-species inversion study (Kong et al., 2023), which included the scaling factors for the emissions of fine-mode unspeciated aerosol (PMF), coarse-mode unspeciated aerosol (PMC), BC, OC, $NO_x$, $SO_2$, CO, and NMVOC, as well as the chemical concentrations of $PM_{2.5}$, $PM_{10-2.5}$ ($PM_{10}$ minus $PM_{2.5}$), $NO_2$, $SO_2$, CO, and daily maximum 8-h $O_3$ (MDA8h $O_3$), which are formulated as follows:

$$x = [c,\ \beta]^T, \tag{1}$$

$$c = [PM_{2.5}, PM_{10-2.5},\ NO_2,\ SO_2, CO,\ MDA8h\ O_3], \tag{2}$$

$$\beta = [\beta_{PMF},\ \beta_{PMC},\ \beta_{BC},\ \beta_{OC}, \beta_{NO_x}, \beta_{SO_2}, \beta_{CO}, \beta_{NMVOC}], \tag{3}$$

where $x$ denotes the vector of the state variable, $c$ denotes the vector of the chemical concentrations of different species, and $\beta$ denotes the vector of the scaling factors for the emissions of different species. Note that although the chemical concentration variables are included in the state variable, they are not optimized simultaneously with the emission in the analysis step and are only used to estimate the covariance between the emission and concentrations. Detailed descriptions of the state variables are available in Table 1.

The ensemble of the scaling factors for different species was generated independently using the same method of Kong et al. (2021), which has a medium size of 50 and considers the uncertainties of major air pollutant emissions in China, including $SO_2$, $NO_x$, CO, NMVOCs, ammonia, $PM_{10}$, $PM_{2.5}$, BC, and OC. The uncertainties of these species were considered to be 12%, 31%, 70%, 68%, 53%, 132%, 130%, 208% and 258%, respectively according to the estimates of Li et al. (2017b) and Streets et al. (2003). Note that in this study we did not perturb the emissions of different sectors to reduce the degrees of freedom in the ill-posed inverse estimation problem. Instead, we only perturbed the total emissions of different species. Therefore, only the total emissions of different species were constrained in this study. The ensemble of the chemical concentrations was then generated through an ensemble simulation based on NAQPMS and the perturbed emissions calculated by multiplying the *a priori* emissions by the ensemble of the scaling factors. This treatment implicitly assumes that the uncertainty in the chemical concentration is mainly caused by the emission uncertainty. This makes sense on a monthly or yearly basis, considering that substantial changes in emissions are expected to have taken place during the clean air action plans, which are subject to large

uncertainty. However, the lack of consideration of other error sources, such as those of the meteorological simulation and the
model itself, may lead to underestimation of the background error covariance and emission adjustment, which is a potential
limitation of this study. In addition, the dust and sea salt emissions were not perturbed and constrained in this study, and thus
the errors in the simulated fine and coarse dust emissions would influence the inversion of $PM_{2.5}$ and $PM_{10}$ emissions. As a
result, the top-down estimated $PM_{2.5}$ and $PM_{10}$ emissions will contain errors in the simulated dust and sea salt emissions.
Particularly, we did not consider the emissions of coarse dust during the inversion process since there is large uncertainty in
the simulated coarse dust emissions by current dust emission schemes (Zeng et al., 2020; Kang et al., 2011). The large errors
in the simulated coarse dust concentration could significantly influence the inversion results of $PM_{10}$ emissions. For example,
the simulated coarse dust concentration could sometimes be several orders of magnitude higher than the observed $PM_{10}$
concentration, leading to too low values of the inverse $PM_{10}$ emissions (approximately 0) over the regions that were not the
typical dust source regions but were influenced by the transportation of coarse dust. Therefore, we only used simulated $PM_{10}$
concentrations from other sources in the inversion of $PM_{10}$ emissions to avoid the influences of the too large errors in simulated.
This is also similar to assume that the coarse dust emission is equal to zero during the assimilation. However, in this way, the
top-down estimated $PM_{10}$ emissions in this study would comprise all coarse dust emissions which should be noted by potential
users. A detailed description of the ensemble generation is available in Kong et al. (2021).

### 2.3.2 Inversion algorithm

We used a deterministic form of EnKF (DEnKF) proposed by Sakov and Oke (2008) to update the scaling factors of the
emissions of different species, which is formulated as follows:

$$\overline{x^a} = \overline{x^b} + \mathbf{K}\left(y^o - \mathbf{H}\overline{x^b}\right), \tag{4}$$

$$\mathbf{X}^a = \mathbf{X}^b - \frac{1}{2}\mathbf{K}\mathbf{H}\mathbf{X}^b \tag{5}$$

$$\mathbf{K} = \lambda \mathbf{B}_e^b \mathbf{H}^T \left(\mathbf{H}\lambda \mathbf{B}_e^b \mathbf{H}^T + \mathbf{R}\right)^{-1}, \tag{6}$$

$$\mathbf{B}_e^b = \frac{1}{N-1}\sum_{i=1}^{N} X_i^b \left(X_i^b\right)^T, \tag{7}$$

$$\overline{x^b} = \frac{1}{N}\sum_{i=1}^{N} x_i^b \,;\, X_i^b = x_i^b - \overline{x^b}, \tag{8}$$

where $\overline{x}$ denotes the ensemble mean of the state variable; the superscript **b** and **a** respectively denote the *a priori* and *a*
*posteriori* estimate; $\mathbf{X}^a$ is the analysed anomalies that can be used to calculate the uncertainty of the a posteriori emissions. $\mathbf{K}$
is the Kalman gain matrix; $\mathbf{B}_e^b$ is the background error covariance matrix calculated by the background perturbation $X^b$; $y^o$ is
the vector of the observation and $\mathbf{R}$ is the observation error covariance matrix; $\mathbf{H}$ is the linear observation operator, which
maps the model space to the observation space; $\lambda$ is the inflation factor used to compensate for the underestimation of the
background error caused by the limited ensemble size and unaccounted error sources, which is calculated using the method of
Wang and Bishop (2003),

$$\lambda = \frac{\left(\mathbf{R}^{-1/2}d\right)^T \mathbf{R}^{-1/2}d - p}{trace\left\{\mathbf{R}^{-1/2}\mathbf{H}\mathbf{B}_e^b\left(\mathbf{R}^{-1/2}\mathbf{H}\right)^T\right\}} \tag{9}$$

$$d = y^o - \mathbf{H}\overline{x^b} \tag{10}$$

where $d$ is the observation innovation and $p$ is the number of observations. Table S2 summarized the calculated average value
(standard deviation) of the used inflation factor for different species. It shows that the inflation factor over the east China
(including NCP and SE region) was generally round 1.0, suggesting that the original ensemble can well represent the simulation
errors of the different air pollutants over these regions. The inflation factor is larger over the western China (including SW,
NW and Central regions), especially for $PM_{10}$ (36.0–78.1) and $SO_2$ (7.8–176.1), suggesting that the original ensemble may
underestimate the simulation errors of the air pollutants. This is associated with the large biases in the simulated air pollutant
concentrations over there and reflect that the emission uncertainties assumed in our studies may be underestimated over these
regions. This also highlighted the importance of the use of inflation method during the inversion, otherwise it would lead to
filter divergency caused by the underestimations of the background error covariance.

In order to reduce the influence of the spurious correlations on the performance of data assimilation, the EnKF was

performed locally in this study in that the analysis was calculated grid by grid with the assumption that only measurements
located within a certain distance (cutoff radius) from a grid point would influence the analysis results of this grid. The use of
this local analysis method also allowed the inflation factor to be calculated locally and to vary in time and space, which can
help characterize the spatiotemporal variations of errors as we illustrated above. Similar to in Kong et al. (2021) and Kong et
al. (2023), the cutoff radius was chosen as 180 km for each species based on the wind speed and the lifespan of the species
(Feng et al., 2020). The same local scheme with a buffer area was also employed during the inversion to alleviate the
discontinuities in the updated state caused by the cut-off radius. A detailed description of the local analysis scheme is available
in Kong et al. (2021).

Table 1 summarizes the corresponding relationships between the emissions and chemical concentrations. Similar to Ma

et al. (2019) and Miyazaki et al. (2012), we did not consider the inter-species correlation during the assimilation to prevent the
spurious correlations between non- or weakly related variables. In most cases, observations of one particular species were only
allowed to adjust the emissions of the same species. The assimilation of $PM_{2.5}$ mass observation was more complicated as
there are multiple error sources in the simulated mass concentrations of $PM_{2.5}$, not only from primary emission, but also from
secondary production. In this study, the $PM_{2.5}$ mass observation was used to constrain the emissions of PMF, BC and OC but
not used to constrain the emissions of its precursors to avoid the spurious correlations and nonlinear chemistry effects, which
is similar to the scheme used in Ma et al. (2019). This is feasible since the emissions of primary $PM_{2.5}$ (i.e., PMF, BC and OC)
and the emissions of $PM_{2.5}$ precursors (e.g., $SO_2$, $NO_2$) were perturbed independently in our method, thus the contributions of
primary $PM_{2.5}$ emission and the secondary $PM_{2.5}$ productions to the $PM_{2.5}$ mass could be isolated through the use of ensemble
simulations. Meanwhile, the use of iteration inversion method (which will be introduced later) can further reduce the influence
of the errors in the precursors' emissions on the inversion of primary $PM_{2.5}$ emission, because the errors of its precursors'
emission would be constrained by their own observations during the iterations. However, the lack of assimilation of speciated
$PM_{2.5}$ observations may lead to uncertainties in the estimated emissions of PMF, BC and OC, which is a potential limitation
in current work. For example, if the a priori simulated $PM_{2.5}$ equals the observations, the emissions of PMF, BC and OC would
not be adjusted by using the current method. However, in such cases, there may still be errors in the proportions of the emissions
of different $PM_{2.5}$ components. To adjust the emissions of PMC, we used the observations of $PM_{10-2.5}$ to avoid the potential
cross-correlations between $PM_{2.5}$ and $PM_{10}$ (Peng et al., 2018; Ma et al., 2019). For the $NO_x$ emissions, although the $O_3$
concentration are chemically related to the $NO_x$ emissions, we did not use the $O_3$ concentrations to constrain the $NO_x$ emission
in this study since there is nonlinear relationship between the $O_3$ concentration and $NO_x$ emission which would lead to wrong
adjustment of $NO_x$ emissions (Tang et al., 2016).

The inversion of NMVOC emission is more difficult than other species due to the lack of long-term nationwide NMVOC

observations and the strong chemical activity. Previous studies usually assimilated the satellite observations of formaldehyde
and glyoxal to constrain the NMVOC emissions, such as Cao et al. (2018) and Stavrakou et al. (2015). However, these
inversion studies are hindered by the $NO_x$-VOC-$O_3$ chemistry and the inherent uncertainty in the satellite observations of
formaldehyde and glyoxal. Considering the strong chemical relationship between the $O_3$ and NMVOC, some pioneer studies
have also explored the method of assimilating ground-level $O_3$ concentrations to constrain the NMVOC emissions (Ma et al.,
2019; Xing et al., 2020), and demonstrated the effectiveness of this approach. For example, Ma et al. (2019) found that the
assimilation of $O_3$ concentration could adjust the NMVOC emissions in the direction resembling the bottom-up inventories,
and the forecast skill of $O_3$ concentrations were also improved, indicating that the constrained NMVOC emissions are improved
relative to their a priori. Inspired by these studies, we have made an attempt to constrain the NMVOC emissions based on the
MDA8h $O_3$. The use of MDA8h $O_3$ rather than the daily mean $O_3$ concentration is to avoid the effects of the nighttime $O_3$
chemistry. For example, the simulation errors in the titration effects of $NO_x$ may influence the simulated $O_3$ concentrations
during nighttime and affect the inversion results of NMVOC. An important issue that should be noted when using the MDA8h
$O_3$ to constrain the NMVOC emission is the nonlinear interactions among $NO_x$, NMVOC and $O_3$. On the one hand, the $O_3$
concentrations are dependent not only on the NMVOC emissions but also on the $NO_x$ emissions. The errors in the a priori
emissions of $NO_x$ would also contribute to the simulation errors of $O_3$, and deteriorate the inversion of NMVOC. The iteration
inversion scheme could help deal with this issue as the errors in the $NO_x$ emissions will be constrained by the $NO_2$ observations
in the next iteration, which can reduce the influences of errors in the $NO_x$ emission on the inversion of NMVOC emission
based on MDA8h $O_3$ concentrations. This is in fact similar to the approach used by Xing et al. (2020) who firstly constrained
the $NO_x$ emissions based on observations of $NO_2$, and then constrained the NMVOC emissions based on $O_3$ concentrations.
Also, in Feng et al. (2024), the $NO_2$ observations were simultaneously assimilated to constrain the $NO_x$ emissions to account
for the influences of errors in $NO_x$ emissions on the NMVOC emissions, suggesting that the iteratively nonlinear joint inversion
of $NOx$ and NMVOCs is an effective way to address the intricate relationship among VOC-$NOx$-$O_3$ (Feng et al., 2024).
Similarly, the errors in the CO emissions which may be significant according to our following analysis are also constrained in
a similar way to reduce the potential influences on the inversion of NMVOC emission. On the other hand, the emission
adjustments of NMVOC may exhibit bidirectionality dependent on the VOC-limited or $NO_x$-limited regimes. According to
Fig. 3, the NMVOC emissions were adjusted in alignment with the direction of the $O_3$ errors, suggesting a VOC-limited regime
over urban areas in China, given that the $O_3$ observation sites are predominantly situated in the urban areas. This agrees with
Ren et al. (2022) who diagnosed the $NO_x$-VOC-$O_3$ sensitivity based on the satellite retrievals and found that the VOC-limited
regimes are mainly located in the urban areas in China. This suggests that the relationship between the $O_3$ concentrations and
VOC emissions could be reasonably reflected by our inversion system, providing the feasibility in utilizing the $O_3$ observations
to constrain the VOC emissions. Note that due to the lack observations of the VOC components, we only optimize the gross
emissions of the VOC during the assimilation.
As we illustrated before, there exists nonlinear effects in the atmospheric chemistry which could influence the inversion
results of different species. In addition, since we did not consider the temporal variations in the *a priori* emissions, it was
expected that there would be significant biases in the *a priori* emissions for the years after 2013, as substantial changes in
emissions were expected owing to the implementation of strict emission control measures. Such bias in the *a priori* emissions
does not conform to the unbiased hypothesis of the EnKF, which could lead to incomplete adjustments of the *a priori* emissions
and degrade the performance of the data assimilation (Dee and Da Silva, 1998). To address these issues, an iteration inversion
scheme was employed in this study, which has been used previously in Kong et al. (2023). The main idea of the iteration
inversion scheme is to preserve the background perturbation $\mathbf{X^b}$ but to update the ensemble mean of the state variable $\overline{x^b}$ based
on the model simulations driven by the inversion results of the $k$th iteration. Therefore, a new single model simulation is
required to be conducted by using the a posteriori emission from the previous iteration as the input to update the ensemble
mean of the original ensemble. This enables the observational information and the adjusted emissions to be promptly
incorporated into the model, thereby providing feedback for the adjustments of emission in the next iteration. However, we
did not reassemble the ensemble simulation for each iteration due to the expensive computational cost of the ensemble
simulation. Therefore, in each iteration calculation, the ensemble perturbation that were used to calculate the background error
covariance matrix remains the same with only the ensemble mean being updated based on the inversion results of the previous
iteration. The state variable used in the $(k + 1)$th inversions is then formulated as follows:
$x_i^{b,k+1} = \left[ c^k + c_i^e - \bar{c^e}, \beta^k + \beta_i^e - \overline{\beta^e} \right]^T,$                    (11)
where $c^k$ represents the model simulations driven by the inversed emissions of the $k$th iteration, $c_i^e$ represents the $i$th member
of ensemble simulations with an ensemble mean of $\bar{c^e}$, $\beta^k$ represents the updated scaling factors at the $k$th iteration, and
$\beta_i^e$ represents the $i$th member of the ensemble of scaling factors with a mean value of $\overline{\beta^e}$. In each iteration, all emissions are
updated simultaneously and two rounds of iteration were conducted in this study based on our previous inversion study to
maintain a balance between the inversion performance and the computational cost of the long-term inversions (Kong et al.,
2023).

**2.3.3 Setup of inversion estimation**

Based on this inversion scheme, we constrained the daily emissions of PMF, PMC, BC, OC, $NO_x$, $SO_2$, CO, and NMVOCs,
from 2013 to 2020, based on the daily averaged observations of $PM_{2.5}$, $PM_{10-2.5}$, $NO_2$, CO, and MDA8h $O_3$. However, due the
lack of enough speciated $PM_{2.5}$ observation, the model performance driven by the inverse emission for the BC, OC and primary
unspeciated $PM_{2.5}$ have not been thoroughly evaluated. It is thus currently unclear for the quality of the inverse emissions of
BC, OC and primary unspeciated $PM_{2.5}$. Also, the lack of speciated $PM_{2.5}$ observations could lead to uncertainties in the
estimated emissions of PMF, BC, and OC as we mentioned before. Considering this, similar to in Kong et al. (2023), although
we made attempt to estimate the emissions of BC, OC and primary unspeciated $PM_{2.5}$, we have reservations about their
inversion results and only provide the emissions of $PM_{2.5}$ (PMC+BC+OC) and PM10 ($PM_{2.5}$ + PMC) in current stage. In future,
we will collect more speciated $PM_{2.5}$ observations to comprehensively quantify the accuracy of their inversion results, after
which the emissions of these species would be released. Meanwhile, the speciated $PM_{2.5}$ observations could be assimilated
under the current inversion framework. This could provide us with further constrains on the emissions of BC, OC and primary
$PM_{2.5}$. Meanwhile, as mentioned in subsection 2.3.1, the meteorological and model uncertainty were not considered in the
ensemble simulation. Thus, the errors in the meteorological simulation would cause fluctuations in the daily emissions that
contaminate the inversion results and are difficult to isolate from the inherent variations of emissions (Tang et al., 2013).
Considering this, the daily emissions were averaged to monthly values to reduce the influences of random model errors after
the assimilation.

**3 Performance of the chemical data assimilation system**

**3.1 Analysis of OmF and emission increment**

The observation-minus-forecast (OmF) and emission increment (*a posteriori* emission minus *a priori* emission) were
firstly analyzed to demonstrate the performance of the data assimilation. As shown in Fig. 3, the *a priori* simulation generally
underestimated the $PM_{2.5}$ concentrations over the NCP, SE and SW regions (positive OmF values) during 2013–2014, but
overestimated the $PM_{2.5}$ concentrations from 2016, reflecting the effects of the emission control measures during these years.
In the NE, NW and central China (hereafter, "Central") regions, obvious underestimation of the $PM_{2.5}$ concentration was found
(positive OmF values) throughout almost the entire assimilation period. Similarly, the OmF values of $PM_{10}$ were positive
throughout the whole assimilation period over all regions of China. In contrast, the OmF values for $SO_2$ were negative for most
regions, and the negative OmF values over the NCP region became larger as the years progressed, which reflects the effects
of the emission control measures. The OmF for $NO_2$ reveals a seasonal variation over the NCP and SE regions, with negative
values during summer and positive values during winter, while there were obvious positive OmF values over the NE, SW, NW
and Central regions. In terms of CO, large positive OmF values were found over all regions of China, and there were decreasing
trends in the OmF values of CO over different regions of China associated with the emission control policies during these
years. The OmF values for $O_3$ were positive over most regions of China, except the NW region. These results provide us with
valuable information on the potential deficiencies in the *a priori* emissions. However, since our inversion method did not
differentiate between anthropogenic and natural emissions, the biases in the model simulation may also be attributable to the
errors in natural emissions such as dust, especially over the major dust-source areas of China (e.g., the NW and Central regions).
In addition, the effects of emission control were not considered in the *a priori* emissions, which is another important contributor
to the errors in the model simulation for the later years. Thus, the emission increments calculated by the assimilation should
reflect the combined effects of errors in the anthropogenic and natural emissions, as well as the emission control.

The calculated emission increments were consistent with the OmF values for all species, which indicates that the data assimilation method can probably constrain the emissions based on the observations. According to Fig. 3, the emission increments were positive for $PM_{2.5}$ over the NE, NW and Central regions, for $NO_2$ over the NE, SW, NW and Central regions, and for $PM_{10}$, CO and NMVOC over almost all regions throughout the assimilation period. In contrast, the emission increments were negative for the $SO_2$ emissions for most cases. Consistent with the OmF values, the emission increments were positive for $PM_{2.5}$ over the NCP, SE and SW regions during 2013–2014, but became negative from 2016 owing to the implementation of strict emission control measures. The emission increments for $NO_x$ also showed significant seasonal variation over the NCP and SE regions, being positive during winter and negative during summer. The *a posteriori* biases for the model simulations of different species were also plotted to assess the performance of the data assimilation. It can be clearly seen that the biases were substantially reduced for all species, and the calculated root-mean-square error (RMSE) reduced by 23.2–52.8% for $PM_{2.5}$, 19.9–37.8% for $PM_{10}$, 36.4–77.3% for $SO_2$, 18.3–25.2% for $NO_2$, 29.9–40.5% for CO, and 4.4–26.1% for $O_3$ over the different regions of China, suggesting a good performance of the data assimilation system.

**3.2 Evaluation of the inversion results**

Table 2 shows the calculated evaluation statistics for the inversion at different temporal scales. It can be clearly seen that the model simulation with the *a posteriori* emission inventory reproduced well the magnitude and temporal variations of the different air pollutants in China, with calculated correlation coefficients of approximately 0.77, 0.72, 0.64, 0.67, 0.69 and 0.71, and normalized mean biases of approximately 4.5%, −4.6%, −9.0%, −3.9%, −8.8% and 9.5%, for the hourly concentrations of $PM_{2.5}$, $PM_{10}$, $SO_2$, $NO_2$, CO and $O_3$, respectively. Moreover, the *a posteriori* model simulation achieved comparable accuracy with the air quality reanalysis data we developed in Kong et al. (2021) in terms of the RMSE, which was 32.4 $\mu g \cdot m^{-3}$, 53.1 $\mu g \cdot m^{-3}$, 24.9 $\mu g \cdot m^{-3}$, 19.9 $\mu g \cdot m^{-3}$, 0.56 $mg \cdot m^{-3}$ and 34.9 $\mu g \cdot m^{-3}$, respectively, for these species at the hourly scale. At the daily, monthly and yearly scales, the constrained model simulation performed better, with RMSEs of about 9.1–20.0 $\mu g \cdot m^{-3}$ ($PM_{2.5}$), 18.5–31.6 $\mu g \cdot m^{-3}$ ($PM_{10}$), 11.5–16.0 $\mu g \cdot m^{-3}$ ($SO_2$), 8.1–12.8 $\mu g \cdot m^{-3}$ ($NO_2$), 0.28–0.39 $mg \cdot m^{-3}$ (CO), and 14.2–26.1 $\mu g \cdot m^{-3}$ ($O_3$), which were respectively reduced by 56.7–67.3%, 49.2–52.1%, 68.8–72.8%, 36.3–39.8%, 47.0–58.0%, and 22.9–30.5% compared to the RMSEs of the *a priori* simulations. We also compared the model performance driven by the inverse inventory with that driven by more recent bottom-up inventories (MEIC and HTAPv3) by taking the simulation results of year 2020 as an example to give us a more objective understanding of the accuracy of the inverse emission inventory. It shows that the inverse emission generally achieves better performance in simulating the air pollutant concentrations in China than the MEIC and HTAPv3 (Table S3). It is also encouraging to find that the model performance driven by CAQIEI and MEIC-HTAPv3 is similar for $PM_{2.5}$, $PM_{10}$, and $SO_2$ over the NCP, NE, SE and SW regions, both significantly improved from the a priori emission inventory. This suggest that both the top-down and recent bottom-up emission inventories have good performance in capturing the emission changes of these species over these regions and they yield consistent estimations. Detailed information on the configurations of the model simulation results driven by MEIC-HTAPv3 and the comparisons results are available in Text S3. All these validation results confirm the good performance of the data assimilation method and suggest that the inversed emissions inventory has the capability to reasonably represent the magnitude and long-term trends of the air pollutant emissions in China during 2013–2020.

**4 Results**

Based on the top-down estimation, the gridded emissions for $PM_{2.5}$, $PM_{10}$, $SO_2$, CO, $NO_x$ and NMVOCs over China from 2013 to 2020 were developed into what we have called the Inversed Emissions Inventory for Chinese Air Quality (CAQIEI). In the following sections, we first analyze the magnitude and seasonality of the air pollutant emissions in China by taking 2015 as a reference year when the number of observation sites became stable. After that, the changes in emissions of different air

pollutants from 2015 to 2020 are analyzed and compared between the two clean air action plans in China. Note that due to the impacts of the changes in observation coverage, it is difficult to estimate the overall emission reduction rates during the 2013–2017 action plan by using our inversion results. The emission change rates during 2015–2017 were then sampled in this study to assess the mitigation effects during the 2013–2017 action plan and to be compared with the emission change rates during 2018–2020. Finally, CAQIEI is compared to the previous bottom-up and top-down emission inventories to validate our top-down estimation and identify the potential uncertainties in the current understanding of China's air pollutant emissions.

**4.1 Top-down estimated Chinese air pollutant emissions in 2015**

The top-down estimated emissions of different species in 2015 are as follows: 25.2 Tg of $NO_x$, 17.8 Tg of $SO_2$, 465.4 Tg of CO, 15.0 Tg of $PM_{2.5}$, 40.1 Tg of $PM_{10}$, and 46.0 Tg of NMVOCs. Note that these values not only contain anthropogenic emissions but also natural (e.g., dust and biogenic NMVOC) emissions. Thus, the top-down estimated emissions of PM and NMVOCs were higher than those estimated by previous studies, as we mention in following sections. Emission maps of all species in 2015 are shown in Fig. 4, and the calculated emissions of different species over different regions are presented in Table 3. According to Fig. 4, higher air pollutant emissions are widely distributed in the megacity clusters (e.g., NCP, Yangtze River Delta and Pearl River Delta) and the developed cities in China, reflecting the influences of human activities. NCP was the region with the largest emission intensity of air pollutants in China, contributing 5.1 Tg of $NO_x$, 3.5 Tg of $SO_2$, 82.2 Tg of CO, 2.7 Tg of $PM_{2.5}$, 8.7 Tg of $PM_{10}$ and 9.0 Tg of NMVOCs to the total emissions in China. The inversion results also demonstrate the contribution of natural sources to the air pollutant emissions, such as the soil $NO_x$ emissions and the biogenic NMVOC emission distributed in the Tibet Plateau region. In general, the majority of air pollutant emissions were located in eastern China (including the NCP, NE and SE regions), where the economy is relatively well developed, which in total accounted for 66.0% of $NO_x$, 60.9% of $SO_2$, 57.5% of CO, 60.4% of $PM_{2.5}$, 60.5% of $PM_{10}$, and 67.8% of NMVOC emissions in China. However, although the GDP of western China (including the SW, NW and Central regions) is less than one third that of eastern China, the top-down estimation indicates that the air pollutant emissions in western China could have accounted for about 32.2–42.5% of the total emissions, which reflects the low emission control levels over these regions.

Figure 5 shows the monthly variations of air pollutant emissions in China for year 2015. The monthly profile of $NO_x$ emissions was relatively flat among the six species. $SO_2$ and CO showed higher emissions during wintertime because of the enhanced residential emissions associated with higher coal consumption for heating during that time of year. Meanwhile, the emission factor for CO from vehicles in winter was also higher than in other seasons, due to additional emissions from the cold-start process (Kurokawa et al., 2013; Li et al., 2017b). $PM_{2.5}$ and $PM_{10}$ had higher emissions during winter and spring, which, on the one hand was due to the enhanced emissions from the residential and industrial sectors during wintertime (Li et al., 2017b), whilst on the other hand was due to the enhanced dust emissions during the spring season (Fan et al., 2021). Emissions of NMVOCs exhibited strong monthly variations, with higher emissions mainly in summer because of the enhanced NMVOC emissions from biogenic sources.

**4.2 Top-down estimated emission changes of different air pollutants**

**4.2.1 Emission changes of particular matter**

Figure 6 shows the top-down estimated emission changes of $PM_{2.5}$ and $PM_{10}$ over China during two clean air action periods. Both $PM_{2.5}$ and $PM_{10}$ emissions decreased substantially, by 44.3% and 21.2% respectively, from 2013 to 2020. On the contrary, the top-down estimates showed increases of $PM_{2.5}$ and $PM_{10}$ emissions in 2014 and 2015, but this would be a spurious trend caused by the changes of observation sites as we discussed in Text S2. Therefore, the emissions in 2013 and 2014 were discarded to prevent the spurious trends. According to Fig. 6, the $PM_{2.5}$ emissions decreased by 14.5% from 2015

(15.0 Tg) to 2017 (12.8 Tg), and the reduction in emissions was roughly uniform throughout the period, which was about 8%
compared to previous years. The $PM_{10}$ emissions showed a smaller reduction rate (−7.2%) than that of $PM_{2.5}$, decreasing from
40.1 Tg in 2015 to 37.2 Tg in 2017. Compared with the emission reduction rate during 2015–2017, both $PM_{2.5}$ and $PM_{10}$
showed larger emission reduction rates during 2018–2020, estimated to be 27.2% and 25.5%, respectively. The emission
reductions in each year were also larger, especially for $PM_{10}$. For example, $PM_{2.5}$ and $PM_{10}$ emissions reduced by about 19.3%
and 14.0% in 2019 compared to 2018. This may have been due to that in addition to the strict controls imposed on the industrial
and power sectors during the 2013–2017 action period, the residential emissions have been strengthened during the 2018–
2020 action period. In particular, "coal-to-electricity" and "coal-to-gas" strategies were vigorously implemented in northern
China during the 2018–2020 action to reduce coal consumption and related air pollutant emissions (Liu et al., 2016; Wang et
al., 2020a). Thus, our inversion results confirm the effectiveness of the controls on residential emissions in terms of reducing
the emissions of $PM_{2.5}$ and $PM_{10}$. In addition, the control of non-point sources, such as blowing-dust emissions, was also
strengthened during the 2018–2020 action period, which is consistent with the faster reduction of $PM_{10}$ emissions during 2018–
2020. The annual trends of $PM_{2.5}$ and $PM_{10}$ emissions were also calculated in China using the Mann–Kendall trend test and
the Theil–Sen trend estimation method, the results of which are summarized in Table 4. The calculation of emission trends can
help extend the existing emission datasets forward in time to produce up-to-date products. The top-down estimated trends of
$PM_{2.5}$ and $PM_{10}$ emissions were −1.4 and −2.6 Tg/year during 2015–2020, attributable to the strict emission control measures
imposed during the two clean air action plans. As mentioned, the decreasing trends were larger during 2018–2020 (−1.5 and
−4.6 Tg/year) than during 2015–2017 (−1.1 and −1.5 Tg/year).
On the regional scale (Fig. S3), it can be clearly seen that the $PM_{2.5}$ emissions decreased consistently over all regions, by
59.8% in NCP, 49.6% in SE, 39.5% in NE, 35.8% in SW, 33.2% in NW, and 41.0% in Central, from 2015 to 2020. The NCP
region showed the largest reduction in emissions among the six regions, with its emission reduction rate being almost larger
than 10% in each year. This is consistent with the strictest emission control policies having been imposed over the NCP region.
The SE region showed a similar emission reductions to the NCP region, with its emission reduction rate being larger than 10%
in most years. Obvious increases of $PM_{2.5}$ emissions could be found over the NW region from 2013 to 2015 owing to the
increase in the number of observation sites in those years. After 2015, $PM_{2.5}$ emissions generally decreased over the NW region,
while there was a slight rebound in $PM_{2.5}$ emissions in 2016 and 2018, possibly due to the influences of the errors in fine dust
emission. The Central region showed different characteristics of emission changes to the other regions insofar as it showed
little change in $PM_{2.5}$ emissions during 2015–2018 but large reductions in 2019. This may be consistent with the control of
emissions over the Fenwei Plain area (the part of the Central region where the emission intensity is largest) being weak during
the 2013–2017 action plan but strengthened during the 2018–2020 action plan. In terms of the $PM_{2.5}$ emission trends over the
different regions, the calculated $PM_{2.5}$ emission trends were about −0.32 Tg/year in NCP, −0.32 Tg/year in SE, −0.24 Tg/year
in NE, −0.21 Tg/year in SW, −0.09 Tg/year in NW, and −0.15 Tg/year in Central, from 2015 to 2020.
The changes of $PM_{10}$ emissions were generally similar to those of $PM_{2.5}$, i.e., with decreases in all regions from 2015 to
2020 (Fig. S4). The top-down estimated $PM_{10}$ emission reductions from 2015 to 2020 were about 3.5 Tg (40.0%) in NCP, 2.6
Tg (35.5%) in SE, 3.0 Tg (36.6%) in NE, 2.0 Tg (35.9%) in SW, 1.0 Tg (25.3%) in NW, and 1.3 Tg (21.6%) in Central; and
the calculated trends were about −0.64 Tg/yr, −0.52 Tg/yr, −0.51 Tg/yr, −0.40 Tg/yr, −0.20 Tg/yr, and −0.27 Tg/yr,
respectively. However, due to the influences of the changes in the number of observation sites, the $PM_{10}$ emissions over the
NE, SW and NW regions increased substantially from 2013 to 2015, while they decreased in almost all years after 2015.
Different from the other regions, the Central region showed increases in $PM_{10}$ emissions from 2015 to 2018, by about 0.92 Tg
(14.9%), but substantial decreases in 2019 and 2020. The result also shows that most $PM_{10}$ emission reductions were achieved
during the 2018–2020 action plan. According to CAQIEI, the $PM_{10}$ emissions decreased by 0.64–2.3 Tg (17.4–31.8%) from
2018 to 2020, which accounted for 48.4–169.0% of the total reduction in emissions from 2015 to 2020. This again emphasizes
the effectiveness of the control of blowing-dust emissions during the 2018–2020 action plan.

### 4.2.2 Emission changes of gaseous air pollutants

#### 4.2.2.1 $SO_2$ and CO

Figure 7 shows the emission changes of different gaseous air pollutants in China from 2013 to 2020. Similar to the PM emissions, $SO_2$ and CO emissions decreased continuously during the two action plan periods, with top-down estimated emission reductions of about 9.6 Tg (54.1%) and 166.3 Tg (35.7%) for $SO_2$ and CO from 2015 to 2020, respectively. Meanwhile, both $SO_2$ and CO showed a significant decreasing trend from 2015 to 2020, with estimated trends of approximately $-2.1$ Tg/yr and $-36.0$ Tg/yr, respectively (Table 5). The reductions in $SO_2$ and CO emissions are closely consistent with the strict emission control measures imposed during the action plan periods, such as the phasing out of outdated industrial capacity and high-emitting factories, the strengthening of emission standards for industry and the power sector, the elimination of small coal-fired industrial boilers, and the replacement of coal with cleaner energies, which reflects the effectiveness of the emission control measures during the two action plan periods. Reductions of $SO_2$ emission were generally steady during the two action plan periods, which were approximately 4.2 Tg (23.8%) from 2015 to 2017 and 2.5 Tg (23.5%) from 2018 to 2020. However, CO showed a different emission reduction rate during the two action plan periods, with its emission reductions (67.1 Tg, 18.3%) during 2018–2020 being larger than those (45.6 Tg, 9.8%) during 2015–2017. This contrast may reflect the different emission control policies during the two clean air action periods, as well as the different emission distributions among the sectors between $SO_2$ and CO. According to the estimates of Zheng et al. (2018), the share of emissions from the industrial and power sector for $SO_2$ (77%) is nearly double that for CO (39%). Thus, the smaller reduction of CO emissions than that of $SO_2$ during 2015–2017 provides evidence that the 2013–2017 action plan mainly focused on controlling the emissions from the industrial and power sectors. During the 2018–2020 action plan, strict control measures targeted on the residential and transportation sectors were also implemented, which together account for 61% of CO emissions but only 23% of $SO_2$ emissions. As a result, CO showed a larger emission reduction rate during 2018–2020, while the emission reduction rate for $SO_2$ was similar to that during 2015–2017. The calculated trends of $SO_2$ and CO emissions during the two action plans are presented in Table 4, which are $-2.1$ Tg/yr and $-1.3$ Tg/yr for $SO_2$, and $-22.8$ Tg/yr and $-33.5$ Tg/yr for CO, respectively.

The reduction of $SO_2$ and CO emissions was also evident on the regional scale (Fig. S5 and S6). According to the top-down estimation, the reduction of $SO_2$ emissions ranged from 0.44 to 2.42 Tg (41.7–69.9%) from 2015 to 2020, with the NCP region exhibiting the largest reductions. The calculated decreasing trend of $SO_2$ emissions was also significant over all regions, ranging from $-0.08$ Tg/yr over the NW region to $-0.57$ Tg/yr over the NCP region (Table 5). With regards to the emission reduction rate during the different action plans, the results suggest that the emission reduction rate of $SO_2$ was higher during 2015 – 2017 (by 20.8–39.8%) than that during 2018–2020 (16.6–29.0%) over the NCP, SE, NE and SW regions. This may have been because, after the strict emission controls imposed upon industry and power plants during the 2013–2017 action plan, the room for further reductions in $SO_2$ emissions become smaller during the 2018–2020 action plan over these regions. Although residential and vehicle emissions were controlled more strictly during the 2018–2020 action plan, in total they account for ~20% of anthropogenic $SO_2$ emissions in China (Zheng et al., 2018). Thus, the enhanced reductions in $SO_2$ emissions from the residential and transportation sectors may not have been able to fully compensate for the weakened reductions from the industrial and power sectors, leading to a smaller $SO_2$ emission reduction rate over these regions. In contrast, the $SO_2$ emission reduction rate during 2018–2020 (31.1–34.8%) was higher than that during 2015–2017 (14.1–20.4%) over the NW and Central regions. This may have been due to the fact that the emission controls over the NW and Central regions were relatively weak during the 2013–2017 action plan (as also evidenced by the emission reduction rates of other species) owing to its less-developed economy. During the 2018–2020 action plan, the emission controls over these two regions were strengthened, which led to their higher emission reduction rates. Accordingly, the enhanced $SO_2$ emission reduction rates over the NW and Central regions compensated for the weakened reduction rates over the other regions, leading to a steady $SO_2$ emission reduction rate on the national scale.

The reductions of CO emissions from 2015 to 2020 were approximately 14.9–42.3 Tg (21.6–51.4%) over the different regions of China, with significant decreasing trends ranging from −3.0 to −8.7 Tg/yr (Fig. S6 and Table 5). Consistent with the comparisons of national CO emission reduction rates between the two action plans, the emission reduction rates during 2015–2017 (4.4–24.6%) were estimated to be smaller than those during 2018–2020 (12.2–24.6%) over all the different regions except the Central region, where the CO emission reduction rate was similar during the two action plans (Fig. S6).

### 4.2.2.2 $NO_x$ and NMVOCs

The top-down estimated $NO_x$ and NMVOC emissions showed different changes to the other four species, by increasing during 2015–2017 but declining during 2018–2020. Specifically, $NO_x$ emissions increased slightly by 5.9% from 2015 (25.2 Tg) to 2017 (26.6 Tg), with a non-significant increasing trend of 0.74 Tg/yr. Then, $NO_x$ emissions began to decrease in 2018, with a top-down estimated emission reduction and calculated trend of approximately 3.1 Tg (12.7%) and −1.6 Tg/yr, respectively, from 2018 to 2020. NMVOCs showed stronger emission increases than did $NO_x$, with top-down estimated emission increases of approximately 12.7 Tg (27.6%) and a calculated emission trend of about 6.3 Tg/yr from 2015 to 2017. Similar to $NO_x$, NMVOC emissions began to decrease after 2018, with a top-down estimated reduction of approximately 2.6 Tg (−4.4%) from 2018 to 2020, and a calculated trend of about −1.3 Tg/yr.

The increases of $NO_x$ and NMVOC emissions during 2015–2017 suggest that the 2013–2017 action plan may not have achieved desirable mitigation effects on these two species. For $NO_x$ emissions, the upward trend may have been associated with the following factors. On the one hand, vehicle exhaust is one of the most important sources of $NO_x$ in China, accounting for 31% of all $NO_x$ emissions nationally (Zheng et al., 2018). From 2013 to 2017, the number of vehicles in China continued to increase and reached 310 million in 2017, approximately 33.5% higher than in 2013 (MEE, 2017), which led to increases of $NO_x$ emissions from vehicles in China. On the other hand, although the 2013–2017 action plan was effective in reducing the $NO_x$ emissions from coal-fired power plants by promoting denitrification facilities and an ultra-low emission standard, the mitigation impacts on industrial $NO_x$ emissions may have been relatively small. For example, Wang et al. (2019a) compiled a unit-based emissions inventory for China's iron and steel industry from 2010 to 2015, based on detailed survey results of approximately 4900 production facilities in mainland China. They found that there were almost no $NO_x$ control measures in China's iron and steel industry during 2010–2015, resulting in a 12.4% increase in China's $NO_x$ emissions from the iron and steel industry in 2015 compared to 2010. In addition, although the penetration rate of denitrification facilities in China's cement industry reached 92% in 2015, the actual operating rate of denitrification facilities in the cement industry was not desirable, due to the lack of online emission monitoring systems. According to the research results of the Ministry of Ecology and Environment, 800, 1300, and 1400 cement production kilns were equipped with selective non-catalytic denitrification facilities from 2013 to 2015, but the actual operating rates were only 51%, 54% and 73%, respectively (Liu et al., 2021). In addition, the new precalciner kilns used in the cement industry have a higher $NO_x$ emission factor, such that the shift from traditional vertical kilns to precalciner kilns has to some extent increased the cement industry's emissions of $NO_x$ (Liu et al., 2021). Thus, there is evidence that the mitigation effects of the industrial control measures on $NO_x$ emissions may not be as significant as expected. Overall, the increased number of vehicles may have offset the emission mitigation effects brought about by the control of power plants, and the mitigation effects of controlling industrial $NO_x$ emissions were also undesirable. Consequently, $NO_x$ emissions in China may not have decreased, and even increased slightly, during the 2013–2017 action plan. Figure S7 further shows the changes in $NO_x$ emissions over different regions of China, revealing that $NO_x$ emissions over the NCP, SE, NE and SW regions were roughly unchanged (by less than 5%) from 2015 to 2017, while they increased over NW (18.6%) and Central (17.5%). This is consistent with previous results and indicates that $NO_x$ emissions may have increased over the NW and Central regions, possibly due to their increased human activities and weak emission controls.

In terms of NMVOC emissions, since the inversion results did not differentiate between anthropogenic and biogenic sources, the changes in NMVOC emissions may have been related to both anthropogenic and biogenic emissions. With respect

to anthropogenic emissions, previous bottom-up studies have suggested that China's NMVOC emissions did not decline during the 2013–2017 action plan, due to the lack of effective control measures on the chemical industry and solvent use (Zheng et al., 2018; Li et al., 2019c). According to the estimates of Li et al. (2019c), China's NMVOC emissions from solvent use increased by 11.1% in 2017 compared to those in 2015. Meanwhile, the increase in the number of vehicles in China may also have led to an increase in NMVOC emissions from transportation. Thus, the increases of NMVOC emission during 2015–2017 estimated by our inversion inventory may be related to the increases in anthropogenic NMVOC emissions from the chemical industry, solvent use, and vehicles. For the trends of biogenic NMVOC emissions, the CAMS global emission inventory shows that there were only little changes in the biogenic NMVOC emissions in China from 2013 to 2018 (Sect. 4.3.3), suggesting little contributions of the biogenic sources to the increased NMVOC emission in China. Figure S8 further shows the changes in NMVOC emissions over different regions of China, which suggests consistent increases in NMVOC emissions from 2015 to 2017 over different regions. According to the top-down estimations, NMVOC emissions increased by 30.5%, 25.2%, 18.5%, 10.9%, 50.5% and 63.1% over the NCP, SE, NE, SW, NW and Central regions, respectively. Again, the NW and Central regions exhibited the largest emission increases among the six regions, which is consistent with their elevated levels of human activity and weak emission controls.

The decrease in $NO_x$ and NMVOC emissions after 2018 suggests that the emission control strategy of the Chinese government had reached a point of optimization. The 2018–2020 action plan not only strengthened the controls over the industrial and power sectors, but also the transportation sector, especially for diesel vehicles with high $NO_x$ emissions. For example, the Chinese government released the "Action Plan for the Control of Diesel Trucks", and vigorously promoted an adjustment of the transportation structure of China by gradually improving the availability of rail transport. As a result, there was a downward trend in $NO_x$ emissions in China. The top-down estimated reductions of $NO_x$ emissions were approximately 0.81 Tg (17.2%) over NCP, 0.98 Tg (14.0%) over SE, 0.37 Tg (9.4%) over NE, 0.51 Tg (12.2%) over SW, 0.13 Tg (11.0%) over NW, and 0.32 Tg (9.2%) over Central (Fig. S7). The decrease in NMVOC emissions after 2018 may on the one hand have been related to the strengthening of vehicle controls during the 2018–2020 action plan, whilst on the other hand it may have been related to the promotion of clean heating plans in the northern region, which reduced the emissions of NMVOCs from residential sources. However, the decreases in NMVOC emissions were smaller than those in $NO_x$, which were estimated to be 0.84 Tg (6.9%) over NCP, 0.47 Tg (2.8%) over SE, 0.98 Tg (10.1%) over NE, and 0.53 Tg (14.1%) over NW (Fig. S6). Different from other regions, the NMVOC emissions over the SW and Central regions remained almost unchanged during the 2018–2020 action plan (Fig. S8).

**4.2.3 Changes in the distribution pattern of emissions in China**

Due to the different emission control intensities over the different regions of China, the emission distribution patterns of the different species may also have been altered, which could have influenced the distributions of air pollution in China. Based on CAQIEI, we further investigated the emission distribution patterns, as well as their changes, during the two action plans. Maps of the emission changes of different species during 2015–2017 and 2018–2020 are presented in Fig. 8. The shares of emissions in 2015, 2017 and 2020 by each subregion of China are also presented (Fig. 9). It can be seen that the emission changes during the 2015–2017 were more heterogenous than those during 2018–2020. The air pollutant emissions after the 2018–2020 action plan showed consistent reductions over most regions of China, while there were obvious emission increases detected from 2015 to 2017. This is consistent with the different emission control effects during the two clean air action plans as mentioned in previous sections. Due to its strictest emission control policies, the NCP region showed consistent emission reductions of $SO_2$, $NO_x$, CO, $PM_{2.5}$ and $PM_{10}$ during the two clean air action plans. Accordingly, the shares of emissions in the NCP region continued to decrease during the two action plan periods (Fig. 9). For example, the share of $SO_2$ emissions in the NCP region decreased from 19.4% to 15.4% during the period of 2015–2017, and from 15.4% to 12.7% during the 2018–2020 action plan. In contrast, NMVOC emissions increased obviously over the NCP region from 2015 to 2017, and decreased during

2018–2020. However, its share did not change significantly, being roughly 20% throughout both periods. As for other regions,
increases of $SO_2$, $NO_x$, $PM_{2.5}$, $PM_{10}$ and NMVOC emissions during 2015–2017 could be found over the Central region. More
specifically, the emission increases were mainly located in the Fenwei Plain area of the Central region, which was due to the
fact that this area was not included as a key region of emission controls during the 2013–2017 action plan. However, the
Fenwei Plain area was added as a key emission control region during the 2018–2020 action plan, which is consistent with the
emission reductions for these species over the Central region (Fig. 8). As a result, the shares of $SO_2$ and $PM_{2.5}$ emissions in the
Central region increased during 2015–2017 but decreased during 2018–2020 (Fig. 9). However, the shares of $NO_x$, $PM_{10}$ and
NMVOC emissions continued to increase over the Central region during the two clean air action plans, which suggests larger
roles of air pollutant emissions in that region. In contrast, the share of CO emissions in the Central region continued to decrease
in the two action plans, from 17.7% in 2015 to 13.4% in 2020.
In terms of the shares of emissions in eastern and western China, the top-down estimation suggests an increased share of
$NO_x$, $PM_{2.5}$, $PM_{10}$ and NMVOC emissions in western China after the two clean air action plans (Fig. 9), which indicates slower
emission reductions for these species in western China. However, the share of CO emissions in western China was reduced
after the two clean air action plans. Although the share of $SO_2$ emissions in western China increased during 2015–2017, it
turned to a decrease during 2018–2020.
**4.3 Comparisons with different emission inventories**
In this section, the CAQIEI is compared with the previous long-term bottom-up and top-down emission inventories in
China to validate our inversion results and provide the clues for the potential uncertainty in the current air pollutant emission
inventories. The bottom-up emission inventories used in the comparison include MEIC (Zheng et al., 2018), ABaCAS (Li et
al., 2023), HTAPv3 (Crippa et al., 2023), EDGARv6 (Jalkanen et al., 2012) and CEDS (Mcduffie et al., 2020), while the top-
down emission inventory is obtained from the updated Tropospheric Chemistry Reanalysis (TCR-2) (Miyazaki et al., 2020b).
However, it is difficult to directly compare our inversion results with these emission inventories considering that the inversion
emission includes both anthropogenic and natural emissions. To better compare our inversion results with previous inventories,
the natural emission sources, including soil $NO_x$ emissions and biogenic emissions obtained from the CAMS global emission
inventory (https://ads.atmosphere.copernicus.eu/cdsapp#!/dataset/cams-global-emission-inventories?tab=overview; last
accessed 26 July 2023) and the biomass burning emissions obtained from the Global Fire Assimilation System (GFAS) (Kaiser
et al., 2012) are taken as a reference to account for the influences of natural sources. The CAMS and GFAS emission inventory
are used because they are state-of-art natural emission inventories and can provide us with independent long-term estimations
of natural emissions. Since the latest year of most emission inventories is 2018, the comparisons were conducted between 2015
and 2018. Note that due to the complexity in the estimations of natural sources, significant uncertainty exists in the estimated
natural emissions. As a result, the comparison results would be sensitive to the used natural emission inventories, especially
for the species with large amount of natural emission, such as the NMVOC and particulate matter. Therefore, it should be
aware of that the comparison conducted here and the derived implications are on the basis of the natural emissions estimated
by CAMS and GFAS. In addition, the natural dust emissions are not considered in the comparisons, which would influence
the comparisons of the PM emissions.
**4.3.1 Magnitude**
**4.3.1.1 $NO_x$**
Figure 10 shows the average emissions of different air pollutants in China during 2015–2018 obtained from CAQIEI and
the previous emission inventories plus the natural sources we considered. Comparisons of the emission estimations on the
regional scale and gridded scale are also presented (Fig. 11 and Fig. S9). The results show that the CAQIEI has slightly higher
$NO_x$ emissions in China than the other inventories. Considering that CAQIEI includes both anthropogenic and natural sources,
this discrepancy could be explained by the natural $NO_x$ sources. According to the estimations of CAMS and GFAS, the soil
and biomass-burning $NO_x$ emissions are approximately 1.9 and 0.08 Tg/yr, which explains well the higher $NO_x$ emissions
given by CAQIEI. After consideration of the natural sources, MEIC, HTAPv3 and EDGARv6 agree well with our inversion
results on the national scale, with their differences within 1.0–7.4%. The $NO_x$ emission estimated by ABaCAS, CEDS and
TCR-2 are slightly lower than CAQIEI and other emission inventories. However, the differences between CAQIEI and these
inventories were found to range from 15.9% to 21.3%, which is within the previous estimated uncertainties of $NO_x$ emissions
in China (Kurokawa and Ohara, 2020; Li et al., 2017b; Li et al., 2023). These results suggest that the total $NO_x$ emissions in
CAQIEI are generally consistent with the current estimations of the anthropogenic and natural $NO_x$ emissions in China. On
the regional scale, the top-down estimated $NO_x$ emissions show good agreement with the previous emission inventories over
the NCP and SE regions, with their differences ranging from 1.0%–26.8%, suggesting good consistency in the estimations of
$NO_x$ emissions over these two regions. This makes sense because NCP and SE are the two most developed regions in China,
and where surveys and research on emissions are most sufficient. The differences are larger over the other regions. In the NE
region, CAQIEI has higher $NO_x$ emissions than the other inventories by 5–70%, suggesting higher anthropogenic or biomass-
burning emissions over there. The estimations made by MEIC, CEDS and TRC-2 are closer to our estimates, with their
differences being approximately 5.4–23.3%, while the differences are larger for ABaCAS, HTAPv3 and EDGARv6 (36.7–
70.0%). Over the SW and Central regions, there are large diversity in the previous emission inventories with estimations by
HTAPv3 and EDGARv6 almost double those of MEIC, ABaCAS, CEDS and TCR-2. The CAQIEI suggests a midst estimation
which is within the range of previous emission inventories. In the NW region, CAQIEI is consistently higher than other
inventories, by 22.7–64.2%, which suggests a potential missing source of the $NO_x$ emissions over this region.
**4.3.1.2 SO₂**
For SO₂ emissions, since natural sources contribute little (only about 0.02 Tg/yr) to them in China, the discrepancies
between CAQIEI and previous emission inventories are mainly attributable to the differences in anthropogenic emissions. As
shown in Fig. 10, CAQIEI agrees well with HTAPv3 and CEDS on the national scale, with their differences being
approximately ±2%, but is higher than MEIC, ABaCAS and TCR-2 by 17.4–32.9%. In contrast, EDGARv6 may have a
positive bias in its estimated SO₂ emissions, which are roughly double those of CAQIEI and other inventories. On the regional
scale, our results agree well with MEIC, ABaCAS, HTAPv3, CEDS and TCR-2 over the NCP region, with their differences
ranging from 1.0 to 18.1%. In the SE region, CAQIEI suggest lower SO₂ emissions than previous emission inventories, except
TCR-2. The differences are relatively smaller for the MEIC and ABaCAS inventories by around −15%, but larger for HTAPv3,
EDGARv6 and CEDS (ranging from −47.3% to −113.2%). In contrast, CAQIEI suggests higher SO₂ emissions than all
previous emission inventories over the NE region by about 14.8–132.0%, indicating possible missing sources over there.
Similarly, the CAQIEI and HTAPv3 suggests higher SO₂ emissions than the MEIC, ABaCAS, CEDS and TCR-2 by 27.0–
75.6% in the NW region, and by 44.3–77.7% in the Central region.
**4.3.1.3 CO**
For CO emissions, CAQIEI is substantially higher than the previous emission inventories, with the estimated CO
emissions of CAQIEI being about three times higher than the bottom-up inventories and more than double those of the top-
down estimates made by TCR-2. According to GFAS, the average rate of CO biomass-burning emissions in China from 2015
to 2018 was about 3.4 Tg/yr. Yin et al. (2019), based on MODIS fire radiative energy data, also estimated China's CO biomass-
burning emissions to be about 5.0 (2.3–7.8) Tg/yr. The biogenic CO emissions obtained from the CAMS global emission
inventory were approximately 2.3 Tg/yr. According to these estimates, natural CO emissions in China have a magnitude of
about $10^1$, which is rather small compared with anthropogenic sources, and cannot explain the large discrepancies between
CAQIEI and other inventories. Thus, the CAQIEI suggest much higher anthropogenic CO emissions in China than the existing
emission inventories. In fact, the potential underestimation of CO anthropogenic emissions has been investigated in previous
studies and is regarded as the main reason for the negative bias in global or hemispheric CO simulations (Stein et al., 2014;
Gaubert et al., 2020). Regionally, Kong et al. (2020) compared a suite of 13 modeling results from six different CTMs—
namely, NAQPMS, CMAQ, WRF-Chem, NU-WRF, NHM-Chem and GEOS-Chem—with observations over the NCP and
Pearl River Delta regions under the framework of the Model Inter-Comparison Study for Asia III (MICS-Asia III), and found
consistent negative biases in the CO simulations of all models, pointing toward potential underestimations of CO emissions in
China. Previous inversion studies have also reported higher a posteriori CO emissions than their *a priori* emission inventories
(Bergamaschi et al., 2000; Miyazaki et al., 2012; Petron et al., 2002; Petron et al., 2004; Tang et al., 2013; Gaubert et al., 2020).
For example, the constrained CO emissions reported by Gaubert et al. (2020) are 80% higher than the CEDS over the northern
China. Our inversion results are consistent with these inversion studies, suggesting higher anthropogenic CO emissions in
China. However, direct evidence in support of such high CO emissions in China reported by our study is still limited currently.
Thus, we compiled more inversion results within the period of 2013–2020 from previous studies to further validate our
inversion results, which are summarized in Table 6. It can be clearly seen that there are large differences in the estimated CO
emissions between the inversion results based on surface observations and those based on satellite data. Our inversion results
are consistent with the results of Feng et al. (2020), with China's CO emissions in December 2017 estimated at approximately
1500.0 kt/day and 1388.1 kt/day, respectively. In addition, Feng et al. (2020) used the CMAQ model to constrain CO emissions,
which is different from the model we used. This may indicate that the model uncertainty would not significantly influence the
inversion results of CO emissions. However, the top-down estimated CO emissions based on satellite data (163.6–553.4 kt/day)
are much lower than those based on surface observations, although they are all higher than their *a priori* emissions. The lower
CO emission estimations based on satellite data assimilation may be attributable to the lower sensitivities of satellite data to
surface concentrations, suggesting that the assimilation of satellite data alone may not be adequate to correct the negative
biases in the *a priori* emissions. This deficiency has also been revealed by Miyazaki et al. (2020b), who found undercorrected
surface CO emissions in the extratropic of the Northern Hemisphere in TCR-2. However, the assimilation of surface
observations can be influenced by the uncertainties in the modeled vertical mixing, which could lead to the uncertainties in the
inversed CO emissions based on surface observations. Therefore, the inversed CO emissions in CAQIEI could be partly
supported by previous inversion studies based on surface observations, but more evidence is still needed to justify the
magnitude of the inversed CO emissions. Besides anthropogenic sources, the chemical production of CO via oxidation of
methane ($CH_4$) and NMVOCs, as well as the CO sinks via the hydroxyl radical (OH) reaction, also influence the simulation
of CO (Stein et al., 2014; Gaubert et al., 2020; Müller et al., 2018). Due to the important role of OH in the chemical production
and sinks of CO, the inversion of CO emissions is sensitive to the modeled OH abundance and the emissions of $CH_4$ and
NMVOCs. According to the estimation of Müller et al. (2018), the magnitude of inversed CO emissions in China could differ
by more than 40% when different levels of OH concentrations are used in the model. Thus, the much higher estimations of
CO emissions in our inversion results may also be partly explained by the underestimation of CO chemical production or the
overestimation of the CO sink.
**4.3.1.4 PM$_{2.5}$**
In terms of PM$_{2.5}$, the CAQIEI suggests higher emissions than ABaCAS, HTAPv3 and EDGARv6 by about 20%, and by
47.7% than MEIC on the national scale. Larger discrepancies mainly occur in the NE and NW regions, where CAQIEI is about
27.2–114.9% and 83.2–143.2% higher than the previous inventories. The differences in the estimated PM$_{2.5}$ emissions may be
related to the uncertainties in the biomass-burning or anthropogenic sources in the NE region (Wu et al., 2020b), while in the
NW region, the errors in the fine-dust emissions may also contribute to the differences in the estimated PM$_{2.5}$ emissions there.
The differences in the estimated PM$_{2.5}$ emissions are relatively smaller in the NCP and SE regions, ranging from −18.9% to
20.4%, suggesting better agreement in the estimated PM$_{2.5}$ emissions over these two regions. In the SW region, CAQIEI is
closer to HTAPv3 and EDGARv6, with their differences being about 6.3% and −9.5% respectively, and is higher than MEIC
and ABaCAS by 54.2% and 28.6%, suggesting higher uncertainty in the estimated PM$_{2.5}$ emissions over there.

**4.3.1.5 PM$_{10}$**

For PM$_{10}$ emissions, it is difficult to directly compare CAQIEI with previous emission inventories since CAQIEI not only
contains anthropogenic and biomass-burning emissions, but also coarse-dust emissions. As a result, the estimated emissions
of PM$_{10}$ by CAQIEI are substantially higher than those by previous inventories, especially over the NW, Central and NE
regions (Fig. 11), which are the typical natural windblown dust-source regions in China (Zeng et al., 2020). Besides the
naturally windblown dust of arid desert regions (Prospero et al., 2002), large amounts of coarse-dust emissions also stem from
anthropogenic sources, including anthropogenic fugitive, combustion and industrial dust from urban sources (AFCID) (Philip
et al., 2017), and anthropogenic windblown dust from human-disturbed soils due to changes in land-use practices, deforestation
and agriculture (Tegen et al., 1996). Therefore, although the other regions are not typical natural windblown dust-source
regions in China, there are still high levels of coarse dust emissions from anthropogenic sources there (also called "urban
dust"), which may be the main reason for the large deviation in the estimated PM$_{10}$ emissions between CAQIEI and previous
inventories. On the one hand, although AFCID is included in MEIC, ABaCAS, HTAPv3 and EDGARv6, it is difficult for
current bottom-up emission inventories to completely represent fugitive sources (Philip et al., 2017). On the other hand, the
anthropogenic windblown dust emissions have not been included in current bottom-up emission inventories, which is an
important source of coarse dust in urban areas according to the estimations of Li et al. (2016) and the another important
contributor to the differences between CAQIEI and previous emission inventories.

**4.3.1.6 NMVOCs**

For NMVOC emissions, since CAQIEI includes both anthropogenic and natural sources, its estimated NMVOC emissions
are much higher than those estimated by previous emission inventories. After consideration of natural sources, the CAQIEI
suggests close estimations of the NMVOC emissions with the MEIC, HTAPv3 and CEDS inventories on the national scale,
with their differences being about 1.5–12.5%. The estimated NMVOC emission by ABaCAS and EDGARv6 are slightly lower
than CAQIEI by 17.8% and 24.6%, respectively. On the regional scale, the CAQIEI suggests higher NMVOC emissions over
the northern China (NCP, NE and NW), with the top-down estimated NMVOC emissions about 30.4–81.4%, 27.3–72.1%,
79.3–116.8%, and 8.7–57.5% higher than those of the previous emission inventories. In contrast, the CAQIEI suggests lower
NMVOC emissions over the SE region, with the estimated NMVOC emissions of CAQIEI being about 21.2–27.6% lower
than those of MEIC, ABaCAS, HTAPv3 and CEDS. These results are consistent with the previous inversion results based on
the satellite observations, which suggest higher NMVOC emissions over the NCP region and lower NMVOC emissions over
the south China (Souri et al., 2020). Over the SW region, CAQIEI shows good agreement with MEIC, ABaCAS and CEDS,
with CAQIEI being slightly lower than these inventories by 1.0–8.9%, but is lower than HTAPv3 and EDGARv6 by about
38.6% and 29.1%, respectively. Again, it should be noted that the comparisons of NMVOC emission are conducted on the
basis of natural emissions estimated by CAMS and GFAS, and could be more sensitive to the used natural sources than other
species considering the larger contributions of the natural source to the NMVOC emissions.

**4.3.2 Seasonality**

Figure 12 presents the monthly profiles of different air pollutants obtained from different emission inventories. Note that
the natural sources have been added to the previous inventories to facilitate the comparisons. The results show that different
emission inventories give similar monthly profiles of NO$_x$ and CO emissions, with higher emissions during wintertime and
lower emissions during summertime, which suggests relatively lower uncertainty in the estimated monthly profiles for these
two species. For SO$_2$ emissions, CAQIEI yields stronger monthly variation than the other inventories, with a higher proportion
from January to March and lower proportion during summertime. Due to the influences of dust emissions, the top-down
estimated PM$_{2.5}$ and PM$_{10}$ emissions show higher proportions than the other emission inventories during the spring season,
especially for PM$_{10}$. However, the proportion of emissions during autumn and winter are lower than in the other inventories.
The monthly profiles of NMVOC emissions are generally consistent, with higher emissions during summer due to the enhanced
biogenic emissions. However, the profile of CAQIEI is flatter than the previous inventories, and suggests a higher proportion
during springtime. In addition, the timings of peak values of NMVOC emissions are also different between CAQIEI and the
previous inventories, with CAQIEI showing peak values during May–July but the other inventories suggesting peaks during
June–August.

### 4.3.3 Emission changes during 2015–2018

The top-down estimated emission changes of different air pollutants during 2015–2018 were also compared with previous
emission inventories. Figure 13 shows the time series of the total emissions of different species from 2013 to 2020 obtained
from the CAQIEI and other emission inventories. Comparisons of the emission changes over the regional scales are also
presented in Fig. S10–S15. Before the comparison, we firstly analyze the trends of natural sources in China to investigate their
influences on the emission changes of different species based on the CAMS emission inventory and GFAS. Note that we only
consider the soil, biogenic and biomass-burning emissions for the natural sources; the trends of dust emissions in China are
not analyzed, which may lead to uncertainty when comparing the emission changes of PM$_{2.5}$ and PM$_{10}$. As shown in Fig. S16,
the natural sources of NO$_x$ and NMVOC emissions changed little during 2013–2018. The other species had small decreasing
trends from 2013 to 2018. However, considering the small contributions of natural sources to their emissions, these small
trends would not significantly influence their emission trends. For the dust emissions, previous studies have indicated a
declining trend in dust activity in China from 2001 to 2020 (Wu et al., 2022; Wang et al., 2021), due to weakened surface wind
and increased vegetation cover and soil moisture. These results suggest that the emission trends in the CAQIEI would be
mainly driven by the anthropogenic sources for the gaseous air pollutants based on the estimations of CAMS and GFAS, while
its estimated emission trends of PM$_{2.5}$ and PM$_{10}$ would be influenced by the declining trends in dust emissions in China, which
should be noted when comparing the emission changes of PM$_{2.5}$ and PM$_{10}$.
As shown in Fig. 14, all the emission inventories agree that the NO$_x$, SO$_2$, CO, PM$_{2.5}$ and PM$_{10}$ emissions in China were
reduced from 2015 to 2018, except for the increases of CO emissions estimated by TCR-2, which confirms the effectiveness
of the emission control policies implemented during the clean air action plans. Meanwhile, most emission inventories agree
that SO$_2$ is the species with the largest emission reduction rate, followed by PM$_{2.5}$, indicating better emission mitigation effects
of these two species (Fig. 14). However, the CAQIEI suggested lower emission reduction rates than the other emission
inventories for most species, especially for NO$_x$, PM$_{10}$ and NMVOCs (Fig. 14). The estimated emission reduction rate of NO$_x$
obtained from CAQIEI is about −2.7%, which is lower than the values of MEIC (−9.7%), ABaCAS (−23.0%), HTAPv3
(−13.0%) and CEDS (−9.0%). As we discussed in Sect. 4.2.2.2, the small reductions of NO$_x$ emission in CAQIEI would be
related to the increased vehicle emissions and the undesirable mitigation effects of the industry control. In fact, these factors
have been considered in some bottom-up emission inventories, such as MEIC. The differences between our inversion results
and previous inventories thus reflect uncertainty in the quantifications of the effects of these factors on the NO$_x$ emissions due
to the lack of sufficient statistics on mobile vehicle or other sectors. Our inversion results suggest larger adverse effects of
these two factors on the reductions of NO$_x$ emissions in China. According to Fig. S17, the differences between CAQIEI and
these inventories mainly occur in the SE, SW, NW and Central regions, with the emission reduction rate estimated by CAQIEI
being substantially lower than those estimated by previous inventories. In particular, CAQIEI suggests increases of NO$_x$
emissions over the Central region, which is opposite to the previous emission inventories. Better agreement is achieved over
the NCP and NE regions, with the emission reduction rate estimated by CAQIEI being closer to those of MEIC, HTAPv3 and
CEDS. The $NO_x$ emission reduction rates estimated by EDGARv6 (−3.3%) and TCR-2 (−1.7%) are closer to our results on
the national scale, but they estimated lower $NO_x$ emission reduction rate than our estimate over the NCP and NE regions.

Similarly, the emission reduction rate of $PM_{10}$ obtained from CAQIEI (−10.8%) is lower than those estimated by MEIC

(−27.9%), ABaCAS (−33.0%) and HTAPv3 (−27.8%) on the national scale (Fig. 14). A lower $PM_{10}$ emission reduction rate
of CAQIEI than these inventories also exist in the different regions of China, except SW (Fig. S17). In particular, different
from previous emission inventories, CAQIEI suggests that $PM_{10}$ emissions may have actually increased over the Central region.
Considering that dust emissions may have decreased from 2015 to 2018 owing to weakened dust events (Wang et al., 2021),
the increase in $PM_{10}$ emissions over the Central region may reflect the increases in anthropogenic sources. Meanwhile, we also
found that CAQIEI estimated the emission reduction rate of $PM_{10}$ to be smaller than that of $PM_{2.5}$. This is different from
previous emission inventories, which show similar emission reduction rates for $PM_{2.5}$ and $PM_{10}$. Considering that $PM_{10}$
emissions include $PM_{2.5}$ and PMC emissions, the lower emission reduction rate of $PM_{10}$ than $PM_{2.5}$ in CAQIEI suggests that
PMC emissions may have decreased slower than $PM_{2.5}$ emissions from 2015 to 2018.

In terms of NMVOCs, most previous inventories, including MEIC, EDGARv6 and CEDS, suggest a weak decrease in

China, with the estimated rates of change in emissions ranging from −0.8% to −4.6%. The emission reduction rate estimated
by ABaCAS is larger, reaching up to −14.2%. In contrast, the CAQIEI suggests an opposite emission change to these
inventories, with estimated NMVOC emissions increasing by 26.6% from 2015 to 2018. HATPv3 also suggests an increase in
NMVOC emissions, but with a much lower rate of increase (2.7%). Similar results could also be found on the regional scale
(Fig. S17), especially over the NCP, NE and Central regions, where NMVOC emissions could have increased by 38.0%, 38.3%
and 60.0%, respectively, according to the estimates of CAQIEI. As we discussed in Sect. 4.2.2.2, the increases of NMVOC
emission estimated in CAQIEI may be related to the increased anthropogenic NMVOC emissions from the chemical industry,
solvent use, and vehicles. Therefore, similar to the $NO_x$ emissions, the differences between CAQIEI and previous inventories
reflects the uncertainty in the quantifications of the impacts of these factors, and suggest larger adverse effects of these factors
on the emission reductions of NMVOC emission than the previous inventories.

The differences in the estimated emission reduction rates between CAQIEI and previous inventories are relatively smaller

for $SO_2$ and $PM_{2.5}$ emissions. The emission reduction rate of $SO_2$ estimated by CAQIEI is close to that estimated by MEIC and
CEDS, ranging from −34.7% to −44.3%. ABaCAS and HTAPv3 estimate a larger emission reduction rate of about −58.5%
and −53.7%, respectively. EDGARv6 and TCR-2 may underestimate the reduction rate of $SO_2$, with estimates of only about
−7.0% and −9.1%, respectively. This may be because EDGARv6 underestimates the FGD (flue-gas desulfurization devices)
penetration or $SO_2$ removal efficiencies of FGD in China. On the regional scale (Fig. S17), the top-down estimated $SO_2$
emission reduction rate agrees reasonably with that of MEIC over the NCP, NE and SE regions, but these inventories estimate
different $SO_2$ emission reduction rates over the SW, NW, and Central regions. The reduction rates estimated by MEIC over
the SW and Central regions is higher than those given by CAQIEI, but lower over the NW region. The other emission
inventories also give different emission reduction rates, suggesting large uncertainty in the estimated $SO_2$ emission reduction
rates over these three regions. In terms of $PM_{2.5}$, CAQIEI's estimated emission reduction rate agrees well with those of MEIC
and HTAPv3 on the national scale, which is about 24–27% from 2015 to 2018. The emission reduction rate of $PM_{2.5}$ estimated
by EDGARv6 are lower than our estimates and other inventories, which were about 9%. On the regional scale, our results
show good consistency with MEIC and HTAPv3 over the NCP, NE, SE and SW regions, but they have large differences over
the NW and SW regions.

Different from the other species, the CO emission reduction rate estimated by CAQIEI (−21.3%) is higher than in most

of the previous inventories, including MEIC (−13.0%), ABaCAS (−11.6%), EDGARv6 (−4.7%), and CEDS (−11.7%),
suggesting larger mitigation effects on CO emissions than other inventories. HTAPv3 agrees with our results, with an estimated
emission reduction rate of about −22.0%. On the regional scale (Fig. S17), our result is consistent with MEIC over the NCP
and SE regions, with estimated emission reduction rates for CO of around 24% and 15%, respectively, while in other regions
the emission reduction rate estimated by CAQIEI is higher than that estimated by MEIC. The TCR-2 shows opposite changes
in CO emissions compared with the other inventories insofar as it suggests increases of CO emissions over different regions
of China. Since the emissions in TCR-2 are constrained by satellite observations, the differences between our results and those
of TCR-2 highlight that the observations used to constrain the emissions may have a large influence on the estimated emission
changes. In this case, the estimated changes of CO emissions by CAQIEI are more consistent with those estimated by other
bottom-up inventories. Considering this, the TCR-2 may have uncertainties in its estimated changes of CO emissions in China
from 2015 to 2017, which could be related the suboptimal performance of the data assimilation caused by the underestimated
background errors of CO or too short assimilation window for the CO emission estimates (Miyazaki et al., 2020).

### 4.4 Uncertainty estimation of CAQIEI

Finally, the uncertainty of the inversed emission inventory product is estimated in this section to facilitate users'
understanding of the data's accuracy. Within the framework of EnKF, the analysis perturbation $\mathbf{X^a}$ estimated by using Eq. (3)
could provide the information regarding the uncertainty of the inversed emission inventory. The Coefficient of variation
(hereinafter, CV), defined as the standard deviation divided by the average, with a larger value denoting higher uncertainty, is
calculated based on the analysis perturbation to measure the uncertainty of the inverse emission inventory. Based on this
method, the uncertainty (CV) of the a posteriori emission was estimated as follows: 92.3% ($PM_{2.5}$), 88.8% ($PM_{10}$), 26.7%
($SO_2$), 46.8% (CO), 31.8% ($NO_x$) and 65.5% (NMVOC). However, it should be noted that such uncertainty was only calculated
under the framework of the EnKF constructed in this study, which is dependent on the assigned value of the a priori emission
uncertainty, observation errors and the number of assimilated observations. In addition, we only considered the a priori
emission uncertainty and the observation errors during the inversion. The influences of the other error sources, such as
uncertainty in the chemistry transport model, meteorology simulations and the inversion method were not considered.
Therefore, the current estimated uncertainty should be considered as a lower bound for the real uncertainty. More systematic
analysis that thoroughly consider the uncertainty sources regarding the emission inversion should be conducted in future to
give a more accurate estimation of the uncertainty in our products.

### 5 Discussion and conclusion

A long-term, top-down emissions inventory of major air pollutants in China was developed and validated in this study by
assimilating surface observations from CNEMC using the modified EnKF method and NAQPMS. It includes gridded emission
maps of $NO_x$, $SO_2$, CO, primary $PM_{2.5}$, primary $PM_{10}$, and NMVOCs in China from 2013 to 2020, on a monthly basis, with a
horizontal resolution of 15 km × 15 km. This new top-down emissions inventory, named CAQIEI, provides new insights into
the air pollutant emissions and their changes in China during the country's two clean air action periods. The estimated total
emissions for the year 2015 in China are 25.2 Tg of $NO_x$, 17.8 Tg of $SO_2$, 465.4 Tg of CO, 15.0 Tg of $PM_{2.5}$, 40.1 Tg of $PM_{10}$
and 46.0 Tg of NMVOCs. Comparisons of CAQIEI with previous inventories, including MEIC, ABaCAS, HTAPv3,
EDGARv6, CEDS and TCR-2, on the basis of the natural emissions obtained from CAMS and GFAS showed reasonable
agreement for the estimation of $NO_x$, $SO_2$ and NMVOC emissions in China. The $PM_{2.5}$ emissions obtained from CAQIEI (13.2
Tg) are slightly higher than in the previous emission inventories (8.3–11.1 Tg), while the CO emissions estimated by CAQIEI
(426.8 Tg) are substantially higher than in previous inventories (120.7–237.7 Tg). However, the reasons for such a large gap
are still not clear, but might be attributable to both the underestimation of CO sources (e.g., anthropogenic, biomass-burning
and chemical-production sources) (Bergamaschi et al., 2000; Miyazaki et al., 2012; Petron et al., 2002; Petron et al., 2004;
Tang et al., 2013; Gaubert et al., 2020), and/or the overestimation of CO sinks in the model (Müller et al., 2018). In addition,
comparisons with previous inversion studies suggest there are larger differences in the top-down estimated CO emissions based
on surface and satellite observations. Our inversion results are consistent with previous inversions based on surface
observations, but are much higher than those based on satellite observations, suggesting large uncertainty in inversion-
estimated CO emissions in China. Therefore, more research is needed to better understand the reasons behind the negative
biases in CO simulation, and to explain the differences between our results and those of previous inventories. Similar to
situation with CO emissions, the $PM_{10}$ emissions estimated by CAQIEI (37.7 Tg) are also substantially higher than in previous
inventories (11.1–15.9 Tg). However, this will be mainly associated with the emissions of coarse dust, which were not included
in previous inventories. The estimation of dust emissions in China is subject to high levels of uncertainty, with the estimated
dust fluxes based on different dust emission schemes differing by several orders of magnitude (Zeng et al., 2020). Therefore,
our inversion results could provide a reference for the magnitude of coarse-dust emissions in China, which could then help to
reduce the large uncertainty in estimations of dust emissions in China.
Several potential important deficiencies in current emission estimations were also indicated by CAQIEI on the regional
scale. For example, the CAQIEI suggests substantially higher air pollutant emissions than the previous emission inventories
over the NW and Central regions. Thus, the air pollutant issues may be more severe than we expected over these two regions.
Meanwhile, our inversion results suggest higher NMVOC emissions over the northern China but suggest lower NMVOC
emissions in southern China, which is consistent with the previous inversion studies based on the satellite. China is now facing
increasingly severe $O_3$ pollution and has an urgent need for a coordinated control of $O_3$ and $PM_{2.5}$. Our results may provide
valuable information on the NMVOC emissions in China, which is important for a proper understanding of $O_3$ pollution and
the development of effective control strategies nationally. Higher emissions were also found in the NE region based on our
inversion results. The NE region is a typical area for open-area biomass burning, with significant emissions from straw
combustion (Wu et al., 2020b). The higher emissions estimated by our inversion result may indicate higher biomass-burning
emissions over there. This is consistent with recent estimations of biomass-burning emissions by Xu et al. (2023) and Wu et
al. (2020b), who showed higher biomass-burning emissions in China than previous estimations, including those of GFEDv4.1s
(https://www.globalfiredata.org/data.html), FINNv1.5 (https://www.acom.ucar.edu/Data/fire/), and GFASv1.2
(https://www.ecmwf.int/en/forecasts/dataset/global-fire-assimilation-system).
Based on CAQIEI, we further quantified the emission changes of different air pollutants in China during the two clean
air action plans. The results confirmed the effectiveness of these campaigns on the mitigation of air pollutant emissions in
China, with estimated emission reductions of 15.1% for $NO_x$, 54.5% for $SO_2$, 35.7% for CO, 44.4% for $PM_{2.5}$, and 33.6% for
$PM_{10}$ from 2015 to 2020. In contrast, NMVOC emissions increased by 21.0% from 2015 to 2020. Comparisons of the estimated
emission reduction rates during the two clean air action plans suggested that emission reductions were larger during the 2018–
2020 than during 2015–2017. The estimated rates of change in emissions were 5.9% for $NO_x$, −23.8% for $SO_2$, −9.8% for CO,
−14.5% for $PM_{2.5}$, −7.2% for $PM_{10}$, and 27.6% for NMVOCs during 2015–2017, which were smaller than the −12.1% for $NO_x$,
−23.5% for $SO_2$, −18.3% for CO, −26.6% for $PM_{2.5}$, −25.5% for $PM_{10}$, and −4.5% for NMVOCs during 2018–2020. On the
one hand, this is due to the fact that more sectors were controlled during the 2018–2020 action plan. Besides the industrial and
power sectors, which were the main points of control in the 2013–2017 action plan, the residential sector, transportation sector,
and non-point sources like blowing-dust emissions, were also strengthened in the 2018–2020 action plan. Consequently, the
emission reduction rates of CO, $PM_{2.5}$ and $PM_{10}$ during 2018–2020 were higher than those during the 2015–2017 when the
2013–2017 action plan was implemented. However, the reduction of $SO_2$ emissions was similar during the two action plan
periods. This is because most $SO_2$ emissions stem from the industrial sector and power plants, which together contribute about
77% of all emissions (Zheng et al., 2018). Thus, the additional control of other sectors in the 2018–2020 action plan may not
have significantly impacted the mitigation of $SO_2$ emissions. On the other hand, strict emission controls were implemented or
strengthened in more areas of China during the 2018–2020 action plans. For example, the inversion results indicated that there
were obvious increases of $SO_2$, $NO_x$, $PM_{2.5}$, $PM_{10}$ and NMVOC emissions during 2015–2017 over the Central region,
especially in the Fenwei Plain area, where the emission controls were relatively weak during the 2013–2017 action plan.
However, all species showed obvious emission reductions almost the whole China during the 2018–2020 action plan.
The estimated rates of change in emissions during 2015–2018 were also compared with those estimated by previous
emission inventories. Although both CAQIEI and previous inventories showed declines of air pollutant emissions in China,
the emission reduction rates estimated by CAQIEI were generally smaller than those estimated by previous inventories,
especially for $NO_x$, $PM_{10}$ and NMVOCs, suggesting a smaller mitigation effects of the air pollution control measures than the
previous emission inventories suggested. In particular, China's NMVOC emissions were shown to have increased by 26.6%
from 2015 to 2018, especially over NCP (38.0%), NE (38.3%) and Central (60.0%). CO was found to be an exception insofar
as the emission reduction rate estimated by CAQIEI was larger than that of most previous emission inventories, except in the
NCP region. The estimated emission reduction rates of $SO_2$ and $PM_{2.5}$ were relatively closer to those of previous inventories,
suggesting better consistency in the estimated emission reduction for these two species.
Overall, the inversion inventory developed in this study could provide us with value information on the complex variations
of air pollutant emissions in China during its two recent clean air action periods, which could help improve our understanding
of air pollutant emissions and related changes in air quality in China. For example, the increases of $O_3$ and nitrate
concentrations may be associated with the undesirable emission reduction effects of the 2013–2017 action plans. The estimated
lower $NO_x$ emission reduction rate by CAQIEI may also help explain the weak responses of nitrogen deposition fluxes to the
clean air action plans. Meanwhile, this top-down emissions inventory can be used to supply the input data for CTMs or server
as a comparable reference for future inversion studies based on other methods or observation data, which is expected to
improve the performance of model simulations and air quality forecasts, and facilitate the development of inversion method.
**6 Limitations**
However, due to the complexity of the emission estimation, it is inevitable that there are some limitations in our inversion
results. Here We summarise some issues that might affect the quality of the CAQIEI which were known at the time of
publication to assist the potential users in properly using this data products.
(1) The changes in the number of observation sites would induce spurious emission trends during 2013–2014, especially
over western China, although the influence of the number of observation sites is smaller over the NCP and SE regions because
of their higher density of observation sites. Therefore, it is recommended that not to use the emissions in 2013 and 2014 when
analyzing the emission trends in China. This limitation makes it difficult to estimate the overall emission control effects of
2013 – 2017 action plan. Consequently, the emission change rate during the 2015–2017 were sampled in this study to represent
the emission control effects of the 2013–2017 action plan, but it may not necessarily reflect the overall reduction rate of the
action plan for the entire period. In addition, although the number of observation sites has become stable since 2015, the limited
number of observation sites makes it difficult to fully constrain China's air pollutant emissions, especially for the natural
sources considering that the majority of the observation sites are located in the urban areas. Therefore, the uncertainty in the
estimated emissions over the remote areas are expected to be higher than those over the urban areas, especially for the species
with large amount of natural emission, such as PM and NMVOC. For example, the coarse-dust emissions over western China
are expected to be underestimated by CAQIEI because of the limited availability of observation sites. Therefore, adding
observations there will help improve the accuracy of the inversion estimates. For example, simultaneous assimilation of the
surface and satellite observation may help alleviate this problem and provide more constrains on the emissions without surface
observations.
(2) The natural and anthropogenic emissions are not differentiated in our inversion method, leading to higher emissions
of $PM_{10}$ and NMVOCs than in other emission inventories. This also hinders the comparisons of our inversion results with the
previous inventories. Therefore, potential readers should be aware of that the current comparisons of our inversion results and
previous inventories are on the basis of the natural emissions estimated by CAMS and GFAS, which does not necessarily
indicate large uncertainties in anthropogenic sources within the bottom-up inventories. The impacts are expected to be smaller

for the NO$_x$, SO$_2$ and CO due to the small contributions of natural sources to their emission, but would be larger for NMVOC and PM which has large amount of natural emission. Assimilation of isotope data, speciated PM$_{2.5}$ and NMVOC observations may help differentiate the natural and anthropogenic emissions, and address this problem in future.

(3) The NMVOC emissions may have larger uncertainty than the other species. On the one hand, a significant amount of NMVOC emission would originate from suburban or rural regions. Therefore, although the O$_3$ observations at the urban sites could provide information on the NMVOC emissions over the suburban or rural areas according to covariance estimated by the ensemble simulation, the NMVOC emissions may not be fully constrained due to the lack of observation sites over the suburban or rural areas. On the other hand, due to the lack of long-term NMVOC observations, the NMVOC emissions were constrained by the O$_3$ concentrations in this study. Although the feasibility of this approach has been demonstrated by previous inversion studies, the nonlinear NO$_x$-VOC-O$_3$ interactions could inevitably introduces greater uncertainty into the inversion of NMVOC than other species. Therefore, more attention should be paid while using the inversion results of NMVOC, and more robust analysis of the effects of nonlinear NO$_x$-VOC-O$_3$ interactions and the number of observation sites should be performed in future to better illustrate the feasibility of assimilating O$_3$ to constrain the NMVOC emissions.

(4) The errors in the meteorological simulation and the CTMs were not considered in the emission inversions, which would lead to uncertainty in our estimated emissions. For example, the errors in the simulated wind would influence the transportation of the air pollutant and lead to uncertainty in the emissions distributions. According to the evaluation results of meteorological simulations (Table S1), the simulated relative humidity is generally lower than the observations, which may weaken the formation of secondary aerosol. On the contrary, the simulated precipitation was higher than the observation for most regions which would lead to overestimations of the wet removal of air pollutants. As a result, there may be a positive tendency in the inversed emission inventory due to the errors in the simulated relative humidity and precipitation. Besides these parameters, the accuracy of the simulated boundary layer is also important for the performance of the emission inversions (Du et al., 2020), although it was not evaluated currently due to the lack of observation. If the WRF systematically underestimates the boundary layer, the vertical diffusions of the air pollutants would be suppressed, which would lead to overestimated surface air pollutant concentrations and a negative tendency in the inverse emission inventory. However, it is difficult to quantify the influences of the meteorological errors on the emission inversions, as the errors in the meteorological simulation and chemical transport model interact with each other. More comprehensive analysis should be conducted in the future to better understand the impacts of the meteorological and model errors on the inverse emission inventory. A multi-model inversion framework, for example that of Miyazaki et al. (2020a), may help alleviate the influences of model errors on emission inversions in future. Using other models (e.g., WRF-Chem, CMAQ) to validate our inversion inventory could also help us assess the impacts of model uncertainty on the emission inversions. Meanwhile, because of the many uses that require a rapid update of emissions, it may be time to organize an intercomparison study focused on the emission inversions.

(5) Current inversion emission inventory is mainly assessed by the surface observations and previous emission inventories. more independent observations, such as the satellite observation data, should be used in future to further validate the inversion results of this study and its derived findings. For example, the independent measurements from field campaign or satellite retrievals (e.g., TropOMI CO data) can help validate the reliability of the much higher a posterior CO emissions in CAQIEI than the previous inventories in the future.

**7 data availability**

The CAQIEI inventory can be freely download at https://doi.org/10.57760/sciencedb.13151 (Kong et al., 2023), which includes monthly grid maps of the air pollutant emissions from 2013 to 2020. The contained species include NO$_x$, SO$_2$, CO, primary PM$_{2.5}$, primary PM$_{10}$ and NMVOC. The horizontal resolution is 15km. There are totally 8 Network Common Data Form files (NetCDF), which were named by the date and contains the monthly emissions of different air pollutants in China

in each year. The description of the content of each NetCDF file and some important notes when using this dataset are also
available in README.txt on the website.

**Tables**
**Table 1. Corresponding relationships between the chemical observations and adjusted emissions**

| Species | Description | Observations used for inversions of this species |
|---|---|---|
| BC | Black carbon | $PM_{2.5}$ |
| OC | Organic carbon | $PM_{2.5}$ |
| PMF | Fine-mode unspeciated aerosol | $PM_{2.5}$ |
| PMC | Coarse-mode unspeciated aerosol | $PM_{10} - PM_{2.5}$ |
| $NO_x$ | Nitrogen oxide | $NO_2$ |
| $SO_2$ | Sulfur dioxide | $SO_2$ |
| CO | Carbon monoxide | CO |
| NMVOCs | Non-methane volatile organic compounds | MDA8h $O_3$ |


**Table 2. Evaluation statistics of the *a posteriori* (*a priori*) model simulation for different species[a]**

| | PM$_{2.5}$ (µg/m$^3$) | | | | PM$_{10}$ (µg/m$^3$) | | | |
|---|---|---|---|---|---|---|---|---|
| | R | MBE | NMB (%) | RMSE | R | MBE | NMB (%) | RMSE |
| Hourly | 0.77 (0.53) | 2.1 (13.3) | 4.5 (28.6) | 32.4 (55.6) | 0.72 (0.44) | −3.7 (−11.5) | −4.6 (−14.3) | 53.1 (74.4) |
| Daily | 0.89 (0.61) | 2.1 (13.3) | 4.4 (28.4) | 20.0 (46.3) | 0.88 (0.51) | −3.7 (−11.2) | −4.6 (−14.1) | 31.6 (62.2) |
| Monthly | 0.94 (0.68) | 2.1 (13.3) | 4.5 (28.3) | 11.7 (32.5) | 0.90 (0.56) | −3.6 (−11.3) | −4.5 (−14.1) | 21.2 (44.1) |
| Yearly | 0.94 (0.62) | 2.2 (11.9) | 4.4 (24.3) | 9.1 (27.7) | 0.89 (0.52) | −3.8 (−13.4) | −4.6 (−16.1) | 18.5 (38.7) |

| | SO$_2$ (µg/m$^3$) | | | | NO$_2$ (µg/m$^3$) | | | |
|---|---|---|---|---|---|---|---|---|
| | R | MBE | NMB (%) | RMSE | R | MBE | NMB (%) | RMSE |
| Hourly | 0.64 (0.16) | −1.8 (19.0) | −9.1 (93.8) | 24.9 (58.7) | 0.67 (0.45) | −1.2 (−0.9) | −3.9 (−2.7) | 19.9 (25.5) |
| Daily | 0.80 (0.20) | −1.8 (19.0) | −9.2 (94.5) | 16.0 (51.4) | 0.80 (0.51) | −1.2 (−0.8) | −3.7 (−2.6) | 12.8 (20.1) |
| Monthly | 0.85 (0.20) | −1.9 (18.9) | −9.3 (93.1) | 12.4 (45.8) | 0.84 (0.57) | −1.2 (−0.8) | −3.8 (−2.6) | 9.4 (15.6) |
| Yearly | 0.83 (0.18) | −2.4 (17.0) | −10.8 (75.9) | 11.6 (42.4) | 0.82 (0.63) | −1.3 (−1.6) | −3.9 (−5.0) | 8.1 (13.0) |

| | CO (mg/m$^3$) | | | | O$_3$ (µg/m$^3$) | | | |
|---|---|---|---|---|---|---|---|---|
| | R | MBE | NMB (%) | RMSE | R | MBE | NMB (%) | RMSE |
| Hourly | 0.69 (0.38) | −0.1 (−0.4) | −8.8 (−45.6) | 0.6 (0.8) | 0.71 (0.51) | 5.6 (−8.4) | 9.5 (−14.0) | 34.9 (41.6) |
| Daily | 0.81 (0.42) | −0.1 (−0.4) | −8.6 (−45.5) | 0.4 (0.7) | 0.71 (0.40) | 5.7 (−8.4) | 9.5 (−14.1) | 26.1 (33.8) |
| Monthly | 0.83 (0.42) | −0.1 (−0.4) | −8.7 (−45.7) | 0.3 (0.7) | 0.76 (0.47) | 5.6 (−8.4) | 9.4 (−14.1) | 19.6 (26.0) |
| Yearly | 0.82 (0.27) | −0.1 (−0.5) | −9.0 (−47.6) | 0.3 (0.7) | 0.53 (0.11) | 5.1 (−7.8) | 8.7 (−13.4) | 14.2 (20.5) |

[a] The time series of the air pollutant concentrations at each station were firstly catenated into a single vector. Then the values of each evaluation metric were calculated based on the catenated time series of the observed and simulated concentrations.


**Table 3. Inversion-estimated emissions (Tg/yr) of different species in China as well as the six regions for year 2015**

|  | China | NCP | SE | NE | SW | NW | Central |
|---|---|---|---|---|---|---|---|
| $NO_x$ | 25.2 | 5.1 | 7.1 | 4.5 | 4.2 | 1.2 | 3.2 |
| $SO_2$ | 17.8 | 3.5 | 3.3 | 4.0 | 2.6 | 0.8 | 3.6 |
| CO | 465.4 | 82.2 | 106.7 | 78.7 | 82.8 | 32.6 | 82.3 |
| $PM_{2.5}$ | 14.9 | 2.7 | 3.3 | 3.1 | 2.9 | 1.2 | 1.9 |
| $PM_{10}$ | 40.1 | 8.7 | 7.5 | 8.2 | 5.5 | 4.1 | 6.2 |
| NMVOC | 46.0 | 9.0 | 13.7 | 8.5 | 7.8 | 2.7 | 4.2 |


**Table 4. The calculated annual trends of PM₂.₅ and PM₁₀ emissions in China based on CAQIEI**

|  | $PM_{2.5}$ (Tg/year) | | | $PM_{10}$ (Tg/year) | | |
|---|---|---|---|---|---|---|
|  | 2015–2020 | 2015–2017 | 2018–2020 | 2015–2020 | 2015–2017 | 2018–2020 |
| China | −1.4* | −1.1 | −1.5 | −2.6* | −1.4 | −4.6 |
| NCP | −0.32* | −0.30 | −0.32 | −0.64* | −0.88 | −0.99 |
| SE | −0.32* | −0.21 | −0.44 | −0.52* | −0.48 | −0.84 |
| NE | −0.24* | −0.25 | −0.11 | −0.52* | −0.22 | −0.73 |
| SW | −0.21* | −0.26 | −0.20 | −0.40* | −0.26 | −0.56 |
| NW | −0.09 | −0.08 | −0.12 | −0.20* | −0.32 | −0.32 |
| Central | −0.15 | 0.01 | −0.32 | −0.27 | −0.32 | −1.14 |

* Trend is significant at the 0.05 significance level

**Table 5. The calculated annual trends of the four gaseous emissions in China based on CAQIEI**

| | $SO_2$ (Tg/year) | | | CO (Tg/year) | | |
|---|---|---|---|---|---|---|
| | 2015–2020 | 2015–2017 | 2018–2020 | 2015–2020 | 2015–2017 | 2018–2020 |
| China | −2.1* | −2.1 | −1.3 | −36.0* | −22.8 | −33.5 |
| NCP | −0.57* | −0.69 | −0.21 | −8.4* | −4.30 | −7.23 |
| SE | −0.34* | −0.39 | −0.20 | −6.1* | −3.54 | −8.37 |
| NE | −0.44* | −0.44 | −0.21 | −6.2* | −1.74 | −3.91 |
| SW | −0.22* | −0.27 | −0.17 | −3.8* | −2.36 | −4.54 |
| NW | −0.08* | −0.08 | −0.08 | −3.0* | −0.73 | −2.95 |
| Central | −0.46* | −0.25 | −0.40 | −8.7* | −10.14 | −6.55 |

| | $NO_x$ (Tg/year) | | | NMVOC (Tg/year) | | |
|---|---|---|---|---|---|---|
| | 2015–2020 | 2015–2017 | 2018–2020 | 2015–2020 | 2015–2017 | 2018–2020 |
| China | −0.67 | 0.74 | −1.6 | 1.9 | 6.3 | −1.3 |
| NCP | −0.32 | 0.05 | −0.40 | 0.66 | 1.37 | −0.42 |
| SE | −0.22 | 0.18 | −0.49 | 0.50 | 1.73 | −0.24 |
| NE | −0.17 | 0.03 | −0.19 | 0.03 | 0.79 | −0.49 |
| SW | −0.06 | 0.10 | −0.26 | 0.23* | 0.43 | 0.03 |
| NW | −0.03 | 0.11 | −0.06 | 0.10 | 0.69 | −0.27 |
| Central | 0.04 | 0.28 | −0.16 | 0.55* | 1.33 | 0.09 |

* Trend is significant at the 0.05 significance level























**Table 6 The top-down estimated CO emissions in China from previous inventories**

| Reference | Region | Period | Method | Assimilated observation | *A priori* CO emission (kt/day) | *A posteriori* CO emission (kt/day) |
|---|---|---|---|---|---|---|
| Feng et al. (2020) | China Mainland | December 2013 | EnKF with CMAQ model | Surface observation | 586.4 | 1678.0 |
| | | December 2017 | | | 499.3 | 1388.1 |
| | NCP | December 2013 | | | 143.9 | 394.3 |
| | | December 2017 | | | 120.5 | 340.7 |
| Muller et al. (2018) | China | 2013 | 4DVar with IMAGES model | IASI CO observation with different constraints on OH levels | 454.8 | 367.1–553.4 |
| Gaubert et al. (2020) | Central China | May 2016 | DART/CAM-CHEM | MOPITT CO observation | 193.6 | 220.3 |
| | North China | | | | 93.5 | 163.6 |
| Jiang et al. (2017) | East China | 2013 | 4DVar with GEOS-Chem | MOPITT CO observation | 564.5 | 439.5–484.4 |
| | | 2014 | | | | 430.4–481.1 |
| | | 2015 | | | | 397.5–439.7 |
| Zheng et al. (2019) | China | 2010–2017 average | Bayesian inversion | MOPITT CO, OMI HCHO, and GOSAT $CH_4$ observation | - | 444.4 |


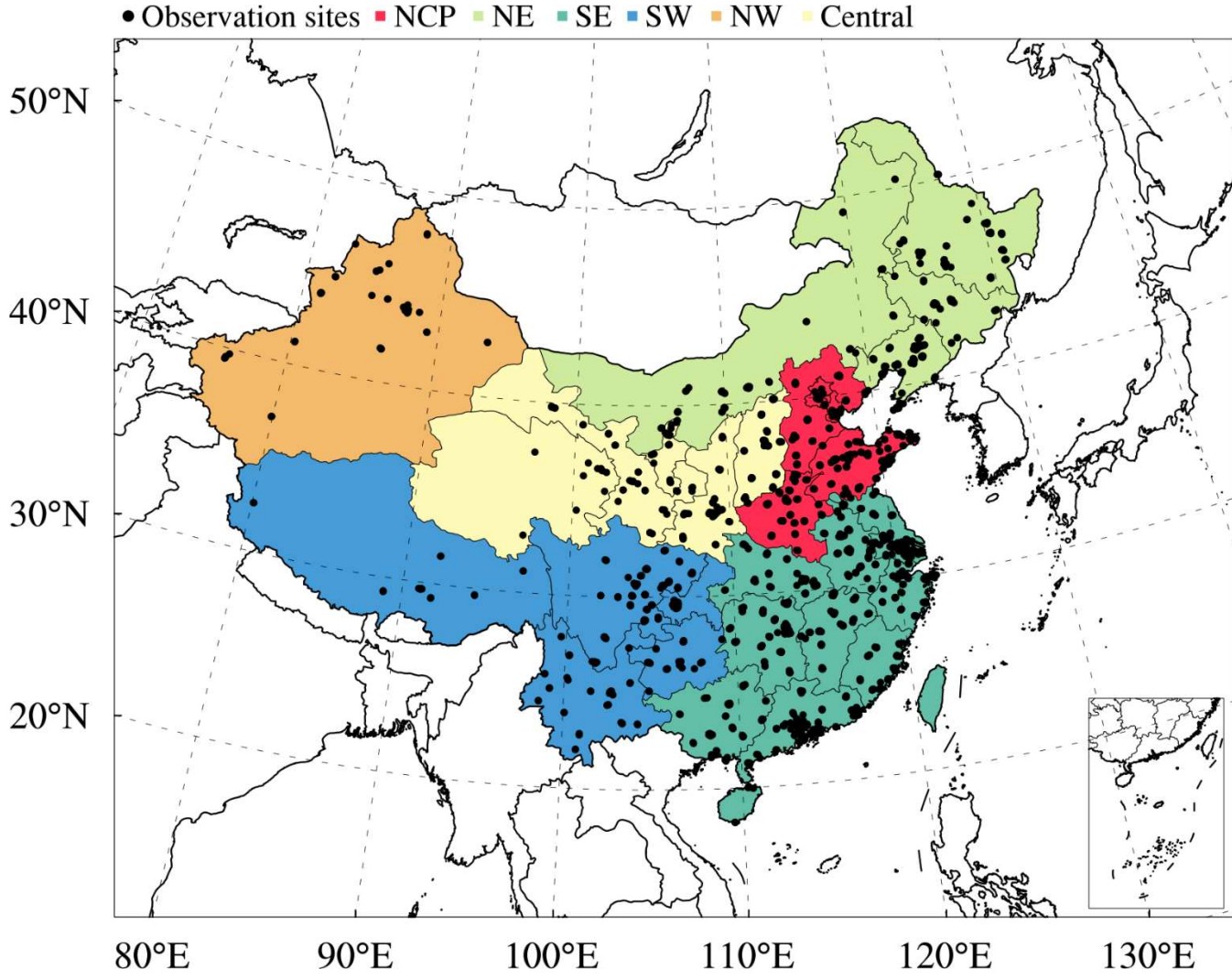


Figure 1: Modeling domain of the ensemble simulation overlaid with the distributions of observation sites from CNEMC. Different
colors denote the different regions in mainland China—namely, the North China Plain (NCP), Northeast China (NE), Southwest
China (SW), Southeast China (SE), Northwest China (NW) and Central China (Central).

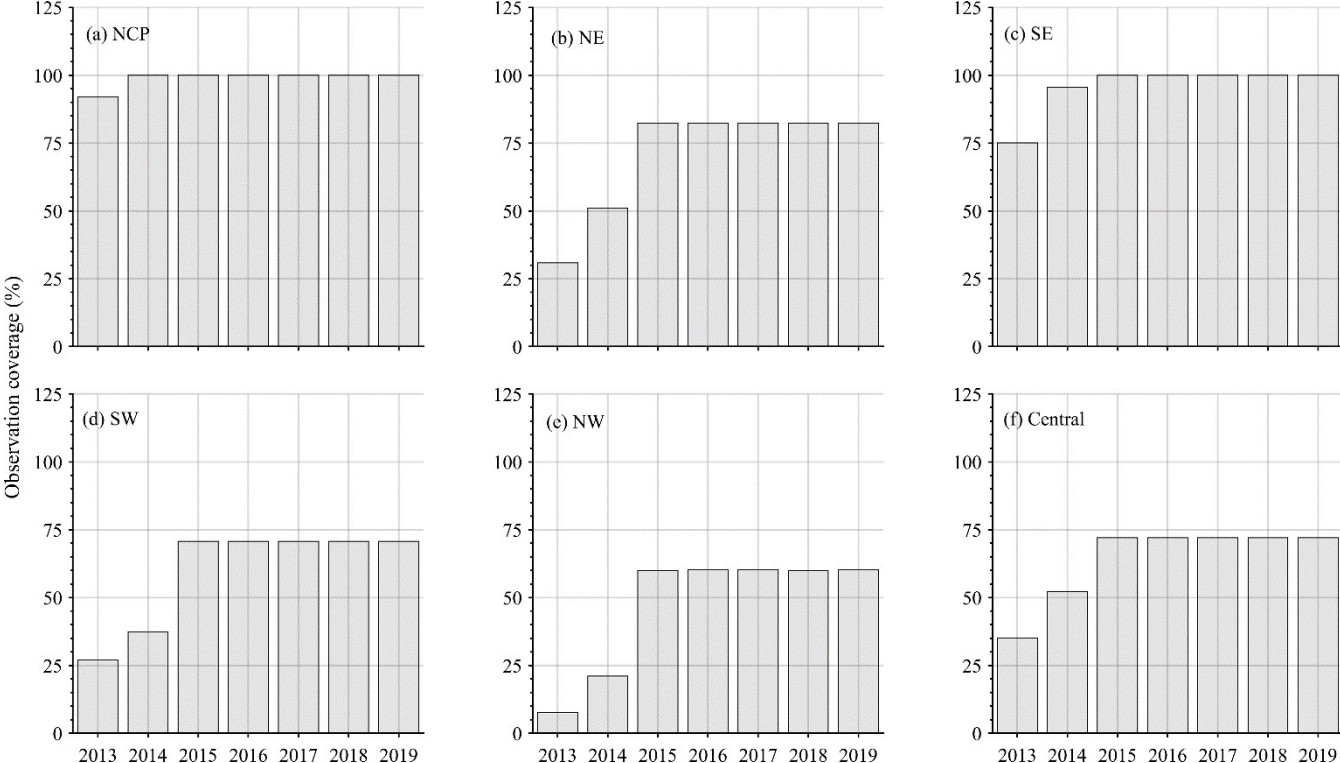

**Figure 2: Time series of the observational coverage from 2013 to 2020 over different regions of China.**

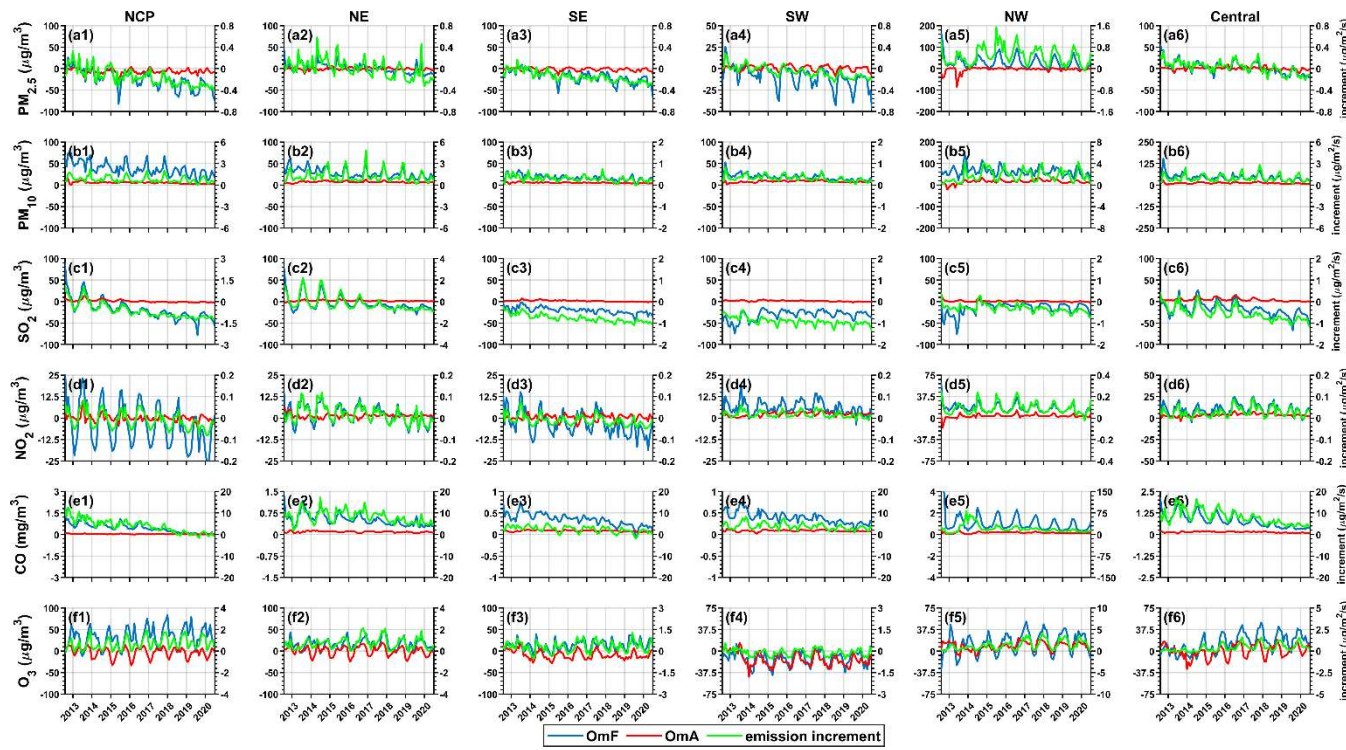

**Figure 3: Time series of the *a priori* bias (blue lines), the *a posteriori* bias (red lines), and the emission increment (green lines) from 2013 to 2020 for different species over the six regions of China.**

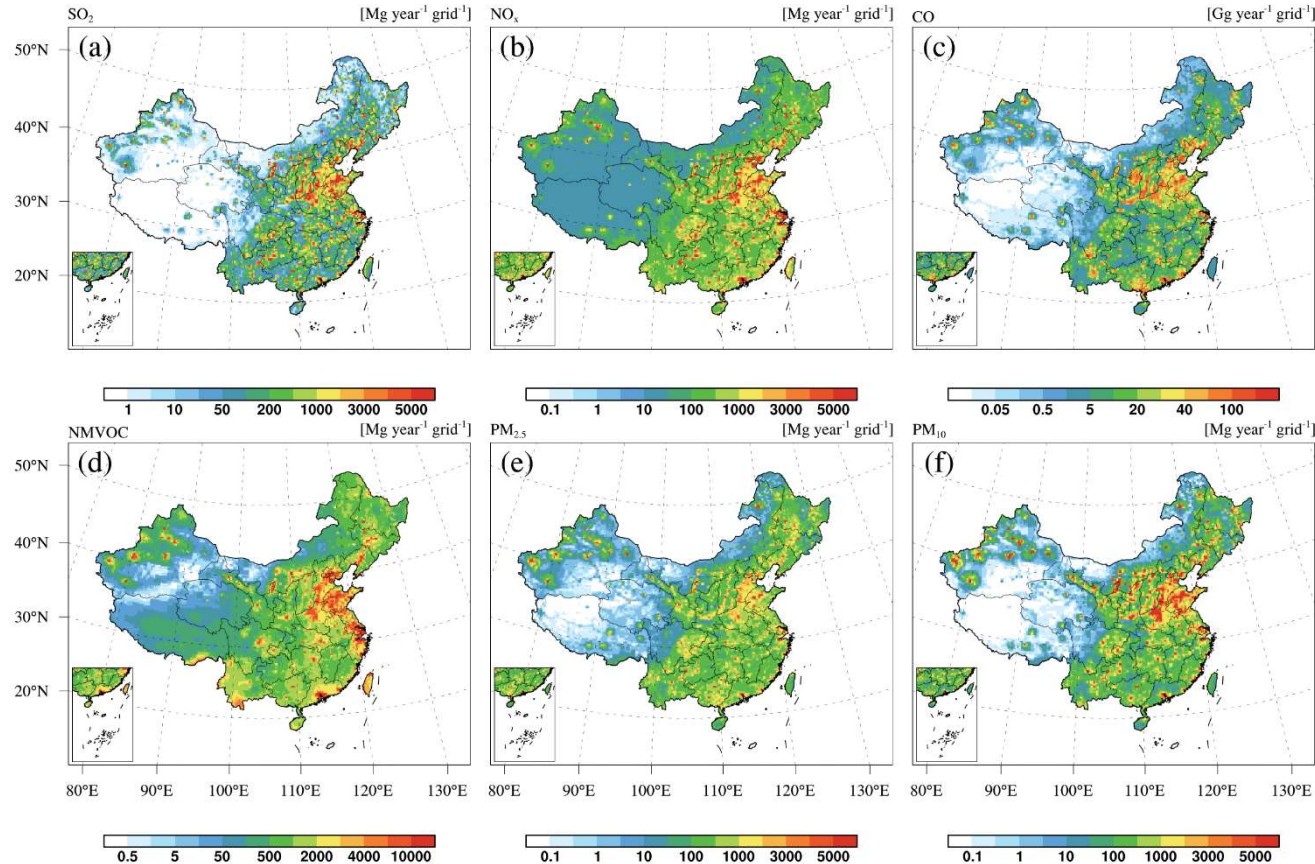

**Figure 4: Spatial distributions of the emissions of (a) SO₂, (b) NOₓ, (c) CO, (d) NMVOCs, (e) PM₂.₅, and (f) PM₁₀ in 2015 obtained from CAQIEI.**




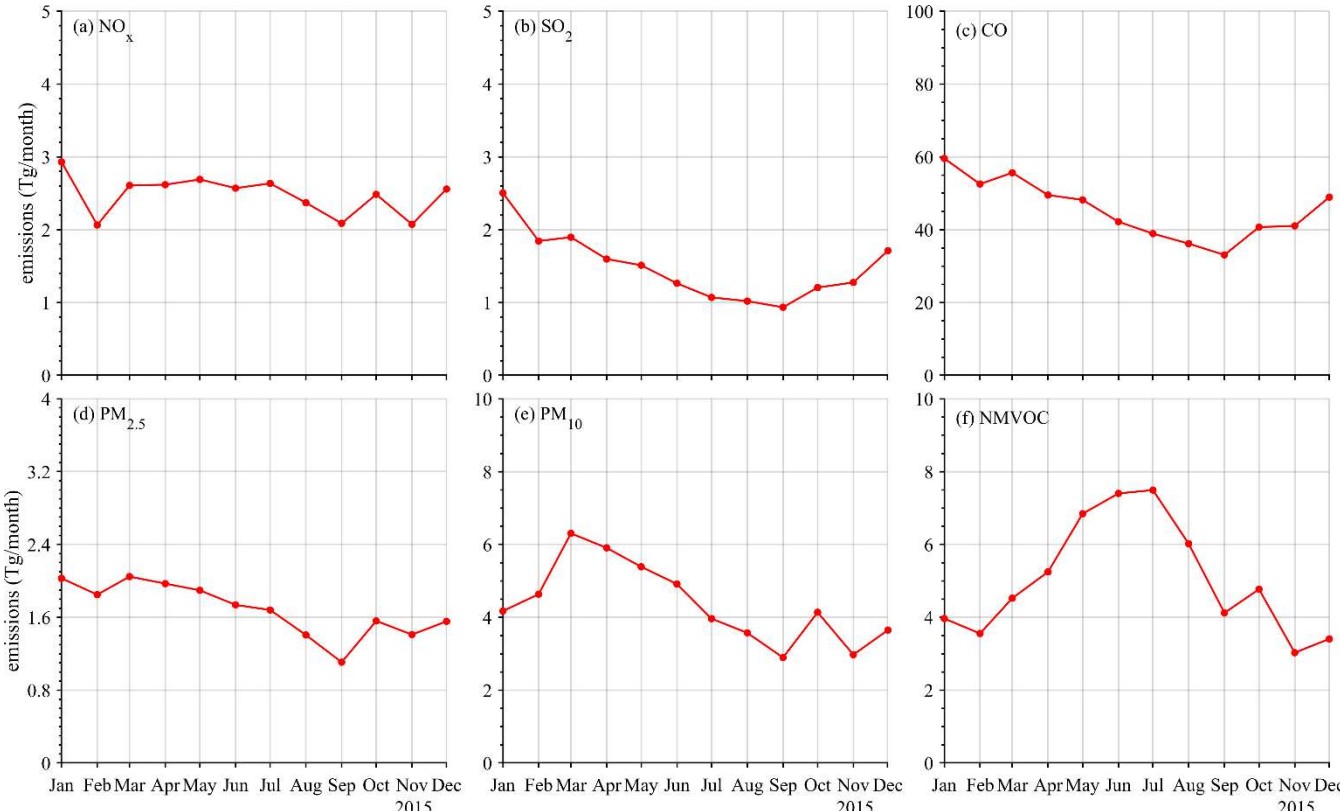

**Figure 5: Monthly series of total emissions of (a) NO$_x$, (b) SO$_2$, (c) CO, (d) PM$_{2.5}$, (e) PM$_{10}$, and (f) NMVOCs in China for year 2015**
**obtained from CAQIEI.**















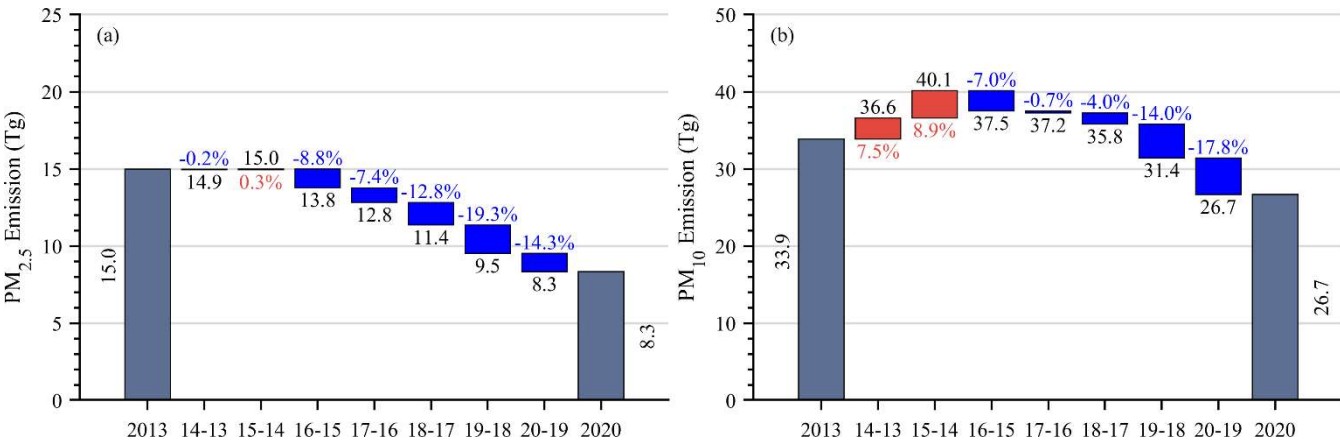

Figure 6: Emission changes in (a) PM2.5 and (b) PM10 obtained from CAQIEI from 2013 to 2020.

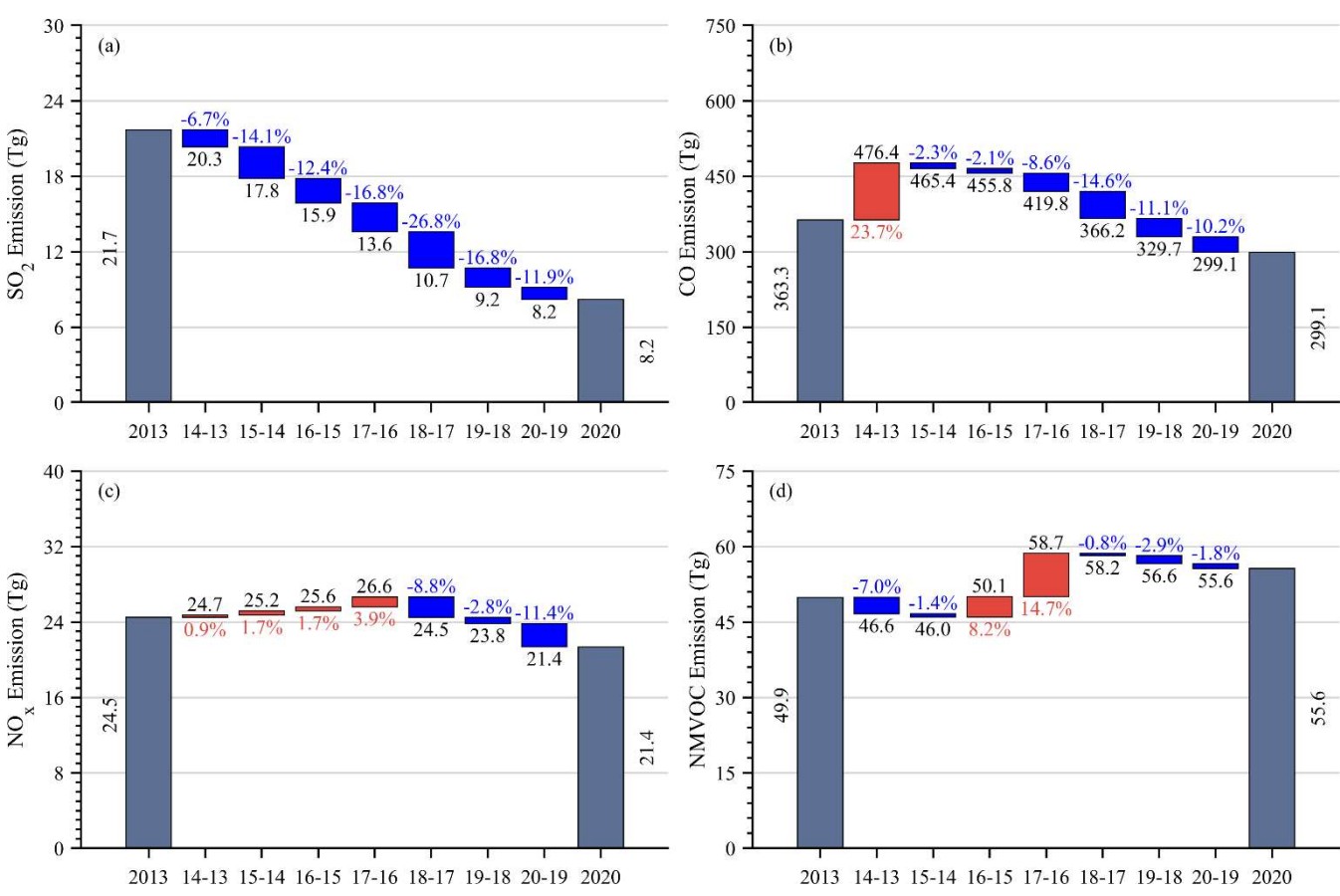

Figure 7: Emission changes in (a) SO2, (b) CO, (c) NOx, and (d) NMVOCs obtained from CAQIEI from 2013 to 2020.

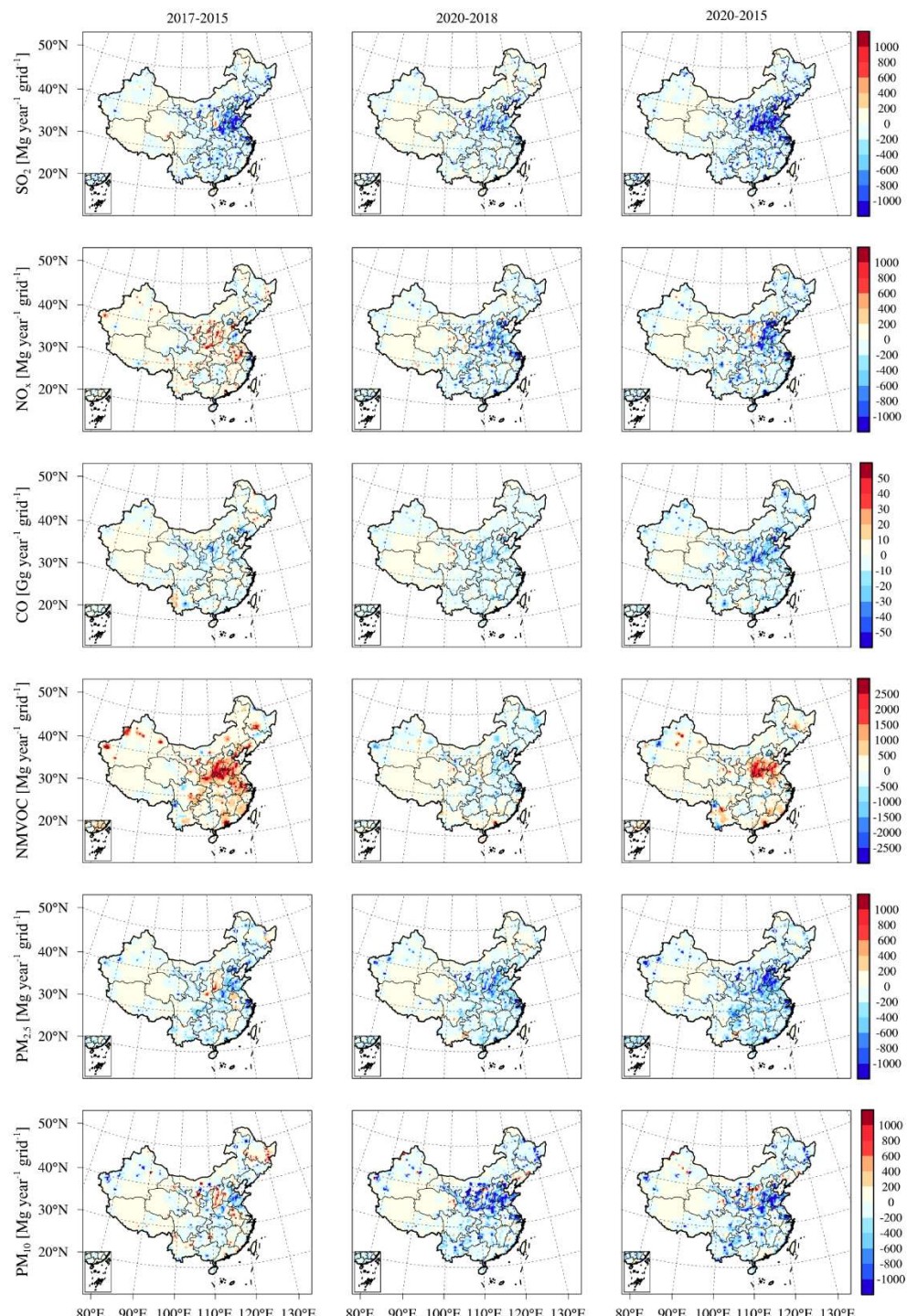


Figure 8: Spatial distributions of the emission changes of different species during 2015–2017 (left panels), 2018–2020 (middle panels), and 2015–2020 (right panels) obtained from CAQIEI from 2013 to 2020.

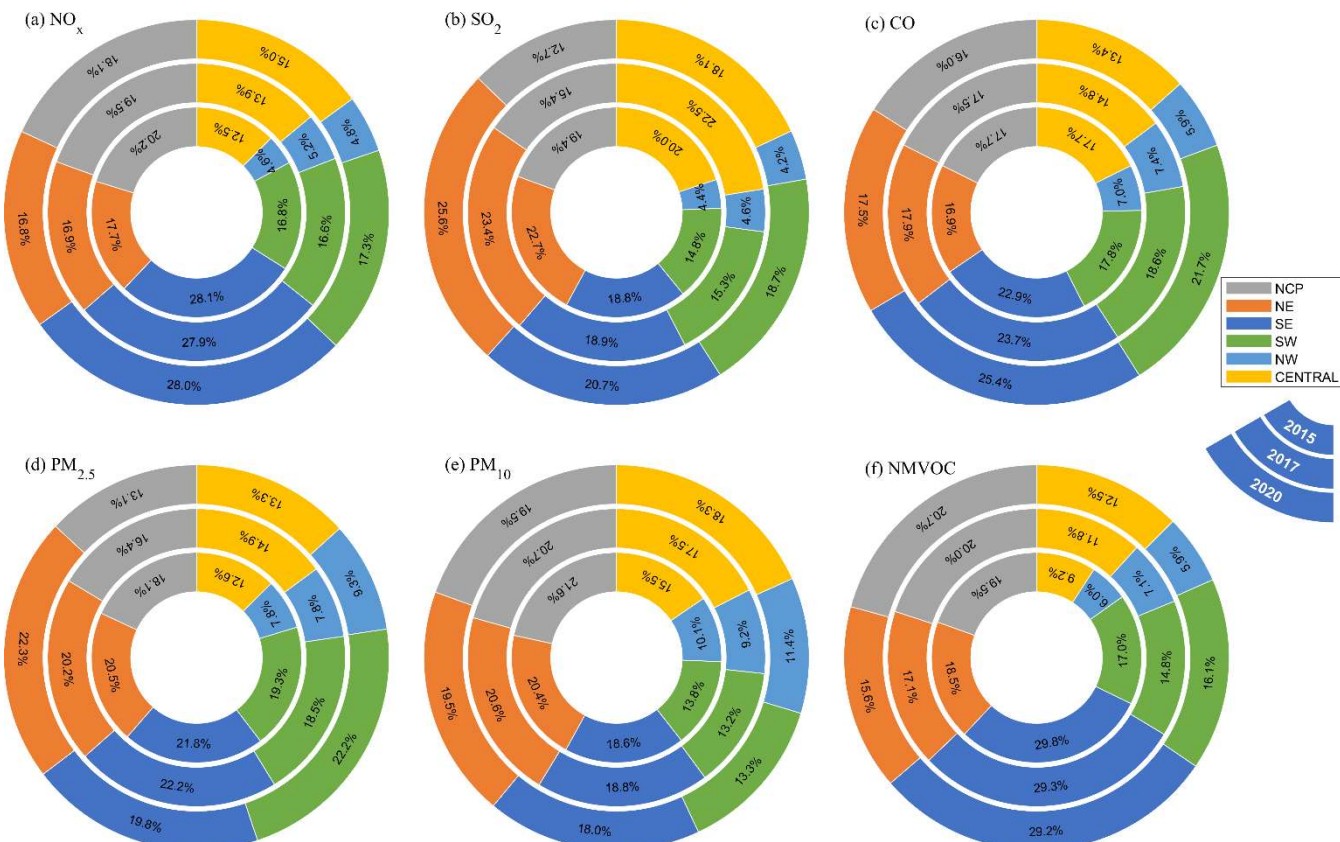

Figure 9: Emission distributions of (a) NOₓ, (b) SO₂, (c) CO, (d) PM2.5, € PM₁₀, and (f) NMVOCs among different regions in China
obtained from CAQIEI in 2015, 2017 and 2020.


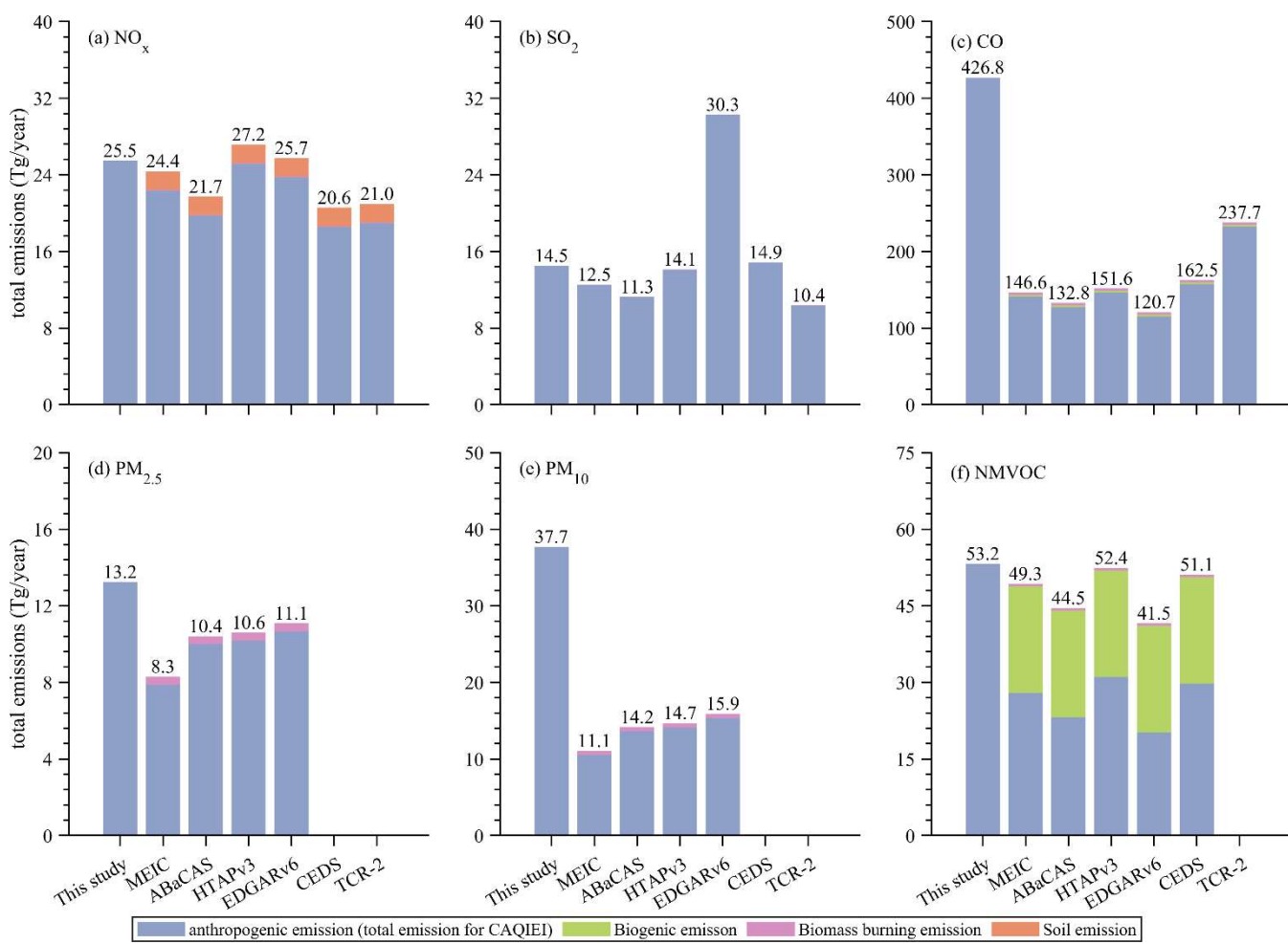


Figure 10: Comparisons of the averaged emissions of (a) NO$_x$, (b) SO$_2$, (c) CO, (d) PM$_{2.5}$, (e) PM$_{10}$, and (f) NMVOCs over China from 2015 to 2018 between CAQIEI and previous inventories added with natural sources.



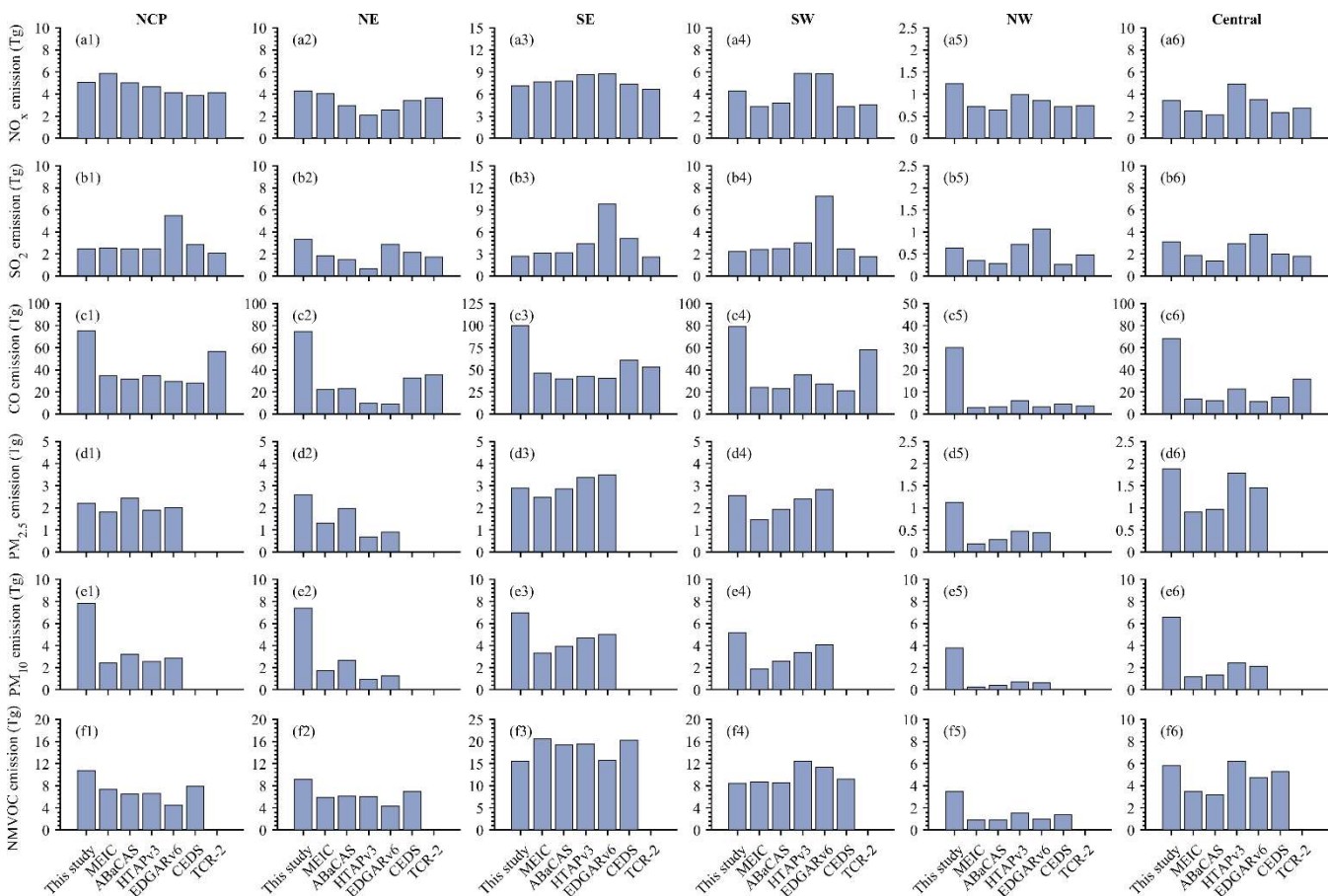


Figure 11: Comparisons of the averaged emissions of (a) NOx, (b) SO2, (c) CO, (d) PM2.5, (e) PM10, and (f) NMVOCs over different regions in China from 2015 to 2018 between CAQIEI and previous inventories added with natural sources.







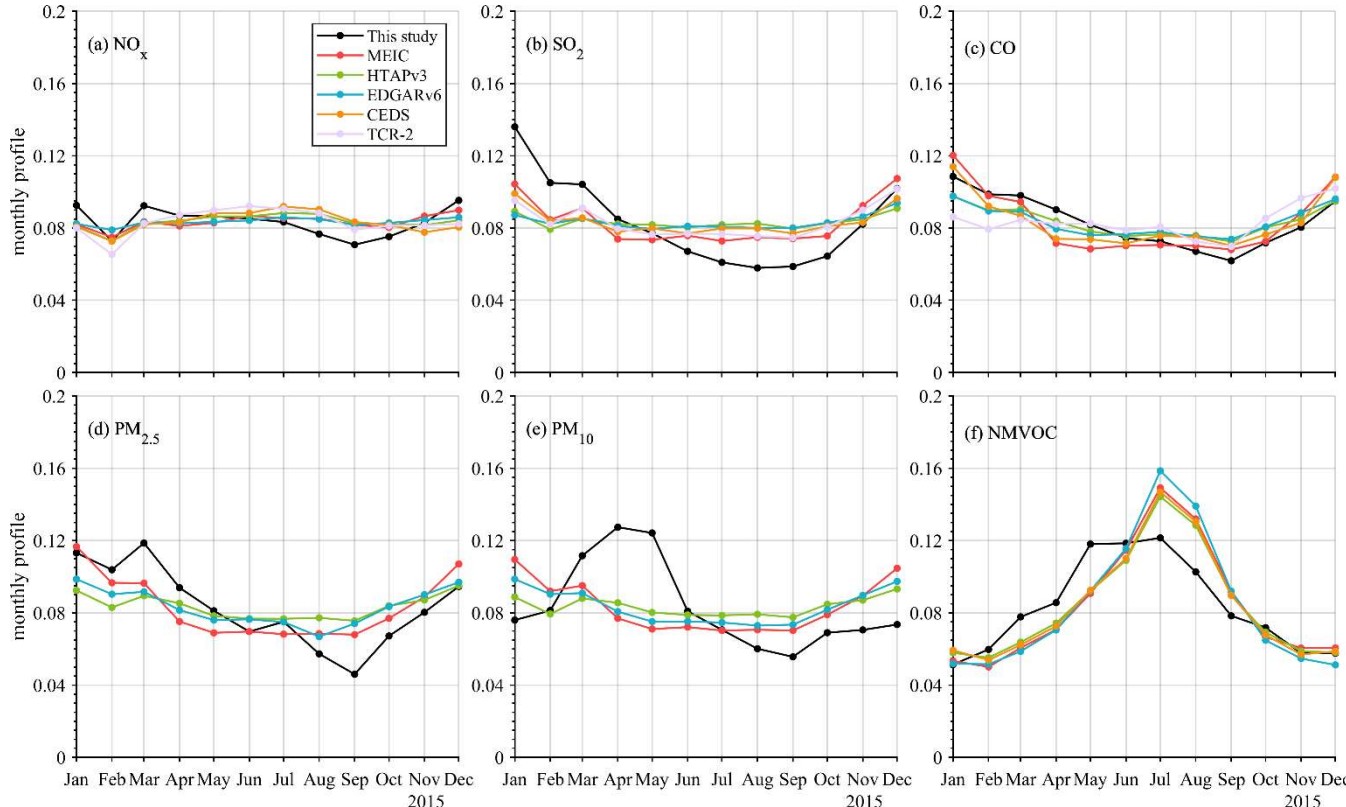

Figure 12: Comparisons of the monthly profiles of (a) NOₓ, (b) SO₂, (c) CO, (d) PM₂.₅, (e) PM₁₀, and (f) NMVOCs over China averaged from 2015 to 2018 between CAQIEI and previous inventories added with natural sources.

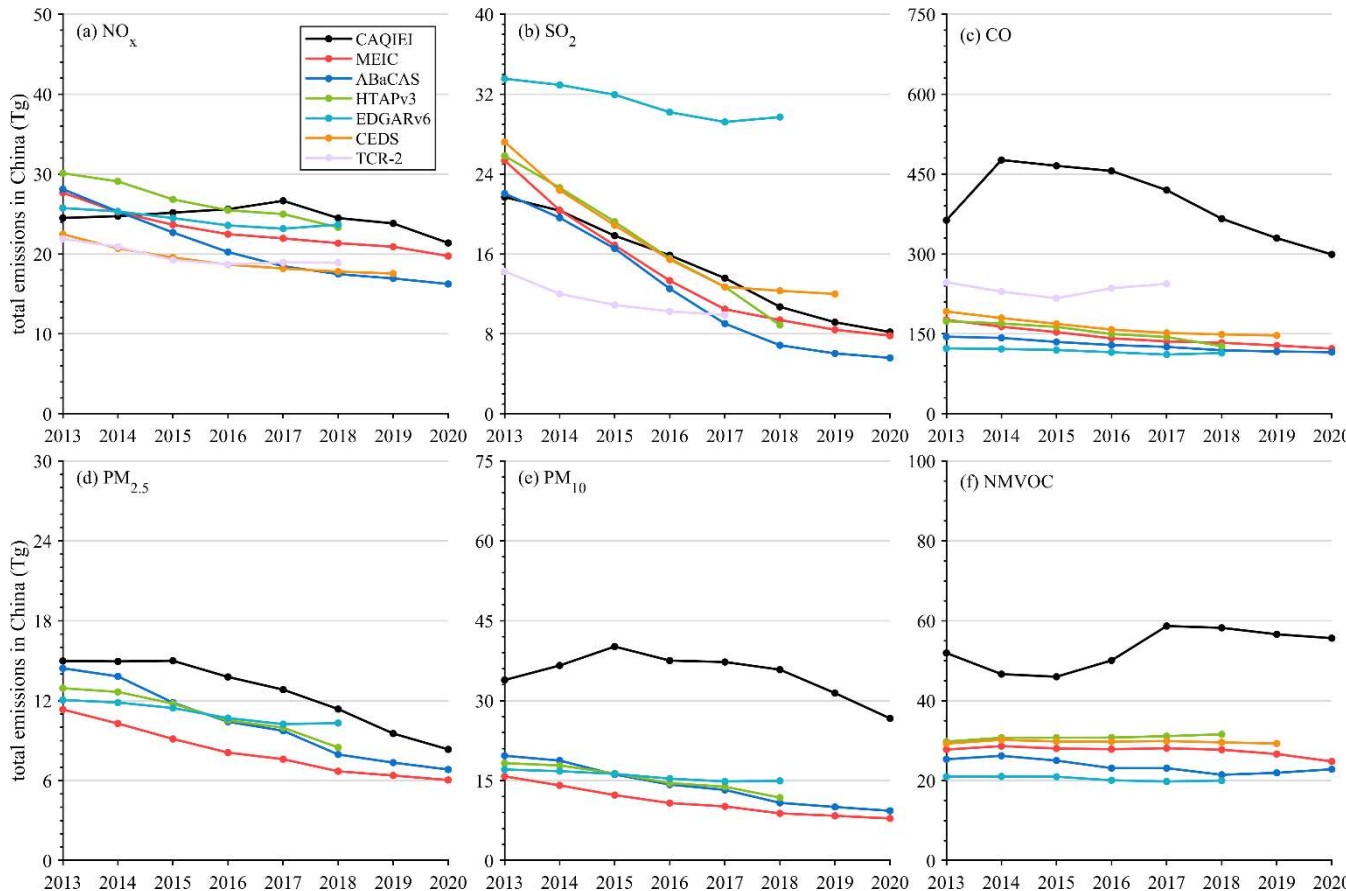


**Figure 13: Time series of annual emissions of (a) NO$_x$, (b) SO$_2$, (c) CO, (d) PM$_{2.5}$, (e) PM$_{10}$ and (f) NMVOC over China from 2013 to**
**2020 obtained from CAQIEI and previous inventories. Note that the natural sources were not included in the previous inventories**
**in this figure.**






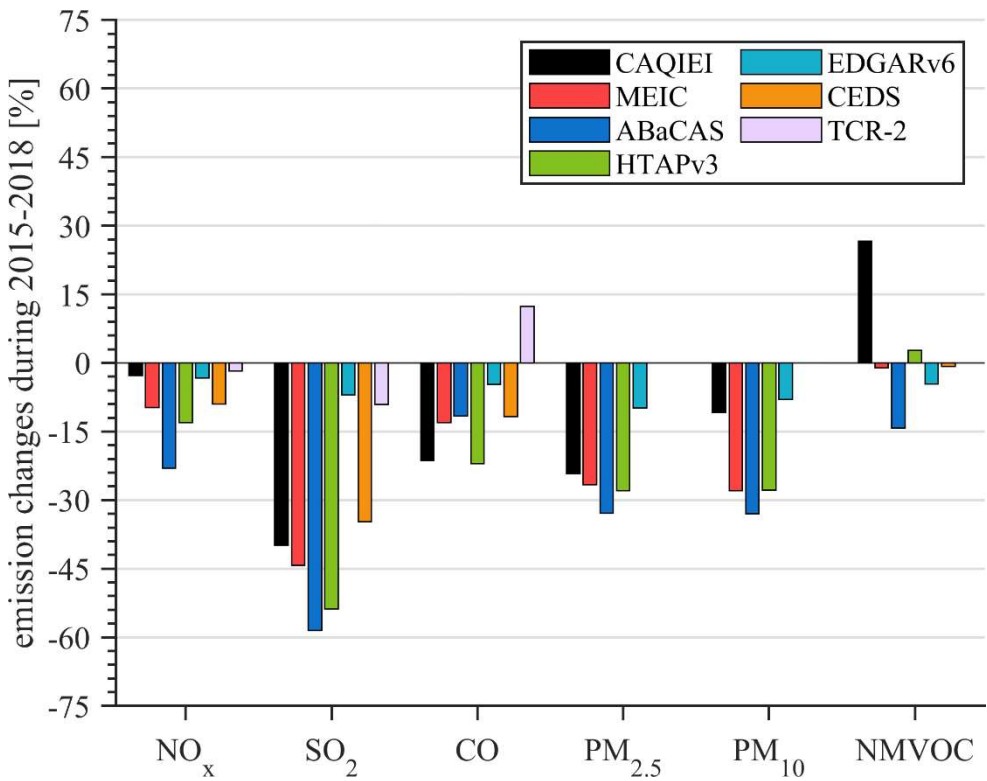


**Figure 14: Comparisons of the calculated emission changes of NOₓ, SO₂, CO, PM₂.₅, PM₁₀, and NMVOCs over China from 2015 to**
**2018 between CAQIEI and previous inventories. Note that the natural sources were not included in the calculation of the emission**
**changes in this figure.**
**Author contributions**
X.T., Z.W., and J.Z. conceived and designed the project; L.K., H.W., X.T., and L.W. established the data assimilation system;
Q.W. and L.K. performed the meteorology simulations; L.K., H.C., and J.L. conducted the ensemble simulation with the
NAQPMS model; J.L., L.Z., W.W., B.L., Q.W., D.C. and Y.P. provided the air quality monitoring data; H.W. performed the
quality control of the observation data; and L.K. performed the inversion estimation, generated the figures, and wrote the paper,
with comments provided by G.R.C.
**Competing interests**
The authors declare no competing financial interest.
**Acknowledgements**
We acknowledge the use of surface air quality observation data from CNEMC and the strong support from the National Key
Scientific and Technological Infrastructure project "Earth System Science Numerical Simulator Facility" (EarthLab), which
provide us with ample computational resources to fill the requirement of the inversion of multiple years using the ensemble
method at a high grid resolution of 15km.
**Financial support**
This research has been sponsored by the National Natural Science Foundation of China (Grant Nos. 42175132, 92044303,
42205119), the National Key R&D Program (Grant No. 2020YFA0607802), the CAS Information Technology Program (Grant
No. CAS-WX2021SF-0107-02).

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
