# Peer review of "Changes of air pollutant emissions in China during two clean air action periods derived from the newly developed Inversed Emission"

_Earth System Science Data, 2023_

## Referee Comment (RC2)

**General comments:**

The manuscript presents a comprehensive study aimed at estimating air pollutant emissions in China through the assimilation of surface observations. The authors found that the emission reduction efforts during the 2018-2020 Action Plan generally exceeded those of the 2013-2017 Action Plan. They also conducted comparisons with various bottom-up emission inventories, and provided detailed explanations for differences and uncertainties. These findings are relevant and potentially important.

However, while reading the manuscript, I encountered several unanswered questions, mainly related to the settings and parameters of the estimation technique, as well as potential uncertainties and biases in the inferred emission estimates. In particular, I have doubts about the credibility of the NMVOC emission inversion. I believe that further analysis and discussion addressing the major and specific issues outlined below are necessary to substantiate the authors' claims and make the manuscript suitable for publication in ESSD.

**Major comments:**

1. The authors compare the posterior results with other sources and frequently employ terms like "underestimate" and "overestimate" without explicitly specifying what is considered an under- or overestimate relative to a reference. For instance, in line 47, the use of these terms lacks clarity. More critically, the terms "underestimate" and "overestimate" imply that the posterior is inherently closer to the truth than the other sources, assuming that the other sources are less accurate. This assumption is not self-evident. In inversion, adjusting emissions to match observations does not conclusively prove that the posterior emissions are improved, nor does it inherently indicate biases in other bottom-up inventories.

To claim that HTAP and other sources are less accurate and to justify the terms "overestimate" and "underestimate," the authors need to provide a more convincing argument. Simply relying on posterior simulations is not sufficient to demonstrate the improvement in posterior emissions and the existence of biases in other bottom-up inventories. If a more convincing argument cannot be made, the authors should consider using more neutral terms to avoid implying a hierarchy of accuracy among different emission sources.

An example of inconsistency can be found in Section 4.3, where the authors, in comparing their emission inventory with others, occasionally use alternative

inventories as a basis to highlight the agreement and reduced uncertainty of their inventory compared to bottom-up inventories. At other times, however, they claim that these alternative inventories exhibit significant uncertainties. This contradiction raises concerns about the clarity and consistency of the manuscript. It is essential that the authors provide a more coherent explanation or rationale for the varying assessments of uncertainty in other inventories.

2. In particular, the authors' comparison of natural and anthropogenic species emissions (such as PM10 and NMVOC) reveals a significant issue. Natural sources inherently exhibit considerable uncertainty, and in many regions, natural sources contribute significantly more than anthropogenic sources. Therefore, using the uncertainty in natural sources as a basis does not necessarily indicate large uncertainties in anthropogenic sources within the bottom-up inventories. An inconsistency arises in Line 790, where the authors' explanation appears contradictory. They simultaneously assume minimal variations in natural sources and cite literature indicating an increasing trend in natural sources. Additionally, the manuscript attributes emission changes to anthropogenic sources while acknowledging substantial uncertainty in natural sources. If it is acknowledged that natural sources indeed carry significant uncertainty (which is indeed the case), the manuscript should avoid using terms such as "not captured," "overestimated," or "underestimated" concerning the bottom-up inventories. These terms imply a clear attribution of error that may not be justified given the uncertainties associated with natural sources. Clear and consistent handling of uncertainties in both natural and anthropogenic sources is crucial for maintaining the credibility of the manuscript.

3. In Line 272, it is mentioned that VOC emissions are optimized through assimilating ground-level O3 observations. However, several factors need consideration. On one hand, VOC-O3 interactions involve strong nonlinear chemical reactions, and emission adjustments exhibit bidirectionality (Tang et al., 2016). Despite the convergence of simulations and observations, VOC inversion results may deteriorate due to these complexities. On the other hand, the majority of the national monitoring stations are situated in urban areas, whereas VOC primarily originates from suburban or rural regions. I am skeptical about the feasibility of assimilating O3 to constrain VOC emissions. As evident from Figure 3, the posterior simulations do not show a significant improvement in O3. As the authors noted, O3 cannot effectively constrain precursor NOx (L278). Therefore, I recommend deleting the VOC emission inversion.

4. The changes in observation coverage each year can significantly impact emission

estimates. If the authors intend to include the years 2013-2014 in this study, they should compare the impact of site differences on emissions for a more robust analysis. If the authors aim to investigate trends, it is advisable to delete emissions in the 2013-2014 period, as this might otherwise potentially mislead readers, given that the changes during this period do not contribute meaningfully to the study's overall trend analysis.

There also appears to be some discrepancies in the manuscript where emission changes are often stated as occurring from 2015-2017, while the text descriptions indicate the period as 2013-2017, as seen in lines 452, 485, and 561, among others. Furthermore, it is important to note that the changes in emissions observed from 2015-2017 not necessarily reflect the overall reduction rate of the action plan for the entire period of 2013-2017. Additionally, the data from 2015-2017 alone may not be sufficient to conclude that the emission reduction rate during the 2013-2017 period is lower than that during the 2018-2020 action plan.

5. The authors use PM2.5 observations to simultaneously constrain BC, OC, and primary PM2.5. If they do not consider inter-species correlations or use random perturbations, and, for instance, if BC and OC increase while PM2.5 decreases in one ensemble member. How do they constrain emissions when the simulated PM2.5 and observations are the same.

6. The authors simultaneously constrain concentrations and emissions, emphasizing that concentration errors arise from emission uncertainties, implying a shared source of uncertainty (L222). In this context, the question arises whether optimizing concentrations would diminish emission uncertainties, thereby affecting emission estimates.

7. The NOx emission changes optimized by the authors appear to contradict existing research findings and are inconsistent with recent emission reduction policies. Despite citing the study by Zheng et al., (2018), the actual NOx emissions reported by Zheng show a significant decrease. Could this discrepancy be attributed to the bottom-up inventory lacking sufficient statistics on mobile vehicle emissions? Moreover, according to Zheng's study, industrial and power plant emissions collectively contribute to over 50% of total emissions. Hence, the second reason provided by the authors may not be suitable if the industrial and power plant emissions are substantial contributors

8. "Figure 12 shows significant discrepancies between the results of the author's inversion and other bottom-up inventories. According to other literature, it is known that China experienced two peaks in VOC emissions in May and July 2015. The

variation in VOC emissions closely follows the changes in O3 levels, suggesting a strong dependence of VOC on O3 variations. This raises the question of whether non-linear changes are being overlooked.

9. The author has provided a high-resolution, multispecies emission inventory. To facilitate users' understanding of the data's accuracy, could you please provide information on the uncertainties associated with different species, allowing users to assess the error range in the data?

**Specific comments**

1.    Change "Fengwei Plain" to "Fenwei Plain"

2.    In L212, the VOC adjustment factor was omitted.

3.    Since MOZART data products are no longer updated, are the boundary conditions in this study based on simulations conducted by the author's team? Additionally, maritime shipping emissions have a significant impact on the generation of NO2 and O3 in coastal provinces. Has the model taken into account inputs from maritime emissions? Why was a localized scale of 180 km chosen?

4.    In L291, is it necessary to reassemble simulations for each iteration, or is it multiple inversions on the original ensemble? If multiple iterations are performed, does it imply that the posterior approaches the observations more closely with each iteration? Why was the choice made to iterate twice?

5.    With such a high grid resolution of 15 km, how does the computational cost for the inversion of multiple years in the ensemble calculations? Additionally, what is the size of the assimilation window?

6.    The inflation factor 'r' varies for each window. Is 'r' a matrix or a scalar? If it is a scalar, could the author provide the specific range of 'r'?

7.    Table 3 lacks information regarding the year.

8.    How were diurnal variations of the emissions specified?

9.    How is the optimization of VOC components conducted when VOC consists of multiple components?

10.   Region name in Figure 1 refers to specific areas. Consider a different expression to avoid potential ambiguity.

11.   Please consider adopting a clearer representation for Figure 11.

12. L483 Change "Fig.3" to "Fig. 4"

---

## Author Comment (AC1)

**Response to Reviewer #1 (ESSD-2023-477)**

We Thank Reviewer for his/her constructive comments

Responses to the Specific comments

**General comments:** This is a valuable high-resolution emission dataset for China, and the paper is well-structured. However, I have some questions and suggestions:

**Reply:** The authors appreciate the reviewer for his/her constructive and up-to-point comments. We have carefully considered the comments and revised the manuscript accordingly. Please refer to our responses for more details given below.

**Comment 1:** In Figure 4, compared to other pollutants, NOx and VOC emissions still show distributions in the Tibet region of China. Could you provide more information about the sources of NOx and VOC in this region?

**Reply:** Thanks for this suggestion. Besides the anthropogenic emissions of $NO_x$ and VOC which are mainly located over the urban areas of the Tibet region of China like the other pollutants, there are also distributions of natural sources distributed over there, for example, the soil $NO_x$ emissions and the biogenic NMVOC emissions. Thus, the differences in the distributions of the emissions of $NO_x$, and VOC compared to other pollutants over the Tibet region of China could be mainly attributed to the natural sources of these two species, considering that the contributions of natural sources to the other pollutants are much smaller. Following the suggestions of the reviewer, we have added more information about the sources of $NO_x$ and VOC in this region in the revised manuscript (please see lines 438–440)

**Comment 2:** Could you display the temporal trends in emissions from 2013 to 2020 for China and its sub-regions, comparing different inventories (multiple lines) to better illustrate the changes in emission patterns over time?

**Reply:** Thanks for this suggestion. we have added the temporal trends in the emissions of different air pollutants in China (Fig. R1) and its sub-regions (Fig. R2–R7) obtained from our inversion results and other emission inventories in the revised manuscript to better illustrate the changes in emission patterns over time. Please see Fig. 13 in the revised manuscript and Fig. S10–S15 in the revised supplement.

[Figure]

Figure R1: Time series of annual emissions of (a) NO$_x$, (b) SO$_2$, (c) CO, (d) PM$_{2.5}$, (e) PM$_{10}$ and (f) NMVOC over China from 2013 to 2020 obtained from CAQIEI and previous inventories. Note that the natural sources were not included in the previous inventories in this figure.

[Figure]

Figure R2: Time series of annual NOx emissions over of different regions of China: (a) NCP, (b) NE, (c) SE, (d) SW, (e) NW and (f) Central from 2013 to 2020 obtained from CAQIEI and previous inventories. Note that the natural sources were not included in the previous inventories in this figure.

[Figure]

Figure R3: Same as Fig. R2 but for SO₂.

[Figure]

Figure R4: Same as Fig. R2 but for CO.

[Figure]

Figure R5: Same as Fig. R4 but for PM$_{2.5}$.

[Figure]

Figure R6: Same as Fig. R5 but for PM$_{10}$

[Figure]

Figure R7: Same as Fig. R2 but for NMVOC

**Comment 3**: As a high-resolution grid product, is it possible to include a comparison at the grid scale with other inventories?

**Reply:** Thanks for this comment. We have added the comparisons of the inversion results with the other emission inventory at the grid scale in the revised manuscript by drawing the spatial distributions of the emissions of different pollutants obtained from different emission inventory (Fig. R8). Please see Fig. S9 in the revised supplement.

[Figure]

Figure R8: Spatial distributions of the averaged emissions of different air pollutants in China during 2015–2018 obtained from CAQIEI, MEIC, HTAPv3, EDGARv6, CEDS and TCR-2. Note the due to absence of gridded products of the ABaCAS inventory, we did not provide its spatial distributions. Also, the natural sources were not added to the previous emission inventories in this figure because of the different spatial resolutions among these inventories.

**Comment 4:** In Figure 12, the profiles presented in this study differ from previous research, especially for SO2 and PM10. Are the monthly simulation results for $SO_2$ and $PM_{10}$ superior to the simulation results using other inventories?

**Reply:** Thanks for this comment. Since the monthly profiles of the a priori SO₂ and PM₁₀ emissions are very similar to those of the other inventories (Fig. R9), the comparisons of the a priori and a posterior simulation could be used to investigate whether the monthly simulation results for SO₂ and PM₁₀ are superior to the simulation results using other inventories. Figure R10 and Figure R11 then show the comparison of the a priori and a posterior monthly simulation of SO₂ and PM₁₀ over different regions of China. It can be clearly seen that the performance of monthly simulations of SO₂ and PM₁₀ are improved significantly by using the a posteriori simulation, suggesting that the monthly simulation results of SO₂ and PM₁₀ would be superior to the simulation results using other inventories.

[Figure]

Figure R9: Comparisons of the monthly profiles of the a priori SO₂ and PM₁₀ emissions with previous emission inventories.

[Figure]

Figure R10: comparison of the a priori and a posterior monthly simulation of $PM_{10}$ concentration over different regions of China.

[Figure]

Figure R11: comparison of the a priori and a posterior monthly simulation of $SO_2$ concentration over different regions of China.

**Comment 5**: Additionally, it would be interesting to see further validation results from more models (e.g., WRF-CMAQ) using this dataset as input in the future.

**Reply:** Thanks for this suggestion. Using other models to validate the inversion inventory is a worthwhile

endeavor because the results can better validate our inversion inventory and also provide us with interesting information about the impacts of model uncertainty on the emission inversions. As suggested by the reviewer, further validation results by using the other models would be analyzed and provided in the future which has been mentioned in the revised manuscript (please see lines 1015–1016).

---

## Author Comment (AC2)

**Response to Reviewer #2 (ESSD-2023-477)**

We Thank Reviewer for his/her constructive comments

Responses to the Specific comments

**General comments:** The manuscript presents a comprehensive study aimed at estimating air pollutant emissions in China through the assimilation of surface observations. The authors found that the emission reduction efforts during the 2018-2020 Action Plan generally exceeded those of the 2013-2017 Action Plan. They also conducted comparisons with various bottom-up emission inventories, and provided detailed explanations for differences and uncertainties. These findings are relevant and potentially important. However, while reading the manuscript, I encountered several unanswered questions, mainly related to the settings and parameters of the estimation technique, as well as potential uncertainties and biases in the inferred emission estimates. In particular, I have doubts about the credibility of the NMVOC emission inversion. I believe that further analysis and discussion addressing the major and specific issues outlined below are necessary to substantiate the authors' claims and make the manuscript suitable for publication in ESSD:

**Reply:** The authors appreciate the reviewer for his/her constructive and insightful comments. We have carefully considered the comments and revised the manuscript accordingly. Please refer to our responses for more details given below.

**Major comments:**

**Comment 1:** The authors compare the posterior results with other sources and frequently employ terms like "underestimate" and "overestimate" without explicitly specifying what is considered an under- or overestimate relative to a reference. For instance, in line 47, the use of these terms lacks clarity. More critically, the terms "underestimate" and "overestimate" imply that the posterior is inherently closer to the truth than the other sources, assuming that the other sources are less accurate. This assumption is not self-evident. In inversion, adjusting emissions to match observations does not conclusively prove that the posterior emissions are improved, nor does it inherently indicate biases in other bottom-up inventories. To claim that HTAP and other sources are less accurate and to justify the terms "overestimate" and "underestimate," the authors need to provide a more convincing argument. Simply relying on posterior simulations is not sufficient to demonstrate the improvement in posterior emissions and the existence of biases in other bottom-up inventories. If a more convincing argument cannot be made, the authors should consider using more neutral terms to avoid implying a hierarchy of accuracy among different emission sources. An example of inconsistency can be found in Section 4.3, where the authors, in comparing their emission inventory with others, occasionally use alternative inventories as a basis to highlight the agreement and reduced uncertainty of their inventory compared to

bottom-up inventories. At other times, however, they claim that these alternative inventories exhibit significant uncertainties. This contradiction raises concerns about the clarity and consistency of the manuscript. It is essential that the authors provide a more coherent explanation or rationale for the varying assessments of uncertainty in other inventories.

**Reply:** Thanks for this important comment. We apology for the ambiguous employment of the terms like "underestimate" and "overestimate" when we compare the posteriori results with other sources. As the reviewer suggested, we have made revisions to these expressions throughout the manuscripts to ensure greater clarity in our intended meaning. For example, the unclear expression in line 47 has been revised to "the CAQIEI suggested higher NMVOC emissions than the other emission inventories by about 30.4–81.4% over the NCP region but suggested lower NMVOC emissions by about 27.6–0.0% over the SE region." Similar revisions have also been made throughout the manuscript.

We agree with the reviewer that adjusting emissions to match observations does not conclusively prove that the posterior emissions are more accurate than the other emission inventories. The purpose of comparisons of CAQIEI with other emission inventories in our manuscript is also not to prove the superiority of the inversion inventory, nor to rank the accuracy among different inventories. In fact, it is difficult to validate the estimated emission inventory because of the unavailability of the truth value or observations of air pollutant emissions, which is the main challenge faced in the research of emission inventory. Thus, one of the most important values of the top-down analysis is to provide valuable clues for verifying the emission inventory (Zhang et al., 2009; Streets et al., 2006) through the assimilations of observations of air pollutant concentrations, as demonstrated in many inversion studies, such as Miyazaki et al. (2017); Zheng et al. (2019); Goldberg et al. (2019) and so on. Therefore, similar to pervious top-down studies, the primary objectives of our manuscript are to utilize the inversion method to investigate the changes of air pollutant emissions in China, and to provide information about potential uncertainty in current understandings of the Chinese air pollutant emissions. The inversion emissions are also useful for improving the emission inventories. We feel sorry that the inappropriate use of the terms like "underestimate" or "overestimate" in the manuscript gives the implications about the hierarchy of the accuracy among different emission sources. This is not the intendency of our work. Following the suggestions of reviewer, we have made a throughout revision to our manuscript, especially for Sect 4.3 by using more neutral terms to make our intendency clearer. In the revised Sect 4.3, we only highlight the similarity and differences among the different inventories, and delete the terms like underestimate and overestimate. Meanwhile, the consistency in Sect 4.3 has also improved in the revised manuscript by giving a more coherent explanation. (please see lines 650–794 in the revised manuscript)

**Comment 2:** In particular, the authors' comparison of natural and anthropogenic species emissions (such as PM10 and NMVOC) reveals a significant issue. Natural sources inherently exhibit considerable uncertainty, and in many regions, natural sources contribute significantly more than anthropogenic sources. Therefore, using the uncertainty in natural sources as a basis does not necessarily indicate large uncertainties in anthropogenic sources within the bottom-up inventories. An inconsistency arises in Line 790, where the authors' explanation appears contradictory. They simultaneously assume minimal variations in natural sources and cite literature indicating an increasing trend in natural sources. Additionally, the manuscript attributes emission changes to anthropogenic sources while acknowledging substantial uncertainty in natural sources. If it is acknowledged that natural sources indeed carry significant uncertainty (which is indeed the case), the manuscript should avoid using terms such as "not captured," "overestimated," or "underestimated" concerning the bottom-up inventories. These terms imply a clear attribution of error that may not be justified given the uncertainties associated with natural sources. Clear and consistent handling of uncertainties in both natural and anthropogenic sources is crucial for maintaining the credibility of the manuscript.

**Reply:** Thanks for this suggestion. We acknowledge the significant uncertainty existed in the natural sources and its implications for the assessment of anthropogenic emissions. As we illustrated in the responses to Comment 1, we are not intended to conclusively prove the biases in existing emission inventories, but to provide clues for their possible deficiencies through the comparison of our inversion results with the other emission inventories. However, since our inversion inventory does not differentiate the natural and anthropogenic sources, the natural sources have to be considered to make different emission inventories comparable. We apology the inappropriate use of the terms "not captured", "overestimated" and "underestimated" in the comparisons of different emission inventories, which did not accurately convey our true intent and caused some inconsistencies and contradictions in the manuscripts as the reviewer mentioned. Following the suggestions of reviewer, we have made following revisions to provide a clear and consistent handling of uncertainties in the natural and anthropogenic sources in our manuscript:

1) Add the discussions about the uncertainty in the natural sources and its implications on the comparisons of inversion results with other anthropogenic emission inventories. We also highlighted in the revised manuscript that the comparisons conducted in this study is on the basis of the natural sources estimated by CAMS and GFAS inventories, which would be sensitive to the used natural emission inventories, and does not necessarily indicate large uncertainties in anthropogenic sources within the bottom-up inventories. (please see lines 657–665, 786–788, 994 – 1001 in the revised manuscript).

2) Rewrote the expressions including the terms such as "not captured," "overestimated," or "underestimated" concerning the bottom-up inventories throughout the manuscript. For example, the lines 766–767 in the original manuscript has been revised to "In particular, CAQIEI suggests increases of $NO_x$

emissions over the Central region, which is opposite to the previous emission inventories." The lines 781–782 in the original manuscript has also been deleted in the revised manuscript. More revision is available in the Sect 4.3 in the revised manuscript.

3) The inconsistent or contradictory expressions in the manuscript have also been revised to maintain the consistency of the manuscript. For example, we do not simultaneously attribute the increase in NMVOC emissions to biogenic sources while also stating that the variation in biogenic sources is not significant. Instead, in the revised manuscript, the emission trends of biogenic NMVOC (also for other species) are only estimated based on the CAMS emission inventory, and further analysis is also made based on this assumption (lines 593–597). The uncertainty of this assumption and its potential impacts on the comparisons of our inversion results with previous emission inventories were then discussed in the manuscript (please see lines 657–665, 786–788, 994 – 1001 in the revised manuscript). This would help increase the consistency of the manuscript. Regarding to the inconsistency that we attribute the emission changes to anthropogenic sources while acknowledging substantial uncertainty in natural sources. Since our inversion result do not differentiate the anthropogenic and natural sources, it is difficult to directly compare our inversion results with previous emission inventories. To deal with this issue, the natural sources estimated by CAMS and GFAS were used in this study to account for the influences of natura sources. Therefore, our primary intendency is to make an attempt to compare our inversion results with previous emission inventories on the basis of state-of-art estimations of natural source. However, as we are aware of that despite the use of state-of-art estimations of natural sources, there is still significant uncertainty in the estimated emission trends of natural sources, which would influence the comparison results of our inversion inventory with previous emission inventories. That's why we acknowledge the large uncertainty in the estimated natural sources. This could help the potential reader better understand the comparison results of CAQIEI with previous emission inventories. We feel sorry that the inappropriate expression in our original manuscript did not correctly convey our intention, and lead to inconsistency. In the revised manuscript, we explicitly pointed out that the comparison conducted in our study is on the basis of natural emissions estimated by CAMS and GFAS at the beginning of Sect. 4.3 (Lines 657 – 665), which would be sensitivity to the used natural sources. This would help potential readers better understand our comparison results and improve the consistency with the discussion of the uncertainty in natural sources in our manuscript. More revision is available in the Sect 4.3 in the revised manuscript.

**Comment 3**: In Line 272, it is mentioned that VOC emissions are optimized through assimilating ground-level O3 observations. However, several factors need consideration. On one hand, VOC-$O_3$ interactions involve strong nonlinear chemical reactions, and emission adjustments exhibit bidirectionality (Tang et al., 2016). Despite the convergence of simulations and observations, VOC inversion results may deteriorate due to these complexities. On the other hand, the majority of the national monitoring stations are situated in urban areas, whereas VOC primarily originates from suburban or rural regions. I am skeptical about the feasibility of assimilating $O_3$ to constrain VOC emissions. As evident from Figure 3, the posterior simulations do not show a significant improvement in $O_3$. As the authors noted, $O_3$ cannot effectively constrain precursor NOx (L278). Therefore, I recommend deleting the VOC emission inversion.

**Reply:** Thanks for this comment. We agree with the reviewer that the $NO_x$-VOC-$O_3$ nonlinear interaction would influence the inversion of NMVOC emission based on the $O_3$ concentration if were not well addressed. On the one hand, the $O_3$ concentrations are dependent not only on the NMVOC emissions but also on the $NO_x$ emissions. The errors in the a priori emissions of $NO_x$ would also contribute to the simulation errors of $O_3$, and deteriorate the inversion of NMVOC. This concern has been considered in our inversion method through two approaches. Firstly, the emissions of NOx and NMVOC were perturbed independently in our study, thus their contributions to the simulation errors of $O_3$ concentrations could be isolated through the use of ensemble simulations. Secondly, the use of iteration inversion method can further reduce the influence of the errors in $NO_x$ emissions on the inversion of NMVOC emission, since the errors in $NO_x$ emission would be constrained by its own observations during the iterations as we illustrated in lines 320–324 in the revised manuscript. This is in fact similar to the approach used by Xing et al. (2020) who firstly constrained the $NO_x$ emissions based on observations of $NO_2$, and then constrained the NMVOC emissions based on $O_3$ concentrations. Also, in Feng et al. (2024), the $NO_2$ concentrations were also assimilated to constrain the $NO_x$ emissions to account for the influences of errors in $NO_x$ emissions on the NMVOC emissions. These studies indicates that the iteratively nonlinear joint inversion of NOx and NMVOCs using multi-species observations adopted in our study is an effective and commonly used way to address the intricate relationship among VOC-NOx-$O_3$ (Feng et al., 2024). On the other hand, the emission adjustments exhibit bidirectionality dependent on the VOC-limited or NOx-limited regimes. According to the Fig 3 in the revised manuscript, the NMVOC emissions were adjusted in alignment with the direction of the $O_3$ errors, suggesting a VOC-limited regime over urban areas in China, given that the $O_3$ observation sites are predominantly situated in the urban areas. This agrees with Ren et al. (2022) who diagnosed the $NO_x$-VOC-$O_3$ sensitivity based on the satellite retrievals and found that the VOC-limited regimes are mainly located in the urban areas in China. This suggests that the relationship between the $O_3$ concentrations and VOC emissions could be reasonably reflected by inverse modeling. Moreover, considering there are transport of VOCs from suburban or rural areas (Liu et al., 2022), the $O_3$

concentrations in the urban areas could also provide information on the NMVOC emission over the suburban or rural areas. Therefore, although the majority of monitoring stations are located in urban areas, the NMVOC emissions over remote regions lacking observations could still be constrained to some extent through the utilization of covariance relationships estimated by the ensemble simulations. However, we agree with the reviewer that the lack of observation sites over the remote areas significantly hinders the fully constrains of the NMVOC emission over there and may lead to larger uncertainty. More observations over the suburban and rural areas are required to better constrain the NMVOC emissions in the future.

To date, we think there remains feasibility in utilizing the $O_3$ observations to constrain the VOC emissions. Besides our study, the assimilation of surface $O_3$ observations to constrain the VOC emissions has also been performed in other inversion studies, such as Xing et al. (2020) and Ma et al. (2019). Both of these studies have demonstrated the effectiveness of assimilating surface $O_3$ concentrations on the inversion of VOC emissions. For example, Ma et al. (2019) found that the assimilation of $O_3$ concentration could adjust the NMVOC emissions in the direction resembling the bottom-up inventories, and the forecast skill of $O_3$ concentrations were also improved, indicating that the constrained NMVOC emissions are improved relative to their priori. Our inversion results suggest similar effectiveness of the assimilation of $O_3$ concentrations on the NMVOC emissions as reflected by the improvement of $O_3$ simulations (Table 2 in the revised manuscript) and the overall consistency with the bottom-up inventories and top-down emission inventories using the satellite observation data (Souri et al., 2020). Meanwhile, there are limited ways to constrain the NMVOC emissions due to the lack of NMVOC observations. Previous inversion studies are mainly relied on the satellite observations of formaldehyde and glyoxal. However, these inversion studies are also hindered by the $NO_x$-VOC-$O_3$ chemistry and the inherent uncertainty in the satellite observations of formaldehyde and glyoxal (Cao et al., 2018; Stavrakou et al., 2015), leading to uncertainty in their estimates. Given that, we think it is still worth a try to advance our understanding of the NMVOC emissions in China by assimilating the surface $O_3$ concentrations. Therefore, we lean towards retaining the inversion results of NMVOCs. This on the one hand could provide the users of interest with some potential valuable information on the NMVOC emissions in China, and on the other hand can serve as a comparable reference for future VOC inversion studies based on other methods or observation data, which could help the development of the inversion method of NMVOC. However, we acknowledge the complexity of the inversion of NMVOC emission due to the nonlinear $NO_x$-VOC-$O_3$ interactions and the limited observation sites which were not fully addressed in our study. Therefore, more descriptions on the rationale and uncertainty in the inversion of NMVOC emissions based on $O_3$ concentrations have been added in the revised manuscript to assist the potential readers in properly utilizing the inversion results of NMVOC. In addition, more robust analysis of the effects of nonlinear $NO_x$-VOC-$O_3$ interactions and the number of observation sites should be performed in future to better illustrate the feasibility

of assimilating O$_3$ to constrain NMVOC emissions. Detailed revisions to the manuscript are available in lines 308–337 and 1002–1011 in the revised manuscripts.

**Comment 4:** The changes in observation coverage each year can significantly impact emission estimates. If the authors intend to include the years 2013-2014 in this study, they should compare the impact of site differences on emissions for a more robust analysis. If the authors aim to investigate trends, it is advisable to delete emissions in the 2013-2014 period, as this might otherwise potentially mislead readers, given that the changes during this period do not contribute meaningfully to the study's overall trend analysis. There also appears to be some discrepancies in the manuscript where emission changes are often stated as occurring from 2015-2017, while the text descriptions indicate the period as 2013-2017, as seen in lines 452, 485, and 561, among others. Furthermore, it is important to note that the changes in emissions observed from 2015-2017 not necessarily reflect the overall reduction rate of the action plan for the entire period of 2013-2017. Additionally, the data from 2015-2017 alone may not be sufficient to conclude that the emission reduction rate during the 2013-2017 period is lower than that during the 2018-2020 action plan.

**Reply:** Thanks for this comment. Following the suggestions of the reviewer, we added more analysis on the influences of the site differences on the emission inversions in the revised manuscript. Figure R1 shows the spatial distributions of the observation sites used in inversion during 2013–2015 when the number of observation sites changed rapidly. It can be seen that the observation sites were mainly concentrated in the megacity clusters (e.g., North China Plain, Yangtze River Delta and Pearl River Delta) and the capital cities of each province in 2013. The number of observation sites continued to increase across the China in 2014 and 2015. In particular, many areas that were previously unobserved in 2013 have added monitoring stations, which significantly increased the observation coverage in China especially over the NW, NE, SW and Central regions. Figure R2 shows the calculated emission increments at the observation sites (a posteriori minus a priori) for different species in China from 2013 to 2015 under the scenario of fixed observation sites (blue lines) and varying observation sites (orange). In the fixed-site scenario, it is assumed that the number of observation sites remains constant at the 2013 level while in the varying-site scenario, the number of observation sites increases over time. The differences in emission increments between these two scenarios are used to analyze the impact of changes in the observation coverage on the emission inversions. Please note that, to simplify calculations, we only computed the emission increments at the locations of the observation sites. Therefore, they may not be equal to the emission increments calculated for the entire grid as reported in the paper. However, they are still useful indicators for the effects of emission inversion. In addition, since we did not consider the temporal variation in the a priori emissions, the changes of emission increments at the observation sites can be used to approximate the temporal variations of the a posterior emissions. It can be

clearly seen that that there are obvious differences in the emission increments between the two scenarios. The emission increment is larger in the varying-site scenario than that in the fixed-site scenario for all species due to the increases of observation sites. Moreover, as indicated in Fig. R2, the changes of observation sites were shown to significantly affect the estimation of the emission trend in 2013 and 2014. Most of species showed decreasing trends in their inversed emission under the fixed-site scenario. However, under the varying-site scenario, the decreasing trends were smaller for $PM_{2.5}$, $NO_x$ and NMVOC, and the emissions of $PM_{10}$ and CO even showed increasing trends. This is due to that the emission increments were positive over most of observation sites for these species as demonstrated in Fig.3 in the revised manuscript. Thus, the increases of observation site would lead to increases of positive emission increments and higher a posteriori emissions, which may counteract the decreasing trends or even lead to an opposite trend. These results provide the evidences that the increasing trends in the total emissions of $PM_{10}$ and CO from 2013 to 2015 seen in Fig. 6 and Fig. 7 are highly likely to be a spurious trend caused by the changes of observation coverage. The weak emission changes in $PM_{2.5}$ and $NO_x$ (Fig. 6 and Fig. 7) may also be related to the changes in the number of observation sites. The $SO_2$ emission is an except that its calculated trend is larger under the varying-site scenario than that under the fixed-site scenario. This is because that the emission increment for the $SO_2$ is generally negative over the most of sites, thus the increased observation sites would lead to larger decreasing trend in the inversed emissions of $SO_2$. These results highlighted the significant influences of the site differences on the estimated emissions and their trends. Therefore, as the reviewer suggested, we recommend not to use the emission in 2013 and 2014 when analyze the trends of the emissions, which has been written in the user notes of our data products. we also only investigated the emission changes from 2015 to 2020 in our manuscript to avoid misleading the potential users. Following the suggestions, the analysis on the influences of the site differences on the emission inversion has been added in the revised manuscript to remind potential users to be aware of this issue. Please see lines 199–207 in the revised manuscript, lines 3–28 and Fig. S1–S2 in the revised supplement.

We feel sorry for the discrepancies in the manuscript. The 2013-2017 is merely used as the names for the clean air action plans during 2013–2017, rather than referring to the years calculating emission changes. This confusion has been revised by using more accurate expression. For example, the lines 36 – 37 in the original manuscript have been revised to "It is also estimated that the emission reductions were larger during 2018–2020 (from -26.6% to -4.5%) than during 2015–2017 (from -23.8% to 27.6%) for most species." in the revised manuscript (lines 36–37). Also, we agree with the reviewer that the changes in emissions observed from 2015-2017 not necessarily reflect the overall reduction rate of the action plan for the entire period of 2013-2017, and that they were not sufficient to conclude that the emission reduction rate during the 2013-2017 period is lower than that during the 2018-2020 action plan. Thanks for the reviewer's reminder. We have softened the

statement of this conclusion and added relevant discussions in the revised manuscript to enhance the rigor of our paper (please see lines 423–427 and 984–987 in the revised manuscript).

[Figure]

Figure R1 Spatial distributions of observation sites in (a) 2013, (b) 2014 and (c) 2015. The observation sites in 2013 were marked as black dots, while the added observation sites from 2013 to 2014 and those from 2014 to 2015 were marked as red and green dots respectively.

[Figure]

Figure R2: the calculated total emission increments at the observation sites for different species under the fixed-site scenario and varying-site scenario.

**Comment 5**: The authors use PM$_{2.5}$ observations to simultaneously constrain BC, OC, and primary PM$_{2.5}$. If they do not consider inter-species correlations or use random perturbations, and, for instance, if BC and OC increase while PM2.5 decreases in one ensemble member. How do they constrain emissions when the simulated PM$_{2.5}$ and observations are the same.

**Reply:** Thanks for this comment. Since we aim to estimate the emissions separately for BC, OC and primary PM$_{2.5}$, it is necessary to perturb the a priori emissions of BC, OC and primary PM$_{2.5}$ randomly during the inversion to avoid the spurious correlations between the non- or weakly related variables. This enables us to statistically differentiate the contributions of their emission errors to the simulation errors of PM$_{2.5}$ concentration through the use of ensemble simulation, making the emissions of BC, OC, primary adjusted by different scaling factors (i.e., $\beta_{BC}, \beta_{OC}$ and $\beta_{PMF}$). Also, it is feasible that using same perturbation coefficient to perturb their emissions. As I understand it, this treatment is closer to what the reviewer mentioned regarding considering the inter-species correlations. This is equivalent to perturbing only the total PM$_{2.5}$ emissions and allows the estimations of total PM$_{2.5}$ emissions by using a same scaling factor. Therefore, applying independent perturbations or using the same perturbation coefficient are both commonly employed methods in the inversion studies. We agree with the reviewer that it is possible for BC and OC to increase while PM$_{2.5}$ decreases in one ensemble member under the conditions of random perturbation. However, as we used the deterministic form of EnKF (DEnKF), the ensemble member is only used to calculate the background perturbation $X_i^b$ and the subsequent background covariance matrix $B_e^b$. The behavior of single ensemble would not significantly influence of the statistical properties of the ensemble, unless there is spurious correlation among the emissions of different PM$_{2.5}$ components. For example, if there is a spurious negative correlation between the perturbed emissions of BC and PMF, there would lead to a false negative correlation between the PM$_{2.5}$ concentrations and the emissions of BC. That's why the emissions of different PM$_{2.5}$ component should be perturbed randomly during the assimilation. Also, in the DEnKF, the observation innovation (observation minus simulation) is only determined by the observation and ensemble mean of the simulated PM$_{2.5}$. Therefore, whether to adjust the a priori emission is only determined by the deviations between the ensemble mean and observations, rather than the simulation results in one ensemble member. Meanwhile, since the emissions were perturbed unbiasedly in our study, the ensemble mean of perturbed emissions is equal to the a priori emission. Thus, the ensemble mean of the model simulation is mainly determined by the a priori emission. If the ensemble mean of PM$_{2.5}$ simulations equals the observed values, it suggests that the a prior emission may have no error, and thus, we won't make adjustments to the prior emission. However, we acknowledge that in such cases, there may still be errors in the emissions of BC, OC, and primary PM$_{2.5}$, such as the underestimation of BC and OC while the overestimation of primary PM$_{2.5}$. This is primarily due to that we only assimilate the observations of total PM$_{2.5}$ mass without the assimilation of speciated PM$_{2.5}$

observations. In the absence of detailed speciated $PM_{2.5}$ observations, assimilating only total $PM_{2.5}$ concentration observations cannot adjust the proportions of emissions for different $PM_{2.5}$ components when the observations and simulations are equal, which is a specific manifestation of the uncertainty resulting from adjusting $PM_{2.5}$ emissions solely based on total $PM_{2.5}$ concentration. This limitation has been explicitly pointed out in our manuscript and thus only the total $PM_{2.5}$ emissions were provided to prevent the potential misuse of $PM_{2.5}$ component emissions without sufficient validation. Following the suggestions of reviewer, we give more discussions about the limitations of only assimilation total $PM_{2.5}$ mass in the revised manuscripts (please see lines 299–303 in the manuscript).

**Comment 6**: The authors simultaneously constrain concentrations and emissions, emphasizing that concentration errors arise from emission uncertainties, implying a shared source of uncertainty (L222). In this context, the question arises whether optimizing concentrations would diminish emission uncertainties, thereby affecting emission estimates.

**Reply:** Thanks for this comment. We feel sorry for this confusion. Since we used the modified EnKF method to constrain the emissions (Wu et al., 2020), the concentrations were not optimized simultaneously with the emissions. As we written in the manuscript, the modified EnKF is an offline application of the EnKF method that decouples the analysis step from the ensemble simulation. In this method, the ensemble simulation was performed firstly with the perturbed emissions, thus the concentration errors estimated by the ensemble simulation mainly stem from the emission uncertainty as we written in line 240–241 in the revised manuscript. After that, the observations were assimilated to constrain the emissions. During this step, the concentration was not required to be optimized but was used to estimate the covariance between the emission and concentration. Therefore, although the concentration was included in the state variable as illustrated in Eq. (1), it was not optimized during the inversion step and thus would not diminish emission uncertainties. The feasibility of this method in the emission inversion has been discussed and tested in Wu et al. (2020) through the observation system simulation experiments, which shows good performances of this method in reducing the errors in the a priori emission inventory. To avoid this confusion, we have added relevant explanations regarding to the optimization of the state of concentrations in the revised manuscript. Please see lines 215–219 and lines 231–233 in the revised manuscript.

**Comment 7:** The NOx emission changes optimized by the authors appear to contradict existing research findings and are inconsistent with recent emission reduction policies. Despite citing the study by Zheng et al., (2018), the actual NOx emissions reported by Zheng show a significant decrease. Could this discrepancy be attributed to the bottom-up inventory lacking sufficient statistics on mobile vehicle emissions? Moreover,

according to Zheng's study, industrial and power plant emissions collectively contribute to over 50% of total emissions. Hence, the second reason provided by the authors may not be suitable if the industrial and power plant emissions are substantial contributors.

**Reply:** Thanks for this comment. The $NO_x$ emission changes are determined by the combined effects of pollution control and growth of activity. If the effects of air pollution control exceed the additional emissions caused by the growth of activity, the $NO_x$ emission would decrease and vice versa. According to Zheng et al. (2021), the increases of activity levels has offset the mitigation effects of the emission controls for the traffic and industrial sectors. For example, the vehicle growth yielded increases of 1.4 Tg $NO_x$ emission compared with its 2010 level, which exceeded the emission reductions of $NO_x$ (1.3 Tg) achieved by the pollution control on the traffic section (Zheng et al., 2018). This indicates that the increase in the activity levels and the insufficient effectiveness of emission control in industrial and traffic sectors do exist, and has been considered in some bottom-up emission inventories, such as MEIC. The discrepancy in the estimated $NO_x$ emission changes between inversion results and other emission inventories thus reflect the uncertainty in the quantification of the combined effects of $NO_x$ emission control and activity growth. Our inversion results suggest that the offset effects of activity growth may be larger than the mitigation effects of the pollution control during 2015–2017, while previous emission inventories suggest larger mitigation effects of air pollution control than the offset effects of activity growth. As the reviewer mentioned, this discrepancy could be attributed to the bottom-up inventory lacking sufficient statistics on the mobile vehicle or other sectors. For example, previous inversion study by Kong et al. (2022) found there are numerous small-to-medium local sources of $NO_x$ emission related to the minor roads or small human settlements in China that are unclear or missing in the MEIC, EDGAR and CEDS emission inventory. The emission trends of these unaccounted local sources are thus not able to be considered by these emission inventory, which could be an important factor for the differences between our inversion results and previous inventories. Following the suggestions of the reviewer, more discussions have been added in the revised manuscript to better explain the discrepancy between our inversion results and previous inventories (please see lines 826–831 in the revised manuscript). We agree with the reviewer that industrial and power plant emissions are substantial contributors to the $NO_x$ emission. Our second reason is mainly related to the control of traffic sector. Following the suggestions of the reviewer, we have deleted this reason in the revised manuscript. Please see lines 566 – 568 in the revised manuscript.

**Comment 8:** "Figure 12 shows significant discrepancies between the results of the author's inversion and other bottom-up inventories. According to other literature, it is known that China experienced two peaks in VOC emissions in May and July 2015. The variation in VOC emissions closely follows the changes in O3

levels, suggesting a strong dependence of VOC on O3 variations. This raises the question of whether non-linear changes are being overlooked.

**Reply:** Thanks for this comment. Firstly, we feel sorry that there is an error in the description of Fig. 12. It actually presents the monthly profiles of the averaged air pollutant emissions from 2015 to 2018 rather than just for 2015. Figure R3 shows the comparisons of the standardized monthly profile of the averaged a posteriori NMVOC emission and MDA8h $O_3$ concentrations in China from 2015 to 2018. The standardized monthly profiles were calculated by dividing them by their mean values. It shows that the monthly variation of the a posteriori NMVOC emissions have significant similarity to the monthly variation of the observed MDA8h $O_3$ concentrations, as the reviewer mentioned. However, there are still obvious differences in their month variations. For example, the peak values of the observed MDA8h $O_3$ concentrations occur in May, while the peak values of the a posteriori NMVOC emissions occur in July. This suggest that the monthly profile of a posteriori NMVOC emissions is not solely dependent on the variations of the MDA8h $O_3$ concentrations, thus some non-linear changes, such as unfavorable meteorological conditions that lead to high $O_3$ concentrations even with relatively low NMVOC emissions, could be represented in our method to some extent. This is also the advantage of the EnKF method which provide an effective way to consider the flow dependent non-linear relationships between the concentrations and emissions. For example, the sensitivity of $O_3$ concentrations to the NMVOC emissions under different meteorological conditions could be represented in the EnKF through the use of ensemble simulation. Nevertheless, we acknowledge that there could still be some unknown nonlinear-changes in the model or the EnKF method that were not well considered during inversion, which leads to uncertainty in the a posteriori NMVOC emissions.

[Figure]

Figure R3: the standard monthly variation of the averaged a posteriori NMVOC emission and MDA8h $O_3$ in China during 2015–2018.

**Comment 9:** The author has provided a high-resolution, multispecies emission inventory. To facilitate users' understanding of the data's accuracy, could you please provide information on the uncertainties associated with different species, allowing users to assess the error range in the data?

**Reply:** Thanks for this suggestion. Within the framework of the EnKF assimilation, the information on the uncertainty of the a posteriori emission for different species could be provided by the analysis ensemble spread estimated form the standard deviation across the analysis ensemble (Miyazaki et al., 2020). According to Sakov and Oke (2008), the analysis ensemble can be calculated as follows:

$$\mathbf{X^a} = \mathbf{X^b} - \frac{1}{2}\mathbf{KHX^b} \tag{R1}$$

Based on the analysis ensemble, the uncertainty of the a posteriori emission was estimated as follows: 101.4% ($PM_{2.5}$), 102.5% ($PM_{10}$), 26.7% ($SO_{2)}$, 46.8% (CO), 31.8% (NOx) and 65.5% (NMVOC). However, it should be noted that such uncertainty was only calculated under the framework of the EnKF constructed in this study, which is dependent on the assigned value of the a priori emission uncertainty, observation errors and the number of assimilated observations. In addition, we only considered the a priori emission uncertainty and the observation errors during the inversion. The influences of the other error sources, such as uncertainty in the chemistry transport model, meteorology simulations and the inversion method were not considered. Therefore, the current estimated uncertainty should be considered as a lower bound for the real uncertainty. More systematic analysis that thoroughly consider the uncertainty sources regarding the emission inversion should be conducted in future to give a more accurate estimation of the uncertainty in our products. Following the suggestions, we have added the descriptions on the uncertainty on the a posteriori emission in the revised manuscript (please see lines 885–899 in the revised manuscript).

**Specific comments:**

**Comment 1:** Change "Fengwei Plain" to "Fenwei Plain".

**Reply:** Done.

**Comment 2:** In L212, the VOC adjustment factor was omitted.

**Reply:** Done.

**Comment 3:** Since MOZART data products are no longer updated, are the boundary conditions in this study

based on simulations conducted by the author's team? Additionally, maritime shipping emissions have a significant impact on the generation of $NO_2$ and $O_3$ in coastal provinces. Has the model taken into account inputs from maritime emissions? Why was a localized scale of 180 km chosen?

**Reply:** We feel sorry for the lack of clear explanation regarding the use of MOZART data products. As mentioned by the reviewer, the MOZART data products have not been updated since 2018. Therefore, the multi-year average results from the MOZART were used for the simulations after 2018. Because most of the model boundaries were set in the clean areas and are located at distance from China, we assumed that the differences in boundary conditions would not significantly affect the modeling results in China. following the suggestions of reviewer, we have clarified the use of MOZART data in the revised manuscript (please see lines in 160–165). The ship emissions have been considered in our inversion study as a part of the HTAP emission inventory as we illustrated in lines 145 – 147 in the revised manuscript. The localized scale of 180km was chosen according our previous inversion study (Kong et al., 2023), and is similar to the localization scales used in Feng et al. (2020) and Ma et al. (2019) which were determined based on the wind speed and the lifespan of the species (please see lines 282 – 284 in the revised manuscript).

**Comment 4:** In L291, is it necessary to reassemble simulations for each iteration, or is it multiple inversions on the original ensemble? If multiple iterations are performed, does it imply that the posterior approaches the observations more closely with each iteration? Why was the choice made to iterate twice?

**Reply:** Thanks for raising this important issue. In this method, we conduct a new simulation by using the a posteriori emission from the previous iteration to update the ensemble mean of the original ensemble. This enables the observational information and the adjusted emissions to be promptly incorporated into the model, thereby providing feedback for the adjustments of emission in the next iteration. However, we did not reassemble the ensemble simulation for each iteration due to the expensive computational cost of the ensemble simulation. Therefore, in each iteration calculation, the ensemble perturbation that were used to calculate the background error covariance matrix remains the same with only the ensemble mean being updated based on the inversion results of the previous iteration.

As mentioned by the reviewer, it is implied that the posteriori should approach the observations more closely with each iteration, which has been demonstrated in our previous inversion studies (Kong et al., 2023). As seen in the Fig.3 of Kong et al. (2023) (fig. R4), four times of iterations were conducted to adjust the $SO_2$ emissions. We can clearly see that due to the large positive biases in the a priori $SO_2$ emissions, the model still has large positive biases in simulated $SO_2$ concentration over all regions of China even after assimilation (first iteration). With the increases in the iteration times, the biases and errors continued to decrease which is consistent with the implication in the iteration inversion. However, the degree of improvement will gradually

diminish with an increasing number of iterations until convergence is achieved. Our previous study shows that the improvement become no longer significant after two iterations. Thus, we choice the two times of the iteration in this study maintain a balance between the filter performance and the computational cost. Following the suggestions of reviewer, we have added more description on the implementation of iteration inversion scheme and the determination of the times of iteration in the revised manuscript. Please see lines 345–354 and 358–360.

[Figure]

Figure R4: Comparisons of the observed and simulated mean SO₂ concentrations using emissions of different iteration time over (a) the NCP region, (b) NE region, (c) SE region, (d) SW region, (e) NW region and (f) central region (taken from Fig 3 in Kong et al. (2023)).

**Comment 5:** With such a high grid resolution of 15 km, how does the computational cost for the inversion of multiple years in the ensemble calculations? Additionally, what is the size of the assimilation window?

**Reply:** Thanks for this comment. We have added more details related to this issue in the revised manuscript. The computational cost is still expensive for the inversion of multiple years in the ensemble calculations with a high grid resolution of 15km. According to our estimation, we used about 12000 CPUs in the ensemble simulation, and the computational time for one-year ensemble simulation reaches approximately 2 million core-hours. Thanks to the "Earth System Science Numerical Simulator Facility" (EarthLab) which provide us

with ample computational resources to complete this research. Since we constrained the daily emissions, the size of the assimilation window is 24h in our study. Please see lines 1255–1258 in the revised manuscript.

**Comment 6:** The inflation factor 'r' varies for each window. Is 'r' a matrix or a scalar? If it is a scalar, could the author provide the specific range of 'r'?

**Reply:** The inflation factor is a scalar but its value varies in the space and time, which is calculated by using the method of Wang and Bishop (2003):

$$\lambda = \frac{\left(\mathbf{R}^{-1/2}d\right)^{\mathrm{T}}\mathbf{R}^{-1/2}d - p}{trace\left\{\mathbf{R}^{-1/2}\mathbf{H}\mathbf{B}_e^b\left(\mathbf{R}^{-1/2}\mathbf{H}\right)^{\mathrm{T}}\right\}} \tag{R2}$$

$$\boldsymbol{d} = \boldsymbol{y}^o - \mathbf{H}\overline{\boldsymbol{x}^b} \tag{R3}$$

where $\lambda$ is the inflation factor, $\boldsymbol{d}$ is the observation innovation, $\mathbf{R}$ is the observation error covariance matrix, and $\boldsymbol{p}$ is the number of observations. following the suggestions of the reviewer, we analyze the calculated value of inflation factor in the revised manuscript. Table R1 shows the calculated average value (standard deviation) of the used inflation factor for the different species over different regions of China. It shows that the inflation factor over the east China (including NCP and SE region) was generally round 1.0, suggesting that the original ensemble can well represent the simulation errors of the different air pollutants over these regions. The inflation factor is larger over the western China (including SW, NW and Central regions), especially for $PM_{10}$ and $SO_2$, suggesting that the original ensemble may underestimate the simulation errors of the air pollutants. This is associated with the large biases in the simulated air pollutant concentrations over there and reflect that the emission uncertainties assumed in our studies may be underestimated over these regions. it also highlighted the importance of the use of inflation method during the inversion, otherwise it would lead to filter divergency caused by the underestimations of the background error covariance. Following the suggestions of the reviewer, we have added the discussions of the inflation factors in the revised manuscript. Please see lines 269 – 277 in the revised manuscript and Table S1 in the revised supplement.

Table R1 The average mean (standard deviation) of the calculated factor for the inflation of the ensemble member over different regions of China for different species

| | NCP | NE | SE | SW | NW | Central |
|---|---|---|---|---|---|---|
| $PM_{2.5}$ | 1.0 (0.2) | 1.7 (1.6) | 1.0 (0.0) | 6.8 (8.5) | 3.1 (3.8) | 3.9 (3.9) |
| $PM_{10}$ | 1.4 (0.7) | 7.2 (8.0) | 2.4 (0.8) | 78.1 (108.2) | 26.3 (36.5) | 36.0 (49.0) |
| $SO_2$ | 1.4 (0.7) | 4.1 (3.2) | 2.3 (0.8) | 176.1 (254.6) | 7.8 (6.5) | 58.6 (72.5) |
| $NO_x$ | 1.0 (0.1) | 1.7 (0.7) | 1.2 (0.3) | 8.1 (5.3) | 2.8 (1.3) | 5.4 (4.1) |

| | | | | | | |
|---|---|---|---|---|---|---|
| CO | 1.0 (0.1) | 2.8 (2.3) | 1.4 (0.4) | 18.8 (16.8) | 6.8 (6.9) | 8.6 (10.0) |
| NMVOC | 1.4 (0.6) | 4.5 (4.4) | 1.6 (0.5) | 8.1 (8.6) | 6.5 (5.8) | 8.1 (10.1) |

**Comment 7:** Table 3 lacks information regarding the year.

**Reply:** Thanks for this comment. we have added the year information in the revised Table 3.

**Comment 8:** How were diurnal variations of the emissions specified?

**Reply:** Since the a priori emission inventory did not provide the information on the diurnal variations of the emissions, and it is difficult to estimate the diurnal variations of the emissions for different sectors over the whole China, we used the constant diurnal variation during the assimilation. We acknowledge that the uncertainty in the diurnal variations of the emission would lead to uncertainty in our inversion results. However, the diurnal variations of the emission may not significantly influence the simulation results of the daily mean concentrations of air pollutants according to the sensitivity experiments conducted by Wang et al. (2010) in China and Mues et al. (2014) in Europe. As shown in Wang et al. (2010), the differences in the simulated concentrations of $SO_2$, $NO_2$ and $O_3$ with or without considerations of diurnal variation were estimated to be within 1 ppbv in China. Therefore, the diurnal variation may not significantly influence our inversion results. Following the suggestion of reviewer, we have added the description of the settings of the diurnal variation of the emissions in the revised manuscript. Please see lines 155–159.

**Comment 9:** How is the optimization of VOC components conducted when VOC consists of multiple components?

**Reply:** Since we did not have the observations of the VOC components, we only optimize the gross emissions of the VOC during our assimilation which has been pointed out in the revised manuscript (lines 336–337).

**Comment 10:** Region name in Figure 1 refers to specific areas. Consider a different expression to avoid potential ambiguity.

**Reply:** Thanks for this suggestion. Since the region names in Fig. 1 were also used in our other papers, we are intended to keep their name to guarantee the consistency among our works.

**Comment 11:** Please consider adopting a clearer representation for Figure 11.

**Reply:** Thanks for this suggestion. We have redrawn the Figure 11 in revised manuscript for a clearer representation as shown in Fig. R5:

[Figure]

Figure R5: Comparisons of the averaged emissions of (a) NOₓ, (b) SO₂, (c) CO, (d) PM₂.₅, (e) PM₁₀, and (f) NMVOCs over different regions in China from 2015 to 2018 between CAQIEI and previous inventories added with natural sources.

**Comment 12:** L483 Change "Fig.3" to "Fig. 4"

**Reply:** Done

**References**

Cao, H. S., Fu, T. M., Zhang, L., Henze, D. K., Miller, C. C., Lerot, C., Abad, G. G., De Smedt, I., Zhang, Q., van Roozendael, M., Hendrick, F., Chance, K., Li, J., Zheng, J. Y., and Zhao, Y. H.: Adjoint inversion of Chinese non-methane volatile organic compound emissions using space-based observations of formaldehyde and glyoxal, Atmospheric Chemistry and Physics, 18, 15017-15046, 10.5194/acp-18-15017-2018, 2018.

Feng, S., Jiang, F., Qian, T., Wang, N., Jia, M., Zheng, S., Chen, J., Ying, F., and Ju, W.: Constraint of non-methane volatile organic compound emissions with TROPOMI HCHO observations and its impact on summertime surface ozone simulation over China, EGUsphere, 2024, 1-34, 10.5194/egusphere-2023-2654, 2024.

Feng, S., Jiang, F., Wang, H., Wang, H., Ju, W., Shen, Y., Zheng, Y., Wu, Z., and Ding, A.: NOx Emission Changes Over China During the COVID-19 Epidemic Inferred From Surface NO2 Observations, Geophys. Res. Lett., 47, e2020GL090080, 10.1029/2020gl090080, 2020.

Goldberg, D. L., Saide, P. E., Lamsal, L. N., de Foy, B., Lu, Z. F., Woo, J. H., Kim, Y., Kim, J., Gao, M., Carmichael, G., and Streets, D. G.: A top-down assessment using OMI NO2 suggests an underestimate in the NOx emissions inventory in Seoul, South Korea, during KORUS-AQ, Atmospheric Chemistry and Physics, 19, 1801-1818, 10.5194/acp-19-1801-2019, 2019.

Kong, L., Tang, X., Zhu, J., Wang, Z., Sun, Y., Fu, P., Gao, M., Wu, H., Lu, M., Wu, Q., Huang, S., Sui, W., Li, J., Pan, X., Wu, L., Akimoto, H., and Carmichael, G. R.: Unbalanced emission reductions of different species and sectors in China during COVID-19 lockdown derived by multi-species surface observation assimilation, Atmos. Chem. Phys., 23, 6217-6240, 10.5194/acp-23-6217-2023, 2023.

Liu, Y., Qiu, P., Li, C., Li, X., Ma, W., Yin, S., Yu, Q., Li, J., and Liu, X.: Evolution and variations of atmospheric VOCs and O3 photochemistry during a summer O3 event in a county-level city, Southern China, Atmos. Environ., 272, 118942, https://doi.org/10.1016/j.atmosenv.2022.118942, 2022.

Ma, C. Q., Wang, T. J., Mizzi, A. P., Anderson, J. L., Zhuang, B. L., Xie, M., and Wu, R. S.: Multiconstituent Data Assimilation With WRF-Chem/DART: Potential for Adjusting Anthropogenic Emissions and Improving Air Quality Forecasts Over Eastern China, J. Geophys. Res.-Atmos., 124, 7393-7412, 10.1029/2019jd030421, 2019.

Miyazaki, K., Eskes, H., Sudo, K., Boersma, K. F., Bowman, K., and Kanaya, Y.: Decadal changes in global surface NOx emissions from multi-constituent satellite data assimilation, Atmospheric Chemistry and Physics, 17, 807-837, 10.5194/acp-17-807-2017, 2017.

Miyazaki, K., Bowman, K., Sekiya, T., Eskes, H., Boersma, F., Worden, H., Livesey, N., Payne, V. H., Sudo, K., Kanaya, Y., Takigawa, M., and Ogochi, K.: Updated tropospheric chemistry reanalysis and emission estimates, TCR-2, for 2005–2018, Earth Syst. Sci. Data, 12, 2223-2259, 10.5194/essd-12-2223-2020, 2020.

Mues, A., Kuenen, J., Hendriks, C., Manders, A., Segers, A., Scholz, Y., Hueglin, C., Builtjes, P., and Schaap, M.: Sensitivity of air pollution simulations with LOTOS-EUROS to the temporal distribution of anthropogenic emissions, Atmospheric Chemistry and Physics, 14, 939-955, 10.5194/acp-14-939-2014, 2014.

Ren, J., Guo, F., and Xie, S.: Diagnosing ozone–NOx–VOC sensitivity and revealing causes of ozone increases in China based on 2013–2021 satellite retrievals, Atmos. Chem. Phys., 22, 15035-15047, 10.5194/acp-22-15035-2022, 2022.

Sakov, P. and Oke, P. R.: A deterministic formulation of the ensemble Kalman filter: an alternative to ensemble square root filters, Tellus Ser. A-Dyn. Meteorol. Oceanol., 60, 361-371, 10.1111/j.1600-0870.2007.00299.x, 2008.

Souri, A. H., Nowlan, C. R., Abad, G. G., Zhu, L., Blake, D. R., Fried, A., Weinheimer, A. J., Wisthaler, A., Woo, J. H., Zhang, Q., Miller, C. E. C., Liu, X., and Chance, K.: An inversion of NOx and non-methane volatile organic compound (NMVOC) emissions using satellite observations during the KORUS-AQ campaign and implications for surface ozone over East Asia, Atmospheric Chemistry and Physics, 20, 9837-9854, 10.5194/acp-20-9837-2020, 2020.

Stavrakou, T., Muller, J. F., Bauwens, M., De Smedt, I., Van Roozendael, M., De Maziere, M., Vigouroux, C., Hendrick, F., George, M., Clerbaux, C., Coheur, P. F., and Guenther, A.: How consistent are top-down hydrocarbon emissions based on formaldehyde

observations from GOME-2 and OMI?, Atmospheric Chemistry and Physics, 15, 11861-11884, 10.5194/acp-15-11861-2015, 2015.

Streets, D. G., Zhang, Q., Wang, L. T., He, K. B., Hao, J. M., Wu, Y., Tang, Y. H., and Carmichael, G. R.: Revisiting China's CO emissions after the Transport and Chemical Evolution over the Pacific (TRACE-P) mission: Synthesis of inventories, atmospheric modeling, and observations, J. Geophys. Res.-Atmos., 111, 16, 10.1029/2006jd007118, 2006.

Wang, X., Liang, X.-Z., Jiang, W., Tao, Z., Wang, J. X. L., Liu, H., Han, Z., Liu, S., Zhang, Y., Grell, G. A., and Peckham, S. E.: WRF-Chem simulation of East Asian air quality: Sensitivity to temporal and vertical emissions distributions, Atmos. Environ., 44, 660-669, https://doi.org/10.1016/j.atmosenv.2009.11.011, 2010.

Wang, X. G. and Bishop, C. H.: A comparison of breeding and ensemble transform Kalman filter ensemble forecast schemes, Journal of the Atmospheric Sciences, 60, 1140-1158, 10.1175/1520-0469(2003)060<1140:Acobae>2.0.Co;2, 2003.

Wu, H., Tang, X., Wang, Z., Wu, L., Li, J., Wang, W., Yang, W., and Zhu, J.: High-spatiotemporal-resolution inverse estimation of CO and NOx emission reductions during emission control periods with a modified ensemble Kalman filter, Atmos. Environ., 236, 117631, https://doi.org/10.1016/j.atmosenv.2020.117631, 2020.

Xing, J., Li, S. W., Jiang, Y. Q., Wang, S. X., Ding, D., Dong, Z. X., Zhu, Y., and Hao, J. M.: Quantifying the emission changes and associated air quality impacts during the COVID-19 pandemic on the North China Plain: a response modeling study, Atmospheric Chemistry and Physics, 20, 14347-14359, 10.5194/acp-20-14347-2020, 2020.

Zhang, Q., Streets, D. G., Carmichael, G. R., He, K. B., Huo, H., Kannari, A., Klimont, Z., Park, I. S., Reddy, S., Fu, J. S., Chen, D., Duan, L., Lei, Y., Wang, L. T., and Yao, Z. L.: Asian emissions in 2006 for the NASA INTEX-B mission, Atmospheric Chemistry and Physics, 9, 5131-5153, 10.5194/acp-9-5131-2009, 2009.

Zheng, B., Zhang, Q., Geng, G., Chen, C., Shi, Q., Cui, M., Lei, Y., and He, K.: Changes in China's anthropogenic emissions and air quality during the COVID-19 pandemic in 2020, Earth Syst. Sci. Data, 13, 2895-2907, 10.5194/essd-13-2895-2021, 2021.

Zheng, B., Chevallier, F., Yin, Y., Ciais, P., Fortems-Cheiney, A., Deeter, M. N., Parker, R. J., Wang, Y. L., Worden, H. M., and Zhao, Y. H.: Global atmospheric carbon monoxide budget 2000-2017 inferred from multi-species atmospheric inversions, Earth System Science Data, 11, 1411-1436, 10.5194/essd-11-1411-2019, 2019.

Zheng, B., Tong, D., Li, M., Liu, F., Hong, C. P., Geng, G. N., Li, H. Y., Li, X., Peng, L. Q., Qi, J., Yan, L., Zhang, Y. X., Zhao, H. Y., Zheng, Y. X., He, K. B., and Zhang, Q.: Trends in China's anthropogenic emissions since 2010 as the consequence of clean air actions, Atmospheric Chemistry and Physics, 18, 14095-14111, 10.5194/acp-18-14095-2018, 2018.

---

## Author Response (AR2)

**Response to Reviewer #3 (ESSD-2023-477)**

We Thank Reviewer for his/her constructive comments

Responses to the Specific comments

**General comments:** The authors effectively answered the reviewer's questions and improved the quality of data visualization and the coherence of the writing. More specifically, the revised manuscript presents more evidence supporting the use of O3 on constraining VOC emissions. The discussion of the uncertainty and potential limitations of top-down emissions derived from this study further enables readers to fully comprehend the data's strengths and weaknesses. There are only a few minor and specific comments that need to be addressed before publication in ESSD:

**Reply:** Many thanks for the careful read and constructive comments/suggestions to our manuscript. We have carefully considered the comments and revised the manuscript accordingly. Please refer to our responses for more details given below.

**Comment 1:** It is encouraging to see that the trend and the amount for most top-down emissions are consistent with previous studies or inventories. The primary concern is the posterior CO emissions, which are substantially higher than values reported in other research and increase by 2-3 times compared to the prior emissions. It is recommended to obtain independent measurements (e.g., data from field campaign) to further evaluate the reliability of the posterior CO emissions. Alternatively, the TROPOMI CO data may be useful for evaluating top-down CO emissions between 2018 and 2020 because the instrument is sensitive to the integrated amount of CO (Landgraf et al., 2016), including the contribution of the planetary boundary layer, making it particularly suitable for detecting surface sources of CO. The authors are not required to do this at this stage, but please consider conducting independent cross-validation for the top-down emissions that shown larger discrepancies to other studies in the future.

**Reply:** Thanks for this good suggestion. We agree with the reviewer that more independent validation should be conducted in future to better explain the discrepancies between the top-down and the bottom-up emission inventories. Following the suggestions of the reviewer, we have added more discussions about this in the revised manuscript. Please see lines 1068–1072:

"*Current inversion emission inventory is mainly assessed by the surface observations and previous emission inventories. more independent observations, such as the satellite observation data, should be used in future to further validate the inversion results of this study and its derived findings. For example, the independent measurements from field campaign or satellite retrievals (e.g., TropOMI CO data) can help validate the reliability of the much higher a posterior CO emission in CAQIEI than the previous inventories in the future*".

**Changes in the manuscript: lines 1068–1072.**

**Comment 2:** While the authors use the NOx-O3-VOC chemistry to support the use of O3 to constrain VOC emissions, the role of CO in O3 formation should be first investigated. I am concerned that the substantially greater posterior CO emissions may bring errors into the O3 simulation and impair the quality of VOC emissions inversion. It may be better to first constrain NOx and CO emissions and ensure that O3 can be properly simulated before optimizing VOC emissions with O3 data.

**Reply:** Thanks for this good suggestion. In fact, this has been considered in our iteration inversion method. At each time of the iteration, the emissions of $NO_x$ and CO are constrained and used for the inversions at next time of iteration. Therefore, after the first round of the iteration, the CO and $NO_x$ emissions has been constrained to account for the possible influences of the errors in the CO emissions on the $O_3$ simulation and VOC emission. As the reviewer stated, constraining the $NO_x$ and CO emissions first is another good way to avoid the influence of the errors in the $NO_x$ and CO emissions on the inversion of VOC, which has been used in Xing et al. (2020). Following the suggestions of reviewer, we have clarified this in the revised manuscript. Please see lines in 346 – 347.

**Changes in the manuscript: lines 346–347.**

**Comment 3**: If both biogenic and anthropogenic VOC emissions are perturbed independently, they can be optimized separately. I'm curious why the authors did not explore doing this, which could aid in the intercomparison of top-down VOC emissions with results from other studies.

**Reply:** Thanks for this suggestion. We feel sorry that we did not make it clear that we did not perturb the biogenic and anthropogenic VOC emission independently in our inversion framework to reduce the freedom of the system. Therefore, we did not optimize the anthropogenic and biogenic separately. To help the potential readers better understand our inversion result, we have clarified this in the revised manuscript. Please see lines 245–247.

**Changes in the manuscript: lines 245–247.**

**Specific comments:**
**Comment 1:** Line 243-245: Underestimating the background error covariance results in an underestimating of the emissions adjustment rather than an overcorrection.

**Reply:** Thanks for this correction. We have corrected this in the revised manuscript. Please see lines 253.
**Changes in the manuscript: lines 253.**

**Comment 2:** Figure 3: Since bias is typically defined as simulation minus observations and this figure displays prior and posterior OmF, please consider replacing "bias" with "OmF" in the figure legend and caption to prevent confusion and consist with the description in the manuscript.

**Reply:** Thanks for this suggestion. We have replaced the "bias" with "OmF" in the Figure 3.

**Changes in the manuscript: Figure 3.**

**Comment 3:** Line 381-382: Shouldn't the NO2 OmF over the NCP and SE show negative values during the winter and positive values during the summer?

**Reply:** Thanks for this comment. We have double checked the Fig.3 and found it is that the $NO_2$ OmF over the NCP and SE show negative values during summer and positive values during winter. The labels in the X-axis represents the June in each year.

**Changes in the manuscript: None**

**Comment 4:** Line 883-884: As the posterior CO emissions derived in this work are substantially greater than those derived in other studies and inventories, it may be an exaggeration to state that the assimilation of CO surface observations is superior to the assimilation of satellite CO measurements. This statement requires further verification (e.g., comparison to independent CO measurement), and I recommend using a more neutral description here.

**Reply:** Thanks for this good suggestion. We have revised the sentence "In this case, the assimilation of surface observations (our study) is shown to be superior to the assimilation of satellite observations (TCR-2), as our results are more consistent with other bottom-up inventories." as follows: "In this case, the estimated changes of CO emissions by CAQIEI are more consistent with those estimated by other bottom-up inventories. Considering this, the TCR-2 may have biases in the estimated changes of CO emissions in China from 2015 to 2017, which could be related the suboptimal performance of the data assimilation caused by the underestimated background errors of CO or too short assimilation window for the CO emission estimates (Miyazaki et al., 2020). Please see lines 917–920.

**Changes in the manuscript: lines 917–920**

**Comment 5:** Figure S17: Since each plot displays different emissions species, please remove the regions (e.g., NW, SE) listed in the figures.

**Reply:** we feel sorry for the typo error in the captions of Fig. S17. In fact, each plot displays the emission changes over different regions. We have revised the caption of the Fig. S17 in the revised manuscript but retained the regions listed in the figures to facilitate the understanding of potential readers.

**Changes in the supplement: Figure S17**

Figure R1 shows the time series of hourly concentrations of different air pollutants in China obtained from observation and simulation driven by the CAQIEI and more recent bottom-up inventories. Comparisons of the evaluation statistics of these two simulation scenarios are also presented in Table R1. It shows that updating the bottom-up emission inventories to more recent years did improve the model performance compared to the outdated a priori emission inventory (Table 2 in the manuscript), suggesting that the bottom-up emission inventory has to some extent captured the changes of air pollutant emissions in China. It is also encouraging to find that the model performance driven by CAQIEI and MEIC-HTAPv3 is similar for the concentrations of $PM_{2.5}$, $PM_{10}$, and $SO_2$ over the NCP, NE, SE and SW regions, both significantly improved from the a priori emission inventory (Table 2 in the manuscript). This suggest that both the top-down and recent bottom-up emission inventories have good performance in capturing the emission changes of these species over these regions and they yield consistent estimations. However, the model simulation driven by MEIC-HTAPv3 still have negative biases in the CO concentrations possibly due to the underestimations of CO emissions as we illustrated in the manuscript. Similarly, due to the errors in the dust emission, there are negative biases in the simulated $PM_{2.5}$ and $PM_{10}$ concentrations over the western China driven by MEIC-HTAPv3. On the contrary, the simulated $NO_2$ concentrations in MEIC-HTAPv3 are higher than the observations over the NCP, NE and SE regions, which also partly contributes to the underestimated $O_3$ concentrations over these regions. The CAQIEI generally achieves better performance in simulating the air pollutant concentrations in China as indicated by higher values of correlation coefficient and lower values of bias and root mean square of error in the model simulation driven by CAQIEI than that driven by MEIC-HTAPv3 (Table R2). Following the suggestions of the reviewer, we have added the comparisons of the model performance driven by CAQIEI with that driven by more recent bottom-up emission inventories in the revised manuscript and supplement. Please see lines 440–448 in the revised manuscript and lines 42–69 in the revised supplement.

**Changes in the manuscript: lines 440 – 448**

**Changes in the supplement: lines 42 – 69, Figure S25 and Table S3.**

[Figure]

Figure R1: Timeseries of observed (black lines) and simulated concentrations of different air pollutants in China driven by CAQIEI (red lines) and MEIC-HTAPv3 (blue lines) over different regions of China.

Table R1 Evaluation statistics of the model simulation driven by CAQIEI (outside brackets) and more recent bottom-up inventories (inside brackets) in 2020

| | PM$_{2.5}$ ($\mu g/m^3$) | PM$_{10}$ ($\mu g/m^3$) | SO$_2$ ($\mu g/m^3$) | NO$_2$ ($\mu g/m^3$) | CO ($mg/m^3$) | O$_3$ ($\mu g/m^3$) |
|---|---|---|---|---|---|---|
| R | 0.77 (0.53) | 0.73 (0.44) | 0.37 (0.19) | 0.69 (0.45) | 0.67 (0.40) | 0.75 (0.48) |
| MB | 3.6 (5.3) | -0.3 (-14.9) | 0.3 (0.7) | -0.9 (6.7) | -0.06 (-0.4) | 6.3 (-13.7) |
| NMB (%) | 10.5 (15.8) | -0.5 (-25.9) | 2.6 (7.6) | -3.4 (26.2) | -8.9 (-52.7) | 10.2 (-22.1) |
| RMSE | 24.6 (34.2) | 37.4 (49.1) | 10.9 (13.6) | 15.9 (25.1) | 0.4 (0.6) | 30.3 (42.3) |

**Comment 3**: The authors mention the exclusion of meteorological and model errors in the ensemble simulation (lines 244 and 367) and briefly discuss future work on model biases (lines 1012-1018). More discussion is necessary. Could the authors provide results on the meteorological model performance against observations and discuss the potential impact of meteorological errors on the inverse emission inventory? For example:

a) If the WRF model systematically overestimates near-surface wind speed, what is the impact on the inventory?

b) If the WRF model systematically underestimates nighttime boundary layer height or mixing (e.g., Du et al., 2020, https://acp.copernicus.org/articles/20/2839/2020/), what is the impact on the inverse inventory?

c) Discussion of meteorological conditions should not be limited to these examples.

**Reply:** Thanks for this good suggestion. Figure R2–R7 presents the evaluations of the simulated meteorological parameters, including zonal wind (U), meridional wind (V), temperature (T), relative humidity (RH) and precipitation, against the observations obtained from China Meteorological Administration (Figure R8). It shows that the WRF simulation can generally captured the main features of the different meteorological parameters over the different regions of China. The calculated correlation coefficient is 0.49–1.00 for different parameters, and the values of MB (RMSE) are -0.36–0.01 (0.3–0.52) m/s for U, -0.37–0.32 (0.32–0.80) m/s for V, -1.11–1.11 (0.6–2.17) ℃ for T, -11.2 to -2.59 (6.94–12.06) % for RH, and -2.05–37.35 (5.45–61.62) mm for precipitation. This suggests WRF simulation generally well reproduce the meteorological conditions for all regions of China, which is acceptable for the inversion estimates. Nevertheless, the random errors in the WRF simulation could lead to uncertainty in the emission inversions. For example, the errors in the simulated wind would influence the transportation of the air pollutant and lead to uncertainty in the emissions distributions. The air temperature and relative humidity would affect the atmospheric chemistry. The simulated relative humidity is generally lower than the observations, which may weaken the formation of secondary aerosol. On the contrary, the simulated precipitation was higher than the observation for most regions which would lead to overestimations of the wet removal of air pollutants. As a result, there may be a positive tendency in the inversed emission inventory due to the errors in the simulated relative humidity and precipitation. Besides these parameters, as the reviewer stated, the accuracy of the simulated boundary layer is also important for the performance of the emission inversions. If the WRF systematically underestimates the boundary layer, the vertical diffusions of the air pollutants would be suppressed, which would lead to overestimated surface air pollutant concentrations. in this case, there would be a negative tendency in the inverse emission inventory to compensate for the positive biases in the surface concentrations caused by the underestimated boundary layer height. However, as we illustrated in the manuscript, it is still difficult to quantify the influences of the meteorological errors on the emission inversions, as the errors in the meteorological simulation and chemical transport model itself would interact with each other. More comprehensive analysis should be conducted in the future to better understand the impacts of the meteorological and model errors on the inverse emission inventory. Following the suggestions of reviewer, we have added the evaluations about the meteorological performance and their potential influences on the inverse emission inventories. please see lines 173–174, 1051–1063 in the revised manuscript and lines 3 – 13 in the revised supplement.

**Changes in the manuscript: lines 173–174 and 1051–1063.**

**Changes in the supplement: lines 3 – 13, Figure S18–24, Table S2.**

Table R2 Evaluation statistics for the meteorology simulation

| Region | U (m/s) | | | V (m/s) | | | T (℃) | | | RH (%) | | | Precipitation (mm/month) | | |
|---|---|---|---|---|---|---|---|---|---|---|---|---|---|---|---|
| | R | MB | RMSE | R | MB | RMSE | R | MB | RMSE | R | MB | RMSE | R | MB | RMSE |
| NCP | 0.95 | 0.01 | 0.30 | 0.95 | -0.02 | 0.49 | 1.00 | -0.42 | 0.84 | 0.95 | -11.24 | 11.66 | 0.95 | 3.74 | 18.56 |
| NE | 0.94 | 0.37 | 0.51 | 0.89 | -0.08 | 0.49 | 0.99 | -1.11 | 2.17 | 0.77 | -2.59 | 7.18 | 0.97 | 12.09 | 19.76 |
| SE | 0.84 | -0.27 | 0.37 | 0.98 | -0.37 | 0.80 | 1.00 | -0.40 | 0.60 | 0.88 | -7.00 | 7.58 | 0.94 | 37.35 | 61.62 |
| SW | 0.63 | -0.44 | 0.52 | 0.69 | 0.04 | 0.37 | 0.99 | 1.11 | 1.27 | 0.87 | -5.84 | 6.94 | 0.92 | 16.85 | 40.18 |
| NW | 0.49 | -0.36 | 0.51 | 0.58 | 0.32 | 0.43 | 0.99 | 0.83 | 1.91 | 0.79 | -9.49 | 12.06 | 0.51 | -2.05 | 5.45 |
| CENTRAL | 0.95 | 0.10 | 0.41 | 0.70 | -0.08 | 0.32 | 1.00 | -0.27 | 0.93 | 0.85 | -8.59 | 10.30 | 0.97 | 4.64 | 10.87 |

**NCP**

[Figure]

Figure R2: Timeseries of observed (red dots) and simulated (blue line) monthly values of (a) zonal wind, (b) meridional wind, (c) temperature, (d) relative humidity and (e) precipitation over NCP region from Jan 2013 to Dec 2020.

**NE**

[Figure]

Figure R3: Same as in Figure R2 but over the NE region.

**SE**

[Figure]

Figure R4: Same as in Figure R2 but over the SE region.

**SW**

[Figure]

Figure R5: Same as in Figure R2 but over the SW region.

**NW**

[Figure]

Figure R6: Same as in Figure R2 but over the NW region.

[Figure]

**Central**

Figure R7: Same as in Figure R2 but over the Central region.

**Comment 4:** Why are the natural emission inventories used in the a priori emissions different from those used when comparing CAQIEI with other inventories? If the authors believe the CAMS and GFAS emission inventories are state-of-the-art, why not use them in the a priori emissions? Please clarify.

**Reply:** Thanks for this comment. On the one hand, some of the natural emission inventories did not update their estimations to more recent years. For example, the MACC emission inventory only provide the estimations of the biogenic emissions from 1980 to 2010. On the other hand, the use of different sources of natural emissions may provide us with more independent validations on our inversion results. As illustrated in our responses to Comment 1, at the time this study was conducted, the CAMS and GFAS emission were not released yet. Also, although we believe that the CAMS and GFAS emission inventory are state-of-the-art estimations of the long-term natural air pollutant emissions, large uncertainty still exists in their estimations and there is no obvious evidence show that the GFAS and CAMS significantly outperform the a priori emission inventory we used for the year of 2013. Meanwhile, our previous study suggests that the choice of a priori emission inventory would not significantly influence the inversion results (Kong et al., 2023). So, it may be better to using the CAMS and GFAS emission inventories as the a priori natural emission inventory, but it is not necessary.

**Changes in the manuscript: None.**

**Comment 5:** The authors admit in lines 364-366 that they only provide emissions of $PM_{2.5}$ (PMF+BC+OC) due to a lack of speciated $PM_{2.5}$ observations. In light of this, what is the rationale behind using $PM_{2.5}$ observations to simultaneously constrain BC, OC, and primary unspeciated $PM_{2.5}$ emissions, instead of using $PM_{2.5}$ observations to constrain primary $PM_{2.5}$ emissions? It seems that the authors try to provide emission inventories for BC, OC, and primary unspeciated $PM_{2.5}$, while only presenting total primary $PM_{2.5}$ emissions due to large uncertainties in its components.

**Reply:** Yes, our primary purpose is to separately estimate the emissions of BC, OC and primary $PM_{2.5}$ through the assimilation of $PM_{2.5}$. However, due the lack of enough speciated $PM_{2.5}$ observation, the model performance driven by the CAQIEI for the BC, OC and primary unspeciated $PM_{2.5}$ have not been thoroughly evaluated. It is thus currently unclear for the quality of the inverse emissions of BC, OC and primary unspeciated $PM_{2.5}$. Considering this, we have reservations about the inverse emissions of BC, OC and unspeciated $PM_{2.5}$ and only provide the emissions of total primary $PM_{2.5}$ in current stage. In future, we will collect more speciated $PM_{2.5}$ observations to comprehensively quantify the accuracy of the inverse emission of BC, OC and primary unspeciated $PM_{2.5}$, after which the emissions of these species would be released. Meanwhile, the speciated $PM_{2.5}$ observations could be assimilated under the current framework of the inversion of $PM_{2.5}$ emission. This may provide us with further constrains on the emissions of BC, OC and primary $PM_{2.5}$. That's why we did not chose to use the $PM_{2.5}$ to directly constrain the primary $PM_{2.5}$ emissions in our study. following the suggestions of this reviewer, we have clarified this in the revised manuscript. Please see lines 381–391.

**Changes in the manuscript: lines 381–391.**

**Specific Comments:**

**Comment 1:** Lines 248-251 are confusing. If the authors do not consider coarse dust emission in the inversion, do they use PM10 concentrations driven by other sources to constrain PM10 emissions? Why assume large errors in simulated coarse dust concentration will impact the inversion of PM10 emissions? Please clarify.

**Reply:** Thanks for this comment. Yes, we used the simulated $PM_{10}$ concentrations driven by other sources to constrain $PM_{10}$ emissions, which were defined as follows:

Simulated $PM_{10}$ = BC + primary organic aerosol + primary unspeciated $PM_{2.5}$ + primary unspeciated $PM_{10}$ + secondary organic aerosol + secondary inorganic aerosol + fine dust + fine sea salt     (R1)

The reason that we did not include the coarse dust components in the calculation of simulated $PM_{10}$ is that there is large uncertainty in the simulated coarse dust emission by the current dust emission schemes, with the difference among different schemes being up to several orders of magnitude (Zeng et al., 2020; Kang et al., 2011). Thus, the errors in the simulated coarse dust would significantly influence the simulation results of $PM_{10}$. Meanwhile, since we did not perturb the dust emissions in current inversion framework, the errors in the dust emission would be attributed to the errors in other sources. Therefore, as we illustrated in the manuscript, the differences between the a posteriori estimates of $PM_{2.5}$ emission and the a priori emission inventory is also partly caused by errors in the fine dust emission. Things were different for $PM_{10}$, as we found that the simulated coarse dust concentration could sometimes be several orders of magnitude higher than the observed $PM_{10}$ concentration, leading to too low inversion results of $PM_{10}$ emission (approximately 0) over the regions that were not the dust source regions but were influenced by the transportation of coarse dust. Considering this, we chose not to include the simulated coarse dust concentration in the calculation of $PM_{10}$. This is similar to assume that the coarse dust emission is equal to zero during the assimilation, which would help deal with the problems of errors in the simulated coarse dust concentration. But as we noted the in the manuscript, the inversion results of $PM_{10}$ emission contains the coarse dust emissions. Following the suggestion of the reviewer, we have clarified this in the revised manuscript. Please see lines 257–266.

**Changes in the manuscript: lines 257–266.**

**Comment 2:** Can the authors clarify whether all emissions are updated simultaneously at each iteration?

**Reply:** Yes, all the emissions are updated simultaneously at each iteration, and we have clarified this in the revised manuscript. Please see lines 375–376.

**Changes in the manuscript: lines 375–376.**

**Comment 3:** Do the figures in Figure 5 show the national total for China? Please specify in the caption.

**Reply:** Yes, it shows the national total for China. We have specified it in the revised caption of Figure 5. Thanks for this reminder.

**Changes in the manuscript: lines 1223 – 1224.**

**Comment 4:** In line 364, should "PM10 (PM2.5+PMF)" be written as "PM10 (PM2.5+PMC)"?

**Reply:** Many thanks for the careful read of our manuscript. Yes, this is a typo error, we have fixed in the revised manuscript.

**Changes in the manuscript: line 387.**

**Comment 5:** In Table 2, how is R estimated? Do the authors average all stations across China and calculate R for the averaged observation versus averaged simulation, or do they calculate R for each station and average the R values across China?

**Reply:** Thanks for this comment. in Table 2, we firstly catenated the time series of the air pollutant concentrations at each station into a single vector. Then the values of R were calculated based on the catenated time series of the observed and simulated concentrations. Therefore, the calculated R in Table 2 represents the whole model performance in capturing the spatial and temporal variations of the observed air pollutant concentrations. following the suggestions of reviewer, we have added a footnote in Table 2 to clarify this in the revised manuscript. Please see lines 1109–1110.

**Changes in the manuscript: lines 1109–1110.**

**Comment 6:** Please specify in line 318 the potential impact of nighttime O3 chemistry on the inversion to better illustrate the rationale for using MDA8 O3.

**Reply:** Thanks for this comment. during the nighttime, the photochemical reaction gradually disappears, so does the chemical relationship between the $O_3$ and NMVOC emissions. the errors in the simulated nighttime O3 chemistry, such as the simulation errors in the titration effects of $NO_x$, may lead to uncertainty in the inversion results of NMVOC. Following the suggestions of reviewer, we have clarified this in the revised manuscript. Please see lines 334–335.

**Changes in the manuscript: lines 334–335.**

**Comment 7:** For Figure S16, consider adjusting the min and max values of the Y-axis to better illustrate annual variations. For example, consider not starting the Y-axis from 0.

**Reply:** Thanks for this comment. we have adjusted the min and max values of the Y-axis in revised Figure S16 to better illustrate the annual variations of the natural emissions in China.

**Changes in the supplement: Figure S16.**

**References:**

[revised manuscript text omitted]

**Supplementary Material**

**Text S1: Evaluation of the meteorological simulation**

The performance of meteorological simulation is important for the inversion estimation since the meteorological parameters influence the transport, chemical and removal process of air pollutants and affect the estimation of flow-dependent background error covariance. Figure S18–S23 presents the comparisons of the simulated meteorological parameters, including zonal wind (U), meridional wind (V), temperature (T), relative humidity (RH) and precipitation, against the observations obtained from China Meteorological Administration (Figure S24). Evaluation statistics of meteorological simulation are also presented in Table S1. It shows that the WRF simulation can generally captured the main features of the different meteorological parameters over the different regions of China. The calculated correlation coefficient is 0.49–1.00 for different parameters, and the values of MB (RMSE) are -0.36–0.01 (0.3–0.52) m/s for U, -0.37–0.32 (0.32–0.80) m/s for V, -1.11–1.11 (0.6–2.17) $^{\circ}$C for T, -11.2 to -2.59 (6.94–12.06) % for RH, and -2.05–37.35 (5.45–61.62) mm for precipitation. This suggests WRF simulation generally well reproduce the meteorological conditions for all regions of China, which is acceptable for the inversion estimates.

**Text S2: Assessment of the influences of site differences on the emission inversions**

The emission increments at the observation sites (a posteriori minus a priori) for different species in China from 2013 to 2015 under the scenarios of fixed observation sites (blue lines) and varying observation sites (orange) were calculated to assess the influences of the site differences on the emission inversions (Fig. S2). In the fixed-site scenario, it is assumed that the number of observation sites remains constant at the 2013 level while in the varying-site scenario, the number of observation sites increases over time. The differences in emission increments between these two scenarios are used to analyze the impact of changes in the observation coverage on the emission inversions. Please note that, to simplify calculations, we only computed the emission increments at the locations of observation sites. Therefore, they may not be equal to the emission increments calculated for the entire grid as reported in the paper. However, they are still useful indicators for the effects of emission inversion. In addition, since we did not consider the temporal variation in the a priori emissions, the changes of the emission increments can be used to approximate the temporal variations of the a posterior emissions. It can be clearly seen that that there are obvious differences in the emission increments between the two scenarios. The emission increment is larger in the varying-site scenario than that in the fixed-site scenario for all species due to the increases of observation sites. Moreover, as indicated in Fig. S2, the changes of observation sites were shown to significantly affect the estimation of the emission trend in 2013 and 2014. Most of species showed decreasing trends in their inversed emission under the fixed-site scenario. However, under the varying-site scenario, the decreasing trends were smaller for $PM_{2.5}$, $NO_x$ and NMVOC, and the emissions of $PM_{10}$ and CO even showed increasing trends. This is due to that the emission increments were positive over most of observation sites for these species as demonstrated in Fig.3. Thus, the increases of observation site would lead to increases of positive emission increments and higher a posteriori emissions, which may counteract the decreasing trends or even lead to an opposite trend. These results provide the evidences that the increasing trends in the total emissions of $PM_{10}$ and CO from 2013 to 2015 seen in Fig. 6 and Fig. 7 are highly likely to be a spurious trend caused by the changes of observation coverage. The weak emission changes in $PM_{2.5}$ and $NO_x$ (Fig. 6 and Fig. 7) may also be related to the changes in the number of observation sites. The $SO_2$ emission is an except that its calculated trend is larger under the varying-site scenario than that under the fixed-site scenario. This is because that the emission increment for the $SO_2$ is generally negative over the most sites, thus the increased observation sites would lead to larger decreasing trend in the inversed emissions of $SO_2$. To date, these results highlighted the significant influences of the site differences on the estimated emissions and their trends. Therefore, we recommend not to use the emission in 2013 and 2014 when analyze the trends of the emissions.

**Text S3: Comparisons of model performance driven by CAQIEI with that driven by more recent bottom-up emission inventories**

To obtain a better understanding of the accuracy of our inverse emission inventory, we conducted a one-year simulation of air pollution in China for year 2020 with more recent bottom-up emission inventories and compared its performance with that driven by the CAQIEI. The used bottom-up inventories in this simulation case includes the HTAPv3 (Crippa et al., 2023) inventory for the anthropogenic emissions outside China with a base year of 2018; the MEIC inventory for the anthropogenic emissions over China with a base year of 2020; the CAMS emission inventory (https://ads.atmosphere.copernicus.eu/cdsapp#!/dataset/cams-global-emission-inventories?tab=overview, last access: 19 June 2024) for the biogenic, soil and oceanic emissions; and the Global Fire Assimilation System (GFAS) (Kaiser et al., 2012) for the biomass burning emissions. Note that since the MEIC emission inventory does not include the ship, air and waste emissions. Emissions from these sectors over China were provided by the HTAPv3 emission inventory. For clarity, in following content, we name this simulation case as the MEIC-HTAPv3 based on the anthropogenic emission inventory used.

Figure S25 shows the time series of hourly concentrations of different air pollutants in China obtained from observation and simulation driven by the CAQIEI and more recent bottom-up inventories. Comparisons of the evaluation statistics of these two simulation scenarios are also presented in Table S3. It shows that updating the bottom-up emission inventories to a more recent year does improve the model performance compared to the outdated a priori emission inventory (Table 2), suggesting that the bottom-up emission inventory has to some extent captured the changes of air pollutant emissions in China. It is also encouraging to find that the model performance driven by CAQIEI and MEIC-HTAPv3 is similar for the $PM_{2.5}$, $PM_{10}$, and $SO_2$ over the NCP, NE, SE and SW regions, both significantly improved from the a priori emission inventory. This suggest that both the top-down and recent bottom-up emission inventories have good performance in capturing the emission changes of these species over these regions and they yield consistent estimations. However, the model simulation driven by MEIC-HTAPv3 still have negative biases in the CO concentrations possibly due to the underestimations of CO emissions as we illustrated in Sect.4.3.1.3. Similarly, due to the errors in the dust emission, there are negative biases in the simulated $PM_{2.5}$ and $PM_{10}$ concentrations over the western China driven by MEIC-HTAPv3. On the contrary, the simulated $NO_2$ concentrations in MEIC-HTAPv3 are higher than the observations over the NCP, NE and SE regions, which also partly contributes to the underestimated $O_3$ concentrations over these regions. The CAQIEI generally achieves better performance in simulating the air pollutant concentrations in China as indicated by higher values of correlation coefficient and lower values of bias and root mean square of error in the model simulation driven by CAQIEI than that driven by MEIC-HTAPv3 (Table S3).

**Table S1: Evaluation statistics for the meteorology simulation**

| Region | U (m/s) | | | V (m/s) | | | T (°C) | | | RH (%) | | | Precipitation (mm/month) | | |
|---|---|---|---|---|---|---|---|---|---|---|---|---|---|---|---|
| | R | MB | RMSE | R | MB | RMSE | R | MB | RMSE | R | MB | RMSE | R | MB | RMSE |
| NCP | 0.95 | 0.01 | 0.30 | 0.95 | -0.02 | 0.49 | 1.00 | -0.42 | 0.84 | 0.95 | -11.24 | 11.66 | 0.95 | 3.74 | 18.56 |
| NE | 0.94 | 0.37 | 0.51 | 0.89 | -0.08 | 0.49 | 0.99 | -1.11 | 2.17 | 0.77 | -2.59 | 7.18 | 0.97 | 12.09 | 19.76 |
| SE | 0.84 | -0.27 | 0.37 | 0.98 | -0.37 | 0.80 | 1.00 | -0.40 | 0.60 | 0.88 | -7.00 | 7.58 | 0.94 | 37.35 | 61.62 |
| SW | 0.63 | -0.44 | 0.52 | 0.69 | 0.04 | 0.37 | 0.99 | 1.11 | 1.27 | 0.87 | -5.84 | 6.94 | 0.92 | 16.85 | 40.18 |
| NW | 0.49 | -0.36 | 0.51 | 0.58 | 0.32 | 0.43 | 0.99 | 0.83 | 1.91 | 0.79 | -9.49 | 12.06 | 0.51 | -2.05 | 5.45 |
| CENTRAL | 0.95 | 0.10 | 0.41 | 0.70 | -0.08 | 0.32 | 1.00 | -0.27 | 0.93 | 0.85 | -8.59 | 10.30 | 0.97 | 4.64 | 10.87 |

**Table S2 The average mean (standard deviation) of the calculated factor for the inflation of the ensemble member over**
**different regions of China for different species**

| | NCP | NE | SE | SW | NW | Central |
|---|---|---|---|---|---|---|
| $PM_{2.5}$ | 1.0 (0.2) | 1.7 (1.6) | 1.0 (0.0) | 6.8 (8.5) | 3.1 (3.8) | 3.9 (3.9) |
| $PM_{10}$ | 1.4 (0.7) | 7.2 (8.0) | 2.4 (0.8) | 78.1 (108.2) | 26.3 (36.5) | 36.0 (49.0) |
| $SO_2$ | 1.4 (0.7) | 4.1 (3.2) | 2.3 (0.8) | 176.1 (254.6) | 7.8 (6.5) | 58.6 (72.5) |
| $NO_x$ | 1.0 (0.1) | 1.7 (0.7) | 1.2 (0.3) | 8.1 (5.3) | 2.8 (1.3) | 5.4 (4.1) |
| CO | 1.0 (0.1) | 2.8 (2.3) | 1.4 (0.4) | 18.8 (16.8) | 6.8 (6.9) | 8.6 (10.0) |
| NMVOC | 1.4 (0.6) | 4.5 (4.4) | 1.6 (0.5) | 8.1 (8.6) | 6.5 (5.8) | 8.1 (10.1) |

**Table S3: Evaluation statistics of the model simulation driven by CAQIEI (outside brackets) and more recent bottom-up**
**inventories (inside brackets) in 2020**

| | $PM_{2.5}$ $(\mu g/m^3)$ | $PM_{10}$ $(\mu g/m^3)$ | $SO_2$ $(\mu g/m^3)$ | $NO_2$ $(\mu g/m^3)$ | CO $(mg/m^3)$ | $O_3$ $(\mu g/m^3)$ |
|---|---|---|---|---|---|---|
| R | 0.77 (0.53) | 0.73 (0.44) | 0.37 (0.19) | 0.69 (0.45) | 0.67 (0.40) | 0.75 (0.48) |
| MB | 3.6 (5.3) | -0.3 (-14.9) | 0.3 (0.7) | -0.9 (6.7) | -0.06 (-0.4) | 6.3 (-13.7) |
| NMB (%) | 10.5 (15.8) | -0.5 (-25.9) | 2.6 (7.6) | -3.4 (26.2) | -8.9 (-52.7) | 10.2 (-22.1) |
| RMSE | 24.6 (34.2) | 37.4 (49.1) | 10.9 (13.6) | 15.9 (25.1) | 0.4 (0.6) | 30.3 (42.3) |

[Figure]

- • observation sites in 2013
- • added sites from 2013 to 2014
- • added sites from 2014 to 2015

**Figure S1: Spatial distributions of observation sites in (a) 2013, (b) 2014 and (c) 2015. The observation sites in 2013 were marked as black**
**dots, while the added observation sites from 2013 to 2014 and those from 2014 to 2015 were marked as red and green dots, respectively.**

[Figure]

**Figure S2: The calculated total emission increments at the observation sites for different species under the fixed-site scenario and varying-**
**site scenario.**

[Figure]

**Figure S3: Emission changes of PM₂.₅ from 2013 to 2020 over different regions of China obtained from CAQIEI.**

[Figure]

**Figure S4: Same as Fig. S3 but for PM₁₀.**

[Figure]

**Figure S5: Same as Fig. S3 but for SO₂.**

[Figure]

**Figure S6: Same as Fig. S3 but for CO.**

[Figure]

**Figure S7: Same as Fig. S3 but for NO$_x$.**

[Figure]

**Figure S8: Same as Fig. S3 but for NMVOC.**

[Figure]

**Figure S9: Spatial distributions of the averaged emissions of different air pollutants in China during 2015–2018 obtained from CAQIEI,**
**MEIC, HTAPv3, EDGARv6, CEDS and TCR-2. Note the due to absence of gridded products of the ABaCAS inventory, we did not provide**
**its spatial distributions. Also, the natural sources were not added to the previous emission inventories in this figure because of the different**
**spatial resolutions among these inventories.**

[Figure]

**Figure S10: Time series of annual NO$_x$ emissions over of different regions of China: (a) NCP, (b) NE, (c) SE, (d) SW, (e) NW and (f) Central from 2013 to 2020 obtained from CAQIEI and previous inventories. Note that the natural sources were not included in the previous inventories in this figure.**

[Figure]

**Figure S11: Same as Fig. S10 but for SO₂.**

[Figure]

**Figure S12: Same as Fig. S10 but for CO.**

[Figure]

**Figure S13: Same as Fig. S10 but for PM2.5.**

[Figure]

**Figure S14: Same as Fig. S10 but for PM₁₀**

[Figure]

**Figure S15: Same as Fig. S10 but for NMVOC.**

[Figure]

Figure S16: Time series of annual natural emissions of (a) NO$_x$, (b) SO$_2$, (c) CO, (d) PM$_{2.5}$, (e) PM$_{10}$ and (f) NMVOC in China from 2013 to 2018. The considered natural sources includes the biogenic, biomass burning and soil emissions.

[Figure]

**Figure S17: Comparisons of the calculated emission changes of NOx, SO2, CO, PM2.5, PM10, and NMVOCs over (a) NCP, (b)**
**NE, (c) SE, (d) SW, (e) NW and (f) Central regions of China from 2015 to 2018 between CAQIEI and previous inventories.**

[Figure]

**Figure S18: Timeseries of observed (red dots) and simulated (blue line) monthly values of (a) zonal wind, (b) meridional**

**wind, (c) temperature, (d) relative humidity and (e) precipitation over NCP region from Jan 2013 to Dec 2020.**

[Figure]

Figure S19: Same as in Figure S18 but over the NE region.

**SE**

[Figure]

**Figure S20: Same as in Figure S18 but over the SE region.**

**SW**

[Figure]

Figure S21: Same as in Figure S18 but over the SW region.

[Figure]

**Figure S22: Same as in Figure S18 but over the NW region.**

**Central**

[Figure]

**Figure S23: Same as in Figure S18 but over the Central region.**

[Figure]

**Figure S24: Spatial distribution of meteorological observation sites used in the evaluation of meteorology simulations over different regions of China**

**Figure S25: Timeseries of observed (black lines) and simulated concentrations of different air pollutants in China driven by CAQIEI (red lines) and MEIC-HTAPv3 (blue lines) over different regions of China.**